# On the Learn-to-Optimize Capabilities of Transformers in In-Context Sparse Recovery

**Renpu Liu**[1]*, **Ruida Zhou**[2]*, **Cong Shen**[1], **Jing Yang**[1]†
[1]University of Virginia, [2]University of California, Los Angeles
{pzw7bx, cong, yangjing}@virginia.edu
ruida@g.ucla.edu

## Abstract

An intriguing property of the Transformer is its ability to perform in-context learning (ICL), where the Transformer can solve different inference tasks without parameter updating based on the contextual information provided by the corresponding input-output demonstration pairs. It has been theoretically proved that ICL is enabled by the capability of Transformers to perform gradient-descent algorithms (Von Oswald et al., 2023a; Bai et al., 2024). This work takes a step further and shows that Transformers can perform learning-to-optimize (L2O) algorithms. Specifically, for the ICL sparse recovery (formulated as LASSO) tasks, we show that a $K$-layer Transformer can perform an L2O algorithm with a provable *convergence rate linear in $K$*. This provides a new perspective explaining the superior ICL capability of Transformers, even with only a few layers, which cannot be achieved by the standard gradient-descent algorithms. Moreover, unlike the conventional L2O algorithms that require the measurement matrix involved in training to match that in testing, the trained Transformer is able to solve sparse recovery problems generated with different measurement matrices. Besides, Transformers as an L2O algorithm can leverage structural information embedded in the training tasks to accelerate its convergence during ICL, and generalize across different lengths of demonstration pairs, where conventional L2O algorithms typically struggle or fail. Such theoretical findings are supported by our experimental results.

## 1 Introduction

Since its introduction in Vaswani et al. (2017), Transformers have become the backbone in various fields such as natural language processing (Radford, 2018; Devlin, 2018), computer vision (Dosovitskiy, 2020) and reinforcement learning (Chen et al., 2021), significantly influencing subsequent research and applications. A notable capability of Transformers is their good performance for in-context learning (ICL) (Brown et al., 2020), i.e., without further parameter updating, Transformers can perform new inference tasks based on the contextual information embedded in example input-output pairs contained in the prompt. Such ICL capability facilitates state-of-the-art few-shot performances across a multitude of tasks, such as reasoning and language understanding tasks in natural language processing (Chowdhery et al., 2023), in-context dialog generation (Thoppilan et al., 2022) and in-context linear regression (Garg et al., 2022; Fu et al., 2023).

Given the significance of transformers' ICL capabilities, extensive research efforts have been directed toward understanding the mechanisms behind their ICL performance. In this context, the ICL capability of a pre-trained Transformer is understood as the Transformer's implicit implementation of learning algorithms during the forward pass. Von Oswald et al. (2023a), Dai et al. (2022), and Bai et al. (2024) suggest that these learning algorithms closely approximate gradient-descent-based optimizers, thus making the Transformer a *universal* solver for various ICL tasks. Specifically, these works demonstrate that Transformers can approximate gradient descent steps implicitly through

---

*These authors contributed equally to this work.
†Corresponding author.

some specific constructions of their parameters, enabling them to adapt to new data points during inference without explicit re-training.

However, it is well known that gradient-descent-based algorithms are not efficient in solving complicated optimization problems and thus may not be sufficient to explain the superior performance of Transformers on a plethora of ICL tasks. A recent work (Ahn et al., 2024) suggests that instead of gradient descent, Transformers actually perform pre-conditioned gradient descent during ICL. In other words, it learns a pre-conditioner during pre-training and then utilizes it during ICL to expedite the optimization process. Von Oswald et al. (2023b) recently demonstrates that the forward pass of a trained transformer can implement meta-optimization algorithms, i.e., it can implicitly define internal objective functions and then optimize these objectives to generate predictions. Similarly, Zhang et al. (2023) show that a mesa-optimizer embedding the covariance matrix of input data can efficiently solve linear regression tasks. Such interpretation of the ICL mechanism shares the same essence as Learning-to-Optimize (L2O) algorithms, and motivates the following hypothesis:

*Transformer does not simply implement a universal optimization algorithm during ICL. Rather, it extracts useful information from the given dataset during pre-training and then utilizes such information to generate an optimization algorithm that best suits the given ICL task.*

In this paper, we examine this hypothesis through the lens of *in-context sparse recovery*. Sparse recovery is a classical signal processing problem that is of significant practical interest across various domains, such as compressive sensing in medical imaging (Shen et al., 2017) and spectrum sensing (Elad, 2010). Recent works show that Transformers are able to implement gradient descent-based algorithms with sublinear convergence rates for in-context sparse recovery (Bai et al., 2024; Chen et al., 2024b). However, empirical findings indicate that Transformers can solve in-context sparse recovery more efficiently than gradient descent-based approaches (Bai et al., 2024). Meanwhile, there exists a plethora of L2O algorithms that solve the classical sparse recovery problem efficiently with linear convergence guarantees (Gregor and LeCun, 2010; Chen et al., 2018; Liu and Chen, 2019). Therefore, examining the L2O capabilities of Transformers in solving the in-context sparse recovery task becomes a promising direction and may serve as a perfect example to validate our hypothesis. Our main contributions are as follows.

- First, we demonstrate that Transformers can implement an L2O algorithm for in-context sparse recovery, and theoretically prove that a $K$-layer Transformer as an L2O algorithm can recover the underlying sparse vector in-context at a convergence rate linear in $K$. The linear convergence results in this work significantly improve the state-of-the-art convergence results for in-context sparse recovery and validate our previous hypothesis.

- Second, we show that the Transformer as an L2O algorithm can actually outperform traditional L2O algorithms for sparse recovery in several aspects: 1) It does not require the measurement matrices involved in training to be the same, which is in stark contrast to traditional L2O algorithms (Gregor and LeCun, 2010; Chen et al., 2018; Liu and Chen, 2019) for sparse recovery, and allows more flexibility to solve various in-context sparse recovery tasks. 2) It allows different numbers of measurements (i.e., prompt length) used for in-context sparse recovery, with guaranteed recovery performance as long as the number of measurements is sufficiently large. 3) It can extract structural properties of the underlying sparse vectors from the training data and utilize them to expedite its ICL convergence.

- We compare the ICL performances of Transformers with traditional iterative algorithms and L2O algorithms for sparse recovery empirically. Our experimental results indicate that Transformers substantially outperform traditional gradient-descent-based iterative algorithms, and achieve comparable performances compared with L2O algorithms that are trained and tested using data generated with the same measurement matrix. This supports our claim that Transformers can implement L2O algorithms during ICL. Besides, Transformers also demonstrate remarkable generalization capability when the measurement matrix varies, and achieve accelerated convergence when additional structure is imposed on the underlying sparse vectors, supporting our theoretical findings.

## 2 RELATED WORKS

**ICL Mechanism for Transformers.** Brown et al. (2020) first show that GPT-3, a Transformer-based LLM, can perform new tasks from input-output pairs without parameter updates, suggesting its ICL ability. This intriguing phenomenon of Transformers has attracted many attentions, leading to various interpretations and hypotheses about its underlying mechanism. For example, Han et al. (2023) empirically hypothesize that Transformers perform kernel regression with internal representations when facing in-context examples, and Fu et al. (2023) empirically show that Transformers learn to implement an algorithm similar to iterative Newton's method for ICL tasks.

To better understand the ICL mechanism in large Transformers, existing works aim to demonstrate the Transformer's *capability* for ICL by construction, e.g., showing that Transformers can perform gradient-based algorithms to solve ICL tasks by iteratively performing gradient descent layer by layer. In this category, Akyürek et al. (2022) show that by construction, Transformers can implement gradient descent-based algorithms for linear regression problems. Von Oswald et al. (2023a) construct explicit weights for a Transformer, claiming it can perform gradient descent on linear and non-linear regression tasks. Bai et al. (2024) provide constructions such that Transformers can make selections between different gradient-based algorithms. Ahn et al. (2024) reason the ICL capability of Transformers to their ability to implement a pre-conditioned gradient descent for linear regression tasks, where the pre-condition matrix is learned from pre-training. Recently, Von Oswald et al. (2023b) demonstrates that the forward pass of a trained transformer can implement meta-optimization algorithms, i.e., it can implicitly define internal objective functions and then optimize these objectives to generate predictions. Similarly, Zhang et al. (2023) show that a mesa-optimizer embedding the covariance matrix of input data can efficiently solve linear regression tasks. Our work belongs to this category, where we construct a Transformer structure that can implement an L2O algorithm to effectively solve the in-context sparse recovery problem.

**L2O Algorithms for Sparse Recovery.** The L2O framework, as summarized in the review paper (Chen et al., 2022), is an optimization paradigm that develops an optimization method (i.e., a solver) by training across a set of similar problems (tasks) sampled from a task distribution. While the training process is often offline and time-consuming, the objective of L2O is to improve the optimization efficiency and accuracy when the method is deployed online and any new task sampled from the same distribution is encountered.

As a general optimization framework, L2O has demonstrated its advantage over classical static optimization frameworks in various optimization problems and applications. In this work, we focus on the L2O algorithms relevant to sparse recovery here and leave more discussions on general L2O in Appendix A. Sparse recovery, typically formulated as a least absolute shrinkage and selection operator (LASSO), has many important applications like magnetic resonance imaging (Meng et al., 2023) and stock market forecasting (Roy et al., 2015), thus motivates the design of efficient algorithms. E.g., the iterative soft-thresholding algorithm (ISTA) (Daubechies et al., 2004) is proposed to solve LASSO and improves over the standard gradient descent algorithm. Motivated by the ISTA structure, Gregor and LeCun (2010) introduce the Learned ISTA (LISTA), a feedforward neural network that incorporates trainable matrices into ISTA updates. Chen et al. (2018) and Liu and Chen (2019) further propose LISTA-Partial Weight Coupling (LISTA-CP) and Analytic LISTA (ALISTA) with fewer trainable parameters, making them easier to train. They also provide theoretical analyses demonstrating a linear convergence rate.

We note that for these existing LISTA-type algorithms, both the training and testing tasks (instances) are randomly generated by *fixing the measurement matrix but varying the underlying sparse vectors*. In general, they cannot handle cases where the training instances are generated with varying measurement matrices, or the measurement matrices during training and testing do not match.

## 3 PRELIMINARIES

**Notations.** For matrix $\mathbf{X}$, we use $[\mathbf{X}]_{p:q,r:s}$ to denote the submatrix that contains rows $p$ to $q$ and columns $r$ to $s$, and we use $[\mathbf{X}]_{:,i}$ and $[\mathbf{X}]_{j,:}$ to denote the $i$-th column and $j$-th row of $\mathbf{X}$ respectively. In some places, we also use $[\mathbf{X}]_i$ to denote its $i$-th column for convenience. We use $\|\mathbf{X}\|_F$ to denote its Frobenius norm. For vector $\mathbf{x}$, we use $\|\mathbf{x}\|_1$, $\|\mathbf{x}\|$ and $\|\mathbf{x}\|_\infty$ to denote its $\ell_1$, $\ell_2$ and $\ell_\infty$ norms,

respectively. We denote by $\mathbb{1}_d$ and $\mathbf{0}_d$ the $d$-dimensional all-1 and all-0 column vectors, respectively. $\mathbb{1}_{a \times b}$ and $\mathbf{0}_{a \times b}$ denote the all-1 and all-0 matrices of size $a \times b$, respectively.

## 3.1 TRANSFORMER ARCHITECTURE

In this work, we consider the decoder-based Transformer architecture (Vaswani et al., 2017), where each attention layer is masked by a decoder-based attention mask and followed by a multi-layer perception (MLP) layer.

**Definition 3.1** (Masked attention layer). *Denote an $M$-head masked attention layer parameterized by $\{(\mathbf{V}_m, \mathbf{Q}_m, \mathbf{K}_m)_{m \in [M]}\}$ as $\mathrm{Attn}_{\{(\mathbf{V}_m, \mathbf{Q}_m, \mathbf{K}_m)\}}(\cdot)$, where $\mathbf{V}_m, \mathbf{Q}_m, \mathbf{K}_m \in \mathbb{R}^{D \times D}$, $\forall m \in [M]$. Then, given an input sequence $\mathbf{H} \in \mathbb{R}^{D \times (N+1)}$, the output sequence of the attention layer is*

$$\mathrm{Attn}_{\{(\mathbf{V}_m, \mathbf{Q}_m, \mathbf{K}_m)\}}(\mathbf{H}) = \mathbf{H} + \sum_{m=1}^{M} (\mathbf{V}_m \mathbf{H}) \times \mathrm{mask}\Big(\sigma\big((\mathbf{K}_m \mathbf{H})^\top (\mathbf{Q}_m \mathbf{H})\big)\Big),$$

*where $\mathrm{mask}(\mathbf{M})$ satisfies $[\mathrm{mask}(\mathbf{M})]_{i,j} = \frac{1}{j} \mathbf{M}_{i,j}$ if $i \leq j$ and $[\mathrm{mask}(\mathbf{M})]_{i,j} = 0$ otherwise. $\sigma(\cdot)$ denotes the activation function.*

**Definition 3.2** (MLP layer). *Given $\mathbf{W}_1 \in \mathbb{R}^{D' \times D}$, $\mathbf{W}_2 \in \mathbb{R}^{D \times D'}$ and a bias vector $\mathbf{b} \in \mathbb{R}^{D'}$, an MLP layer following the decoder attention layer, denoted as $\mathrm{MLP}_{\{\mathbf{W}_1, \mathbf{W}_2, \mathbf{b}\}}$, maps each token in the input sequence (i.e, each column $\mathbf{h}_i$ in $\mathbf{H} \in \mathbb{R}^{D \times N}$) to another token as*

$$\mathrm{MLP}_{\{\mathbf{W}_1, \mathbf{W}_2, \mathbf{b}\}}(\mathbf{h}_i) = \mathbf{h}_i + \mathbf{W}_2 \sigma(\mathbf{W}_1 \mathbf{h}_i + \mathbf{b}),$$

*where $\sigma$ is the non-linear activation function.*

In this work, we set the activation function in Definition 3.1 and Definition 3.2 as the element-wise ReLU function. Next, we define the one-layer decoder-based Transformer structure.

**Definition 3.3** (Transformer layer). *A one-layer decoder-based Transformer is parameterized by $\Theta := \{\mathbf{W}_1, \mathbf{W}_2, \mathbf{b}, (\mathbf{V}_m, \mathbf{Q}_m, \mathbf{K}_m)_{m \in [M]}\}$, denoted as $\mathrm{TF}_\Theta$. Therefore, give input sequence $\mathbf{H} \in \mathbb{R}^{d \times N}$, the output sequence is:*

$$\mathrm{TF}_\Theta(\mathbf{H}) = \mathrm{MLP}_{\{\mathbf{W}_1, \mathbf{W}_2, \mathbf{b}\}}\Big(\mathrm{Attn}_{\{(\mathbf{V}_m, \mathbf{Q}_m, \mathbf{K}_m)\}}(\mathbf{H})\Big).$$

## 3.2 IN-CONTEXT LEARNING BY TRANSFORMERS

For an in-context learning (ICL) task, a trained Transformer is given an ICL instance $\mathcal{I} = (\mathcal{D}, \mathbf{x}_{N+1})$, where $\mathcal{D} = \{(\mathbf{x}_i, y_i)\}_{i \in [N]}$ and $\mathbf{x}_{N+1}$ is a query. Here, $\mathbf{x}_i \in \mathbb{R}^d$ is an in-context example, and $y_i$ is the corresponding label for $\mathbf{x}_i$. We assume $y_i = f_{\boldsymbol{\beta}}(\mathbf{x}_i) + \epsilon_i$, where $\epsilon_i$ is an added random noise, and $f_{\boldsymbol{\beta}}$ is a deterministic function parameterized by $\boldsymbol{\beta}$. Unlike conventional supervised learning, for each ICL instance, $\boldsymbol{\beta} \sim P_\beta$, i.e., it is randomly sampled from a distribution $P_\beta$.

To perform ICL in a Transformer, we first embed the ICL instance into an input sequence $\mathbf{H} \in \mathbb{R}^{D \times N'}$. The Transformer then generates an output sequence $\mathrm{TF}(\mathbf{H})$ with the same size as $\mathbf{H}$, based on which a prediction $\widehat{y}_{N+1}$ is generated through a read-out function $F$, i.e., $\widehat{y}_{N+1} = F(\mathrm{TF}(\mathbf{H}))$. The objective of ICL is then to ensure that $\widehat{y}_{N+1}$ closely approximates the target value $y_{N+1} = f_{\boldsymbol{\beta}}(\mathbf{x}_{N+1}) + \epsilon_{N+1}$ for any ICL instance.

When a Transformer is pre-trained for ICL, it first samples a large set of ICL instances. For each instance, the Transformer generates a prediction $\widehat{y}_{N+1}$ and calculates the prediction loss by comparing it with $y_{N+1}$ using a proper loss function. The training loss is the aggregation of all prediction losses for every ICL instance used in pre-training, and the Transformer is trained to minimize this training loss.

Previous studies about the mechanism of how Transformers performs in-context learning have attracted a lot of attention recently. To start with, it is believed that the ICL capability is due to the Transformer's implicit implementation of learning algorithms in the forward pass. Von Oswald et al. (2023a), Dai et al. (2022), and Bai et al. (2024) suggest that these learning algorithms closely approximate gradient-descent-based optimizers, thus making the Transformer a universal solver for various

ICL tasks. A recent work (Ahn et al., 2024) suggests that instead of gradient descent, Transformers actually perform pre-conditioned gradient descent for in-context least square linear regression. In general, these results corroborate the claim that the mechanism of the Transformer in-context learning is a L2O algorithm. We study this perspective and further provide the evidence supporting this claim by considering a more complicated in-context problem: in-context sparse recovery.

## 4 TRANSFORMER AS A LISTA-TYPE ALGORITHM FOR IN-CONTEXT SPARSE RECOVERY

In this section, we demonstrate that a decoder-based Transformer *can* implement a novel LISTA-type L2O algorithm, specifically LISTA-VM, as detailed in Theorem 4.1, for in-context sparse recovery. We begin by formally defining the in-context sparse recovery problem.

### 4.1 IN-CONTEXT SPARSE RECOVERY

Sparse recovery is a fundamental problem in fields such as compressed sensing, signal denoising, and statistical model selection. The core concept of sparse recovery is that a high-dimensional sparse signal can be inferred from very few linear observations if certain conditions are satisfied. Specifically, it aims to identify an $S$-sparse vector $\boldsymbol{\beta}^* \in \mathbb{R}^d$ from its noisy linear observations $\mathbf{y} = \mathbf{X}\boldsymbol{\beta}^* + \boldsymbol{\epsilon}$, where $\mathbf{X} \in \mathbb{R}^{N \times d}$ is a measurement matrix, $\boldsymbol{\epsilon} \in \mathbb{R}^N$ is an isometric Gaussian noise vector with mean vector $\mathbf{0}_N$ and covariance matrix $\mathbf{I}_{N \times N}$. Typically, we assume $d \gg N$, which is the so-called under-determined case. One common assumption for the measurement matrix $\mathbf{X}$ (Pitaval et al., 2015; Ge et al., 2017; Zhu et al., 2021) is that each row of the matrix is independently sampled from an isometric sub-Gaussian distribution with zero mean and covariance matrix $\mathrm{diag}(\sigma_1^2 \cdots, \sigma_d^2)$, denoted as $P_\mathbf{x}$. which guarantees the critical *restricted isometry property* under mild conditions on the sparsity level (Candes and Tao, 2007). In this work, we also assume $\boldsymbol{\beta}^*$ is randomly sampled from a distribution $P_\boldsymbol{\beta}$, which admits an $S$-sparse vector with random support.

A popular approach to tackling sparse recovery is the least absolute shrinkage and selection operator (LASSO), which aims to find the optimal sparse vector $\boldsymbol{\beta} \in \mathbb{R}^d$ that minimizes the following loss:

$$\mathcal{L}(\boldsymbol{\beta}) = \frac{1}{2}\|\mathbf{y} - \mathbf{X}\boldsymbol{\beta}\|_2^2 + \alpha\|\boldsymbol{\beta}\|_1.$$

Here $\alpha$ is a coefficient controlling the sparsity penalty. We denote the transpose of the $i$-th row in $\mathbf{X}$ by $\mathbf{x}_i$, i.e., $\mathbf{x}_i = [\mathbf{X}^\top]_{:,i}$.

In this work, we study how Transformers solve the sparse recovery problem *in context*. For the pre-training process, a set of in-context sparse recovery instances $\{(\mathbf{X}^j, \mathbf{y}^j, \mathbf{x}_{N+1}^j, y_{N+1}^j)\}_{j=1}^{N_{\mathrm{train}}}$ is generated according to the relationship $y_n^j = (\mathbf{x}_n^j)^\top \boldsymbol{\beta}^j + \epsilon_n^j$ for $j \in [N_{\mathrm{train}}]$ and $n \in [N+1]$, where $\boldsymbol{\beta}^j \sim P_\beta$, $\mathbf{x}_n^j \sim P_\mathbf{x}$, and $\epsilon_n^j \sim P_\epsilon$, respectively. Given the set of training instances, a pre-trained Transformer is obtained by minimizing a certain loss function.

After pre-training, during the inference process for ICL, a sparse recovery instance $(\mathbf{X}, \mathbf{y}, \mathbf{x}_{N+1})$ is sampled randomly according to the same distributions as in the pre-training, and the Transformer then aims to predict $y_{N+1}$ using the input $(\mathbf{X}, \mathbf{y}, \mathbf{x}_{N+1})$ without any further parameter updating.

### 4.2 CLASSICAL ALGORITHMS

Gradient descent is known to struggle in solving the LASSO problem due to its inefficiency in effectively handling the sparsity constraint (Chen et al., 2018). This inefficiency has led to the development of more specialized algorithms that can better address the unique challenges posed by the LASSO formulation. A popular approach to solving the LASSO problem is the Iterative Shrinkage Thresholding Algorithm (ISTA). Starting with a fixed initial point $\boldsymbol{\beta}^{(1)}$, the update rule in the $k$-th iteration is given by

$$\boldsymbol{\beta}^{(k+1)} = \mathcal{S}_{\alpha/L}\left(\boldsymbol{\beta}^{(k)} - \frac{1}{L}\mathbf{X}^\top(\mathbf{X}\boldsymbol{\beta}^{(k)} - \mathbf{y})\right).$$

Here, $\mathcal{S}_{\alpha/L}$ is the soft-thresholding function defined as $[\mathcal{S}_{\alpha/L}(\mathbf{x})]_i = \mathrm{sign}([\mathbf{x}]_i)\max\{0, |[\mathbf{x}]_i| - \alpha/L\}$, and $L$ is typically chosen as the largest eigenvalue of $\mathbf{X}^\top\mathbf{X}$ (Chen et al., 2018; Liu and Chen, 2019).

Generally, for any ground-truth sparse vector $\boldsymbol{\beta}^*$ and any given $\mathbf{X}$, ISTA converges at a *sublinear* rate (Beck and Teboulle, 2009). The sublinear convergence rate of ISTA is considered inefficient, which has led to the development of various LISTA-type L2O algorithms, such as LISTA (Gregor and LeCun, 2010), LISTA-CP (Chen et al., 2018), and ALISTA (Liu and Chen, 2019). These algorithms learn the weights in the matrices in ISTA rather than fixing them.

Among them, LISTA-CP is one state-of-the-art (SOTA) method that has been well-studied. The update rule in the $k$-th iteration of LISTA-CP can be expressed as

$$\boldsymbol{\beta}^{(k+1)} = \mathcal{S}_{\theta^{(k)}}\Big(\boldsymbol{\beta}^{(k)} - (\mathbf{D}^{(k)})^\top(\mathbf{X}\boldsymbol{\beta}^{(k)} - \mathbf{y})\Big), \tag{4.1}$$

where $\{\theta^{(k)}, \mathbf{D}^{(k)}\}$ are learnable parameters. Compared with ISTA with fixed parameters, LISTA-CP obtains $\{\theta^{(k)}, \mathbf{D}^{(k)}\}$ through pre-training. Specifically, with a fixed measurement matrix $\mathbf{X}$, it randomly samples $n$ $S$-sparse vectors $\{\boldsymbol{\beta}_j\}_{j=1}^n \sim P_\beta$ and generates $\{\mathbf{y}_j\}_{j=1}^n$, which is then utilized to optimize $\{\theta^{(k)}, \mathbf{D}^{(k)}\}$ by minimizing the total predicting loss for $\{\boldsymbol{\beta}_j\}_j$. Chen et al. (2018) show that, for the same measurement matrix $\mathbf{X}$, given any random instance $(\boldsymbol{\beta}^*, \mathbf{y})$, a pre-trained LISTA-CP will converge to the ground-truth $\boldsymbol{\beta}^*$ *linearly in $K$* under certain necessary conditions on $\mathbf{X}$.

### 4.3 Transformer Can Provably Perform LISTA-type Algorithms

Noting that LISTA-type algorithms can efficiently solve sparse recovery problems, in this section, we argue that a trained Transformer can implement a LISTA-type algorithm and efficiently solve a sparse recovery problem in context. To distinguish the algorithm implemented by the Transformer with the classical LISTA-type algorithms, we term it as LISTA with Varying Measurements (LISTA-VM). Towards this end, we provide an explicit construction of a $K$-layer decoder-based Transformer as follows. A $K$-layer Transformer is the concatenation of $K$ blocks, where each block comprises a self-attention layer followed by an MLP layer. The input to the first self-attention layer, denoted as $\mathbf{H}^{(1)}$, is an embedding of the given in-context sparse recovery instance $\mathcal{I} = (\mathbf{X}, \mathbf{y}, \mathbf{x}_{N+1})$.

**Embedding.** Given an in-context sparse recovery instance $\mathcal{I} = (\mathbf{X}, \mathbf{y}, \mathbf{x}_{N+1})$ we embed the instance into an input sequence $\mathbf{H}^{(1)} \in \mathbb{R}^{(2d+2) \times (2N+1)}$ as follows:

$$\mathbf{H}^{(1)}(\mathcal{I}) = \begin{bmatrix} \mathbf{x}_1 & \mathbf{x}_1 & \cdots & \mathbf{x}_N & \mathbf{x}_N & \mathbf{x}_{N+1} \\ 0 & y_1 & \cdots & 0 & y_N & 0 \\ \boldsymbol{\beta}_1^{(1)} & \boldsymbol{\beta}_2^{(1)} & \cdots & \boldsymbol{\beta}_{2N-1}^{(1)} & \boldsymbol{\beta}_{2N}^{(1)} & \boldsymbol{\beta}_{2N+1}^{(1)} \\ 1 & 0 & \cdots & 1 & 0 & 1 \end{bmatrix}, \tag{4.2}$$

where $\{\boldsymbol{\beta}_i^{(1)}\}_{i \in [2N+1]} \in \mathbb{R}^d$ are implicit parameter vectors initialized as $\mathbf{0}_d$, and $\mathbf{x}_i$ is the $i$-th column of the transposed measurement matrix, i.e, $[\mathbf{X}^\top]_{:,i}$. We note that a similar embedding structure is adopted in Bai et al. (2024).

**Self-attention layer.** The self-attention layer takes as input a sequence of embeddings and outputs a sequence of embeddings of the same length. Let the $K$-layer decoder-based Transformer feature four attention heads uniquely indexed as $+1$, $-1$, $+2$, and $-2$. We construct these heads according to the structure specified in Appendix C.1. This construction ensures that the self-attention layer performs the $\boldsymbol{\beta}$ updating inside the soft-thresholding function in Equation (4.1). Furthermore, with our construction, the learnable matrix $\mathbf{D}^{(k)}$ in LISTA-CP becomes *context-dependent* instead of fixed. In the update rule implemented by the self-attention layer, this matrix becomes $\mathbf{D}^{(k)} = \frac{1}{2N+1}\mathbf{X}(\mathbf{M}^{(k)})^\top$, where $\mathbf{M}^{(k)} \in \mathbb{R}^{d \times d}$ is fixed.

**MLP layer.** For the MLP layer following the $k$-th self-attention layer, it functions as a feedforward neural network that takes the output of the self-attention layer as its input, and outputs a transformed sequence of the embeddings. Recall that $\mathbf{h}_i$ is the $i$-th column in an embedding sequence $\mathbf{H}$. We parameterize $(\mathbf{W}_1, \mathbf{W}_2, \mathbf{b})$ in the $k$-th MLP layer to let it function as a partial soft-threshold function:

$$\mathrm{MLP}(\mathbf{h}_i) = \begin{bmatrix} [\mathbf{h}_i]_{1:d+1} \\ \mathcal{S}_{\theta^{(k)}}([\mathbf{h}_i]_{d+2:2d+1}) \\ [\mathbf{h}_i]_{2d+2} \end{bmatrix}, \tag{4.3}$$

where $\mathcal{S}_{\theta^{(k)}}$ is the soft-threshold function. Essentially, the soft-threshold function is effectively implemented by the MLP layer utilizing the $\mathrm{ReLU}$ activation. This can be realized by combining

$-\text{ReLU}(x)$, $\text{ReLU}(-x)$, $\text{ReLU}(x - \theta)$, $-\text{ReLU}(-x + \theta)$, and $x$. The implementation details of the MLP layer can be found in Appendix C.1.

**Read-out function.** Given the output sequence of the Transformer $\text{TF}_{\boldsymbol{\Theta}}(\mathbf{H}^{(1)})$, to obtain the estimation $\widehat{y}_{N+1}$, it is necessary to *read out* from the output sequence. In this work, we consider two types of read-out functions:

**Definition 4.1** (Linear read-out). $\mathcal{F}_{linear}$ *is defined as the class of linear readout functions such that*

$$\mathcal{F}_{linear} = \{F(\cdot) \mid F(\mathbf{h}) = \mathbf{v}^\top \mathbf{h}, \mathbf{v} \in \mathbb{R}^D\}.$$

**Definition 4.2** (Query read-out). $\mathcal{F}_{query}$ *is defined as the class of explicit quadratic readout functions such that*

$$\mathcal{F}_{query} = \{F(\cdot\,; \widetilde{\mathbf{X}}) \mid F(\mathbf{h}, i; \widetilde{\mathbf{X}}) = \mathbf{h}^\top \mathbf{V} \widetilde{\mathbf{X}}_{\lfloor \frac{i+1}{2} \rfloor, :}, \mathbf{V} \in \mathbb{R}^{D \times d}\},$$

*where $i \in [N + 1]$ is an index number and $\widetilde{\mathbf{X}} = [\mathbf{X}^\top \; \mathbf{x}_{N+1}]^\top$.*

Given a readout function $F_{\mathbf{v}} \in \mathcal{F}_{\text{linear}}$ parameterized by $\mathbf{v}$ or $F_{\mathbf{V}} \in \mathcal{F}_{\text{query}}$ parameterized by $\mathbf{V}$, the estimation $\widehat{y}_i$ obtained by the $K$-layer Transformer is $\widehat{y}_i = F_{\mathbf{v}}(\mathbf{h}_{2i+1}^{K+1})$ or $\widehat{y}_i = F_{\mathbf{V}}(\mathbf{h}_{2i+1}^{K+1}, 2i - 1)$ respectively.

Before we formally present our main results, we introduce the following assumptions.

**Assumption 1.** *For $\mathbf{x} \sim P_{\mathbf{x}}$ and $\boldsymbol{\beta}^* \sim P_{\boldsymbol{\beta}}$, we assume $\|\mathbf{x}\| \leq b_{\mathbf{x}}$ and $\|\boldsymbol{\beta}^*\|_1 \leq b_{\boldsymbol{\beta}}$ almost surely. Besides, we consider the noiseless scenario where $\boldsymbol{\epsilon} = \mathbf{0}$.*

We note that the boundedness assumption over $\mathbf{x}$ and $\boldsymbol{\beta}^*$ ensures the robustness of the Transformer and prevents it from blowing up under ill conditions. A similar assumption is adopted in Bai et al. (2024). The noiseless assumption is for ease of analysis and is common in the analysis of Transformers (Ahn et al., 2024; Fu et al., 2023; Bai et al., 2024). We note that the following Theorem 4.1 can be straightforwardly extended to the noisy case when the noise is bounded.

We denote the input sequence to the $k$-th self-attention layer as $\mathbf{H}^{(k)}$, and use $\boldsymbol{\beta}_{2n+1}^{(k+1)}$ to represent the vector $[\mathbf{H}^{(k+1)}]_{d+1:2d+1, 2n+1}$. Then, we state the following theorem.

**Theorem 4.1** (Equivalence between ICL and LISTA-VM). *With the Transformer structure described above, under Assumption 1, there exists a set of parameters in the Transformer so that for any $k \in [1 : K]$, $n \in [N]$, we have*

$$\boldsymbol{\beta}_{2n+1}^{(k+1)} = \mathcal{S}_{\theta^{(k)}}\left(\boldsymbol{\beta}_{2n+1}^{(k)} - \frac{1}{2n+1}\mathbf{M}^{(k)}[\mathbf{X}]_{1:n,:}^\top([\mathbf{X}]_{1:n,:}\boldsymbol{\beta}_{2n+1}^{(k)} - \mathbf{y}_{1:n})\right), \qquad (4.4)$$

*where $\mathbf{M}^{(k)} \in \mathbb{R}^{d \times d}$ is embedded in the $k$-th Transformer layer.*

The proof of Theorem 4.1 is detailed in Appendix C.2.

**Remark 1.** *As mentioned above, matrix $\mathbf{D}^{(k)}$ in the update rule of LISTA-CP in Equation (4.1) is learned during pre-training and remains fixed across different in-context sparse recovery instances. As a result, LISTA-CP requires the measurement matrix $\mathbf{X}$ to stay the same during pre-training and inference. In contrast, if we denote $\mathbf{D}_n^{(k)} = \frac{1}{2n+1}[\mathbf{X}]_{1:n,:}(\mathbf{M}^{(k)})^\top$, then within the update rule of the LISTA-VM algorithm implemented by the Transformer, as detailed in Equation (4.4), the matrix $\mathbf{D}_n^{(k)}$ depends on a fixed matrix $\mathbf{M}^{(k)}$ after pre-training as well as on the measurement matrix $\mathbf{X}$ during inference. As a result, the Transformer can adaptively update $\mathbf{D}_n^{(k)}$ for instances with different $\mathbf{X}$'s, enabling more flexibility and improved performance for the ICL tasks.*

## 5 PERFORMANCE OF TRANSFORMERS FOR IN-CONTEXT SPARSE RECOVERY

In this section, we demonstrate the effectiveness of the constructed Transformer in implementing the LISTA-VM algorithm and solving in-context sparse recovery problems. We first show that the LISTA-VM algorithm implemented by the Transformer recovers the underlying sparse vector in context at a convergence rate linear in $K$. We then demonstrate that the Transformer can accurately predict $y_{N+1}$ at the same time.

## 5.1 Sparse Vector Estimation

**Theorem 5.1** (Convergence of ICL). *Let $\delta \in (0,1)$, $N_0 = 8(4S-2)^2 \frac{\log d + \log S - \log \delta}{c}$, $\alpha_n = -\log\left(1 - \frac{2}{3}\gamma + \gamma(2S-1)\sqrt{\frac{\log d - \log \delta}{nc}} + \sqrt{\frac{\log S - \log \delta}{nc}}\right)$, where $c$ is a positive constant and $\gamma$ is a positive constant satisfies $\gamma \leq \frac{3}{2}$. For a $K$-layer Transformer model with the structure described in Section 4.3, under Assumption 1, there exists a set of parameters such that for any randomly generated sparse recovery instance and any $n \in [N_0 : N]$, with probability at least $1 - \delta$, we have*

$$\left\| \boldsymbol{\beta}_{2n+1}^{(K+1)} - \boldsymbol{\beta}^* \right\| \leq b_{\boldsymbol{\beta}} e^{-\alpha_n K}.$$

*Main challenge and key ideas of the proof.* Similar to the proofs in Chen et al. (2018) and Liu and Chen (2019), the core step in proving convergence is to ensure that $\mathbf{D}^{(k)}$ exhibits small coherence with $\mathbf{X}$, i.e., $(\mathbf{D}^{(k)})^\top \mathbf{X} \approx \mathbf{I}_{d \times d}$. In Chen et al. (2018) and Liu and Chen (2019), such a $\mathbf{D}^{(k)}$ is obtained by minimizing the generalized mutual coherence to a *fixed* $\mathbf{X}$. However, this results in poor generalization across different $\mathbf{X}$'s. In our proof, we consider $\mathbf{X}$ as a random matrix and leverage its sub-Gaussian properties to prove that if $\mathbf{D}^{(k)} = \mathbf{X}(\mathbf{M}^{(k)})^\top$, where $\mathbf{M}^{(k)}$ is associated with the covariance of $\mathbf{X}$, then $\mathbf{D}^{(k)}$ will have small coherence with $\mathbf{X}$ with high probability. We defer the detailed proof of Theorem 5.1 to Appendix D.1. $\qquad\square$

**Remark 2** (Linear convergence rate). *The linear convergence rate demonstrated in Theorem 5.1 is due to the incorporation of curvature information into the update rule. Specifically, the learned matrices $\mathbf{M}^{(k)}$ serve as approximations of the inverse Hessian. By leveraging the statistical properties of the problem, these matrices effectively accelerate convergence, enabling the Transformer to mimic second-order optimization methods. This allows the Transformer to overcome the traditional limitations of first-order methods, which are typically restricted to a sublinear $\mathcal{O}(1/K)$ convergence rate unless additional assumptions are introduced (Nesterov, 2005; Nemirovskij and Yudin, 1983).*

**Remark 3** (Generalization across measurement matrix $\mathbf{X}$). *Theorem 5.1 shows that for any $\mathbf{X}$ satisfying Assumption 1, the Transformer can estimate the ground-truth sparse vector $\boldsymbol{\beta}^*$ in-context at a convergence rate linear in $K$. This is in stark contrast to traditional LISTA-CP type of algorithms, which only work for fixed $\mathbf{X}$. Such generalization is enabled by the input-dependent matrices $\{\mathbf{D}^{(k)}\}_k$. Besides, we also note that $\boldsymbol{\beta}_{2n+1}^{(K+1)}$ only depends on $\mathbf{x}_1, \ldots, \mathbf{x}_n$. This implies that even if the measurement matrix $\mathbf{X}$ is of dimension $n \times d$ instead of $N \times d$, when $n \in [N_0, N]$, the Transformer can still recover $\boldsymbol{\beta}^*$ accurately. Such results demonstrate the robustness of Transformers to variations in in-context sparse recovery tasks.*

**Remark 4** (Effective utilization of the hidden patterns in ICL tasks). *We note that the Transformer can be slightly modified to exploit certain hidden structures in the in-context sparse recovery tasks. Specifically, if the support of $\boldsymbol{\beta}$ lies in a subset $\mathbb{S} \subset [1:d]$ with $S < |\mathbb{S}| \leq d$, then by slightly modifying the parameters of the Transformer to ensure $[\boldsymbol{\beta}^{(k)}]_i = 0$ for all $i \notin \mathbb{S}$, the ICL performance can be improved by replacing all $d$ involved in Theorem 5.1 by $|\mathbb{S}|$. We defer the corresponding result and analysis to Appendix D.2.*

## 5.2 Label Prediction

In Section 5.1, we have demonstrated that Transformers can successfully recover the ground-truth sparse vector $\boldsymbol{\beta}^*$ with linear convergence by implementing a LISTA-type algorithm. In this section, we bridge the gap between this theoretical claim and the explicit objective of in-context sparse recovery, which is to predict $\widehat{y}_{N+1}$ given an in-context instance $(\mathbf{X}, \mathbf{y}, \mathbf{x}_{N+1})$. This gap might seem trivial at first glance, as given $\mathbf{x}_{N+1}$ and an accurate estimate of $\boldsymbol{\beta}$, the label $\widehat{y}_{N+1}$ can be obtained through a simple linear operation. However, we will show that, for a decoder-based Transformer, generating $\widehat{y}_{N+1}$ using the predicted sparse vector $\boldsymbol{\beta}$, which is implicitly embedded within the forward-pass sequences, critically depends on the structure of the read-out function.

**Theorem 5.2.** *Under the same setting as in Section 4.3, for any $n \in [N_0 + 1 : N + 1]$, with probability at least $1 - n\delta - \delta'$, we have*

$$\|y_n - \widehat{y}_n\| \leq b_{\mathbf{x}}\left(1 - \frac{2}{3}\gamma\right)^K + \frac{c_4 K}{\sqrt{n}}\left(1 - \frac{2}{3}\gamma\right)^{K-1}$$

*for a linear read-out function, and with probability at least $1 - \delta$, we have $\|y_n - \widehat{y}_n\| \leq c_5 e^{-\alpha_n K}$ for a query read-out function, where $c_4$, $c_5$ are constants.*

We defer the proof of Theorem 5.2 to Appendix E.

**Remark 5.** *Theorem 5.2 indicates that adopting a linear read-out function results in a prediction error of order $\mathcal{O}\big(e^{-K} + \frac{K}{\sqrt{n}}e^{-K}\big)$, which still exhibits linear convergence with respect to $K$. When a query-based read-out function is employed, the convergence rate improves to $\mathcal{O}(e^{-K})$. However, there exists a gap of order $\mathcal{O}\big(\frac{K}{\sqrt{n}}e^{-K}\big)$, which diminishes as $n$ becomes large. Empirically, we observe the superiority of using a query-based read-out function in our experimental results, as detailed in Section 6, and we also observe that the gap decreases as $n$ grows.*

## 6 Experimental Results

**Problem setup.** In all experiments, we adhere to the following steps to generate in-context sparse recovery instances. First, we sample a ground truth sparse vector $\boldsymbol{\beta}^*$ from a $d = 20$ dimensional standard normal distribution, and we fix the sparsity of $\boldsymbol{\beta}^*$ to be 3 by randomly setting 17 entries in $\boldsymbol{\beta}^*$ to zero. Next, we independently sample $N = 10$ vectors form a $d$ dimensional standard normal distribution and then contract the measurement matrix $\mathbf{X} \in \mathbb{R}^{10 \times 20}$ (each sampled $d$ dimensional random vector is a row in $\mathbf{X}$). We also sample an additional $\mathbf{x}_{N+1}$ from the $d$-dimensional standard Gaussian distribution. We follow the noiseless setting in Bai et al. (2024) for sparse recovery, i.e., $\mathbf{y} = \mathbf{X}\boldsymbol{\beta}^*$.

**Baselines.** The baselines for our experiments include traditional iterative algorithms such as ISTA and FISTA (Beck and Teboulle, 2009). We also evaluate three classical LISTA-type L2O algorithms: LISTA, LISTA-CP, ALISTA (Gregor and LeCun, 2010; Chen et al., 2018; Liu and Chen, 2019). For all of these algorithms, we set the number of iterations $K = 12$. We generate a single fixed measurement matrix $\mathbf{X}$. For each training epoch, we create $I = 50,000$ instances from $50,000$ randomly generated sparse vectors. During inference, we evaluate the pre-trained LISTA-type models under two settings: (1) when the measurement matrix remains identical to that used during pre-training, reported as "Fixed $X$", and (2) when the measurement matrix is varying through random sampling, reported as "Varying $X$".

We also evaluate the LISTA-VM algorithm introduced in Theorem 4.1, where we set the number of iterations $K = 12$ as well. For each training epoch, we randomly sample $100$ measurement matrices, each generating $500$ instances from $500$ randomly generated sparse vectors, which results in a total of $50,000$ instances. For comparison, we also meta-train LISTA and LISTA-CP using the same training method as LISTA-VM. We do not perform meta-training for ALISTA, as the training process of ALISTA involves solving a non-convex optimization problem for each different measurement matrix $\mathbf{X}$, which makes meta-training for ALISTA unrealistic. For all baseline algorithms, we minimize the sparse vector prediction loss $\sum_{i=1}^{I} \|\widehat{\boldsymbol{\beta}}_i - \boldsymbol{\beta}_i\|^2$ using gradient descent for each epoch. We run all baseline experiments for 340 epochs.

**Transformer structure.** We consider two Transformer models, i.e., a small Transformer model (denoted as Small TF) and GPT-2. Small TF has 12 layers, each containing a self-attention layer followed by an MLP layer. Each self-attention layer has 4 attention heads. We set the embedding dimension to $D = 42$, and the embedding structure according to Equation (4.2). For GPT-2, we employ 12 layers, and set the embedding dimension to $256$ and the number of attention heads per layer to $8$. We note that the Small TF model shares the same 4-head configuration and ReLU activation function as described in Theorem 4.1, while the configuration of GPT-2 is commonly used in practical applications. In order to train Small TF and GPT-2, we randomly generate 64 instances per epoch and train the algorithms for $10^6$ epochs. The training process minimizes the label prediction loss $\sum_{j=1}^{N+1}(y_j - \widehat{y}_j)^2$. We run the experiments for Small TF and GPT-2 on an NVIDIA RTX A5000 GPU with 24G memory. The training time for Small TF is approximately 8 hours, while the training time for GPT-2 is around 12 hours.

**Results.** We test the prediction performance of the baseline algorithms and Transformers on a sparse recovery instance $(\mathbf{X}, \mathbf{y}, \mathbf{x}_{N+1})$, and plot the label prediction loss in Figure 1.

We first do not impose any support constraint on $\boldsymbol{\beta}$. We start with the general setting where the testing instance is randomly generated (Varying $\mathbf{X}$). As shown in Figure 1a, GPT-2 outperforms

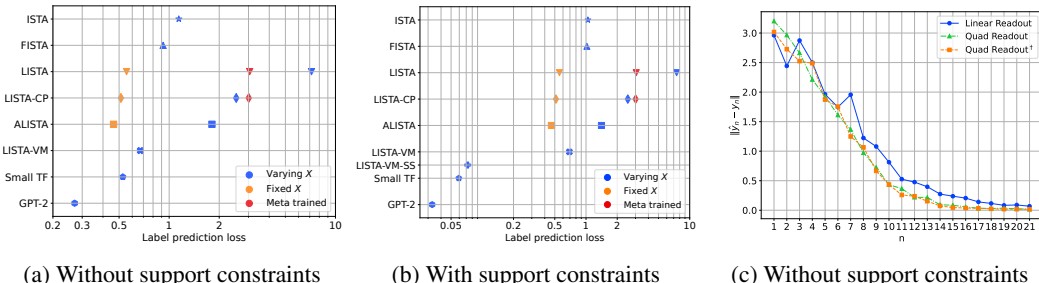

(a) Without support constraints  (b) With support constraints  (c) Without support constraints

Figure 1: Experimental results for sparse recovery. (a) $S = 3$. (b) $S = 3$, and the support is restricted to be within the first 10 entries. (c) Prediction with different read-outs functions.

Small TF, followed by LISTA-VM, which outperforms iterative algorithms FISTA and ISTA, while the classical LISTA-type algorithms LISTA, LISTA-CP, and ALISTA perform the worst. Such results highlight the efficiency of LISTA-VM, Small TF and GPT-2 in solving ICL sparse recovery problems, corroborating our theoretical result in Theorem 5.1. Meanwhile, classical LISTA-type algorithms cannot handle mismatches between the measurement matrices in pre-training and testing, leading to poor prediction performance. When $\mathbf{X}$ during testing is fixed to be the same as that in pre-training, all LISTA-type algorithms achieve performances comparable with Small TF. For meta-trained LISTA and LISTA-CP, the corresponding prediction loss (denoted by red marks in Figure 1a) are still much higher than that under LISTA-VM. This indicates that meta-training cannot help those classical LISTA-type algorithms achieve performances comparable with LISTA-VM, further corroborating the strong generalization capability induced by the constructed Transformer.

Next, we impose an additional constraint on the support of the sparse vectors. For in-context sparse recovery instances used in both pre-training and testing, we set the support of the sparse vector $\boldsymbol{\beta}$ to the first 10 entries, i.e., $\mathbb{S} = \{1, 2, \cdots, 10\}$. As observed in Figure 1b, Small TF and GPT-2 significantly improve their performances in Figure 1a, while other baseline algorithms do not exhibit significant performance improvements. In Figure 1b, we present the experimental results for a support-selected version of LISTA-VM, referred to as LISTA-VM-SS. This algorithm is a simple variation of LISTA-VM, where we incorporate prior knowledge of the support by setting all columns in $\mathbf{X}$ whose indices are not in the prior support set to be zero vectors. As we claim in Remark 4 and Corollary D.1, a Transformer could perform this LISTA-VM-SS by utilizing prior knowledge of the support. Our results show that the LISTA-VM-SS achieves an in-context prediction error of approximately $0.07$, which is almost $10\times$ better than the standard version of LISTA-VM and is comparable to the prediction error of Small TF. This empirical finding corroborates Remark 4.

Finally, we examine the label prediction loss under Small TF with three types of read-out functions, i.e., linear read-out function (*Linear Readout*), query read-out function (*Query Readout*), and another quadratic read-out function with parameters selected according to the proof of Theorem 5.2 (*Query Readout$^\dagger$*). In Figure 1c, we observe that the label prediction error is lower with the query read-out functions than with the linear read-out function. When $n$ becomes large, the gap between the linear read-out function and the other two types of query read-out functions becomes insignificant, which is consistent with our theoretical result in Theorem 5.2. Meanwhile, those two query read-out functions behave very similarly.

## 7 CONCLUSION

In this work, we demonstrated that Transformers' known ICL capabilities could be understood as performing L2O algorithms. Specifically, we showed that for in-context sparse recovery tasks, Transformers can execute the LISTA-VM algorithm with a provable linear convergence rate. Our results highlight that, unlike existing LISTA-type algorithms, which are limited to solving individual sparse recovery problems with fixed measurement matrices, Transformers can address a general class of sparse recovery problems with varying measurement matrices during inference without requiring parameter updates. Experimentally, we demonstrated that Transformers can leverage prior knowledge from training tasks and generalize effectively across different lengths of demonstration pairs, where traditional L2O methods typically fail.

ACKNOWLEDGMENTS

The work of R. Liu and J. Yang was supported in part by the U.S. National Science Foundation under the grants ECCS-2133170 and ECCS-2318759. The work of R. Zhou was supported in part by NSF grants 2139304, 2146838 and the Army Research Laboratory grant under Cooperative Agreement W911NF-17-2-0196. The work of C. Shen was supported in part by the U.S. National Science Foundation under the grants CNS-2002902, ECCS-2029978, ECCS-2143559, ECCS-2033671, CPS-2313110, and ECCS-2332060.

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

# Supplementary Materials

CONTENTS

## A  ADDITIONAL RELATED WORKS

**General L2O Techniques.**    L2O leverages machine learning to develop optimization algorithms, aiming to improve existing methods and innovate new ones. As highlighted by Sucker et al. (2024) and Chen et al. (2022), L2O intersects with meta-learning (also known as "learning-to-learn") and automated machine learning (AutoML).

Unlike meta-learning, which focuses on enabling models to quickly adapt to new tasks with minimal data by leveraging prior knowledge from diverse tasks (Finn et al., 2017; Hospedales et al., 2021), L2O aims to improve the optimization process itself by developing adaptive algorithms tailored to specific tasks, leading to faster convergence and enhanced performance in model training (Andrychowicz et al., 2016; Li and Malik, 2016). Thus, while meta-learning enhances task adaptability, L2O refines the efficiency of the optimization process. In contrast, AutoML focuses on model selection, optimization algorithm selection, and hyperparameter tuning (Yao et al., 2018); L2O distinguishes itself by its ability to generate new optimization techniques through learned models.

L2O has demonstrated significant potential across various optimization fields and applications. For instance, Andrychowicz et al. (2016) introduced a method where optimization algorithms are learned using recurrent neural networks trained to optimize specific classes of functions. Li and Malik (2016) proposed learning optimization algorithms through reinforcement learning, utilizing guided policy search to develop optimization strategies. Furthermore, Hruby et al. (2022) applied L2O to address the "minimal problem", a common challenge in computer vision characterized by the presence of many spurious solutions. They trained a multilayer perceptron model to predict initial problem solutions, significantly reducing computation time.

**Training Dynamics of Transformers.**    There exist some works aiming to theoretically understand the ICL mechanism in Transformer through their training dynamics. Ahn et al. (2024); Mahankali et al. (2023); Zhang et al. (2023); Huang et al. (2023) investigate the dynamics of Transformers with a single attention layer and a single head for in-context linear regression tasks. Cui et al. (2024) prove that Transformers with multi-head attention layers outperform those with single-head attention. Cheng et al. (2023) show that local optimal solutions in Transformers can perform gradient descent in-context for non-linear functions. Kim and Suzuki (2024) study the non-convex mean-field dynamics of Transformers, and Nichani et al. (2024) characterize the convergence rate for the training loss in learning a causal graph. Additionally, Chen et al. (2024a) investigate the gradient flow in training multi-head single-layer Transformers for multi-task linear regression. Chen and Li (2024) propose a supervised training algorithm for multi-head Transformers.

The training dynamics of Transformers for binary classification (Tarzanagh et al., 2023b;a; Vasudeva et al., 2024; Li et al., 2023; Deora et al., 2023; Li et al., 2024a) and next-token prediction (NTP) (Tian et al., 2023a;b; Li et al., 2024b; Huang et al., 2024) have also been studied recently.

# B  TABLE OF NOTATIONS

| Notation | Definition |
|---|---|
| $\mathbf{X}$ | Measurement Matrix |
| $\boldsymbol{\beta}^*$ | Ground-truth sparse vector |
| $\epsilon$ | Noise Vector |
| $Y$ | $Y = \mathbf{X}\boldsymbol{\beta}^* + \epsilon$ |
| $d$ | Number of columns of $\mathbf{X}$ |
| $N$ | Number of Measurement Vectors, i.e., number of rows of $\mathbf{X}$ |
| $S$ | Sparsity of $\boldsymbol{\beta}^*$, i.e., $\|\boldsymbol{\beta}^*\|_0 = S$ |
| $\mathbb{S}$ | Support set of $\boldsymbol{\beta}^*$ |
| $D$ | Dimension in the Self-attention Layer |
| $M$ | Number of Heads in the Self-attention Layer |
| $D'$ | Hidden Dimension in the MLP Layer |
| $K$ | Number of Layers in Transformer |
| $\mathbf{Q}_i^{(k)}$ | Query Matrix of the $k$-th Layer of Transformer's $i$-th Head |
| $\mathbf{K}_i^{(k)}$ | Key Matrix of the $k$-th Layer of Transformer's $i$-th Head |
| $\mathbf{V}_i^{(k)}$ | Value Matrix of the $k$-th Layer of Transformer's $i$-th Head |
| $\mathbf{H}^{(k)}$ | Input Sequence of the $k$-th Layer of Transformer |
| $\mathbf{h}_i^{(k)}$ | $[\mathbf{H}^{(k)}]_{:,i}$ |
| $\text{sign}(x)$ | Sign Function: $\text{sign}(x) = |x|/x$ if $x \neq 0$, $\text{sign}(x) = 0$ if $x = 0$ |
| $\mathcal{S}_\theta(x)$ | Soft Thresholding Function: $\mathcal{S}_\theta(x) = \text{sign}(x)\max\{0, |x| - \theta\}$ |
| $\sigma : \mathbb{R}^d \to \mathbb{R}^d$ | ReLU Function: $[\sigma(x)]_i = x_i$ if $x_i \geq 0$, $[\sigma(x)]_i = 0$ if $x_i < 0$ |
| $\neg\mathcal{E}$ | Complement of an event $\mathcal{E}$ |

# C  DEFERRED PROOFS IN SECTION 4.3

## C.1  TRANSFORMER STRUCTURE

**Attention layer.** Consider a model consisting of $K$ Transformer layers, where each layer is equipped with four attention heads. These heads are uniquely indexed as $+1$, $-1$, $+2$, and $-2$ to distinguish their specific roles within the layer.

$$
\mathbf{Q}_{\pm 1}^{(k)} = \begin{bmatrix} \mathbf{0}_{(d+1)\times(d+1)} & \mathbf{0}_{(d+1)\times d} & \mathbf{0}_{d+1} \\ \mathbf{0}_{d\times(d+1)} & \mathbf{M}_{\pm 1}^{Q,(k)} & \mathbf{0}_d \\ \mathbf{0}_{1\times(d+1)} & \mathbf{0}_{1\times d} & -B \end{bmatrix}, \quad \mathbf{Q}_{\pm 2}^{(k)} = \begin{bmatrix} \mathbf{0}_{d\times(2d+1)} & \mathbf{0}_d \\ \mathbf{0}_{1\times(2d+1)} & m_{\pm 2}^{Q,(k)} \\ \mathbf{0}_{(d+1)\times(2d+1)} & \mathbf{0}_{d+1} \end{bmatrix}
$$

$$
\mathbf{K}_{\pm 1}^{(k)} = \begin{bmatrix} \mathbf{0}_{(d+1)\times d} & \mathbf{0}_{(d+1)\times(d+1)} & \mathbf{0}_{d+1} \\ \mathbf{I}_{d\times d} & \mathbf{0}_{d\times(d+1)} & \mathbf{0}_d \\ \mathbf{0}_{1\times d} & \mathbf{0}_{1\times(d+1)} & 1 \end{bmatrix}, \quad \mathbf{K}_{\pm 2}^{(k)} = \begin{bmatrix} \mathbf{0}_{d\times d} & \mathbf{0}_d & \mathbf{0}_{d\times d+1} \\ \mathbf{0}_{1\times d} & 1 & \mathbf{0}_{1\times(d+1)} \\ \mathbf{0}_{(d+1)\times d} & \mathbf{0}_{d+1} & \mathbf{0}_{(d+1)\times(d+1)} \end{bmatrix}
$$

$$
\mathbf{V}_{\pm 1}^{(k)} = \begin{bmatrix} \mathbf{0}_{(d+1)\times d} & \mathbf{0}_{(d+1)\times(d+2)} \\ \mathbf{M}_{\pm 1}^{V,(k)} & \mathbf{0}_{d\times(d+2)} \\ \mathbf{0}_{1\times d} & \mathbf{0}_{1\times(d+2)} \end{bmatrix}, \qquad \mathbf{V}_{\pm 2}^{(k)} = \begin{bmatrix} \mathbf{0}_{(d+1)\times d} & \mathbf{0}_{(d+1)\times d+2} \\ \mathbf{M}_{\pm 2}^{V,(k)} & \mathbf{0}_{d\times(d+2)} \\ \mathbf{0}_{1\times d} & \mathbf{0}_{1\times(d+2)} \end{bmatrix}, \qquad \text{(C.1)}
$$

where $\mathbf{M}_{+1}^{Q,(k)}$, $\mathbf{M}_{-1}^{Q,(k)}$, $\mathbf{M}_{+1}^{V,(k)}$, $\mathbf{M}_{-1}^{V,(k)}$, $\mathbf{M}_{+2}^{V,(k)}$, and $\mathbf{M}_{-2}^{V,(k)}$ are all $d \times d$ matrices, and $m_{+2}^{Q,(k)}$, $m_{-2}^{Q,(k)}$ are scalars.

**MLP layer.** For the MLP layer following the $k$-th self-attention layer, we set

$$
\mathbf{W}_1 = \begin{bmatrix} \mathbf{W}_{1,\text{sub}} \\ -\mathbf{W}_{1,\text{sub}} \\ \mathbf{W}_{1,\text{sub}} \\ -\mathbf{W}_{1,\text{sub}} \end{bmatrix}, \mathbf{W}_2^\top = \begin{bmatrix} -\mathbf{I}_{(2d+2)\times(2d+2)} \\ \mathbf{I}_{(2d+2)\times(2d+2)} \\ \mathbf{I}_{(2d+2)\times(2d+2)} \\ -\mathbf{I}_{(2d+2)\times(2d+2)} \end{bmatrix}, \mathbf{b}^{(k)} = \begin{bmatrix} \mathbf{0}_{5d+5} \\ -\theta^{(k)} \cdot \mathbf{1}_d \\ \mathbf{0}_{d+2} \\ \theta^{(k)} \cdot \mathbf{1}_d \\ 0 \end{bmatrix}. \qquad \text{(C.2)}
$$

where the submatrix $\mathbf{W}_{1,\text{sub}}$ is defined as $\mathbf{W}_{1,\text{sub}} = \text{diag}(\mathbf{0}_{(d+1)\times(d+1)}, \mathbf{I}_{d\times d}, 0)$. Therefore, the output of the MLP layer is

$$\text{MLP}(\mathbf{h}_i) = \begin{bmatrix} [\mathbf{h}_i]_{1:d+1} \\ \mathcal{S}_{\theta^{(k)}}([\mathbf{h}_i]_{d+2:2d+1}) \\ [\mathbf{h}_i]_{2d+2} \end{bmatrix}. \tag{C.3}$$

where $\mathcal{S}_{\theta^{(k)}}$ is the soft-thresholding function parameterized by $\theta^{(k)}$.

## C.2 Proof of Theorem 4.1

We start by stating an equivalent form of Theorem 4.1 below, where we specific $\mathbf{M}^{(k)}$ in Theorem 4.1 to be $\gamma^{(k)}\mathbf{M}^V$.

**Theorem C.1** (Equivalent form Theorem 4.1). *Suppose Assumption 1 holds. For a Transformer with $K$ layers as described in Section 4.3, set the input sequence as:*

$$\mathbf{H}^{(1)} = \begin{bmatrix} [\mathbf{X}^\top]_{:,1} & [\mathbf{X}^\top]_{:,1} & \cdots & [\mathbf{X}^\top]_{:,N} & [\mathbf{X}^\top]_{:,N} & \mathbf{x}_{N+1} \\ 0 & y_1 & \cdots & 0 & y_N & 0 \\ \boldsymbol{\beta}_1^{(1)} & \boldsymbol{\beta}_2^{(1)} & \cdots & \boldsymbol{\beta}_{2N-1}^{(1)} & \boldsymbol{\beta}_{2N}^{(1)} & \boldsymbol{\beta}_{2N+1}^{(1)} \\ 1 & 0 & \cdots & 1 & 0 & 1 \end{bmatrix}. \tag{C.4}$$

*Denote $\mathbf{H}^{(k+1)}$ as the output of the $K$-th layer of the Transformer and define $\boldsymbol{\beta}_{2n+1}^{(K+1)} = \mathbf{H}_{d+2:2d+1,2n+1}^{(k+1)}$. There exists a set of parameters within the Transformer such that for all $k \in [1, K]$, we have:*

$$\boldsymbol{\beta}_{2n+1}^{(k+1)} = \mathcal{S}_{\theta^{(k)}}\left(\boldsymbol{\beta}_{2n+1}^{(k)} - \gamma^{(k)}(\mathbf{D}_n)^\top([\mathbf{X}]_{1:n,:}\boldsymbol{\beta}_{2n+1}^{(k)} - \mathbf{y}_{1:n})\right),$$

*where $\mathbf{D}_n = \frac{1}{2n+1}[\mathbf{X}]_{1:n,:}(\mathbf{M}^V)^\top$ and $\mathbf{M}^V \in \mathbb{R}^{d\times d}$ is embedded in the $k$-th Transformer lsyer.*

Proving the theorem is equivalent to demonstrating the existence of a Transformer for which the following proposition holds for any $k \geq 2$:

**Proposition 1.** *Suppose Assumption 1 holds. For a $K$ layers Transformer with structure described in Section 4.3, the input sequence of the $k$-th layer of the Transformer satisfies:*

$$\mathbf{H}^{(k)} = \begin{bmatrix} [\mathbf{X}^\top]_{:,1} & [\mathbf{X}^\top]_{:,1} & \cdots & [\mathbf{X}^\top]_{:,N} & [\mathbf{X}^\top]_{:,N} & \mathbf{x}_{N+1} \\ 0 & y_1 & \cdots & 0 & y_N & 0 \\ \boldsymbol{\beta}_1^{(k)} & \boldsymbol{\beta}_2^{(k)} & \cdots & \boldsymbol{\beta}_{2N-1}^{(k)} & \boldsymbol{\beta}_{2N}^{(k)} & \boldsymbol{\beta}_{2N+1}^{(k)} \\ 1 & 0 & \cdots & 1 & 0 & 1 \end{bmatrix},$$

*where it holds that $\boldsymbol{\beta}_{2n+1}^{(k)} = \mathcal{S}_{\theta^{(k-1)}}\left(\boldsymbol{\beta}_{2n+1}^{(k-1)} - \gamma^{(k-1)}(\mathbf{D}_n)^\top([\mathbf{X}]_{1:n,:}\boldsymbol{\beta}_{2n+1}^{(k)} - \mathbf{y}_{1:n})\right)$, $\mathbf{D}_n = \frac{1}{2n+1}[\mathbf{X}]_{1:n,:}(\mathbf{M}^V)^\top$ and $|\boldsymbol{\beta}_{2n+1}^{(k-1)}| \leq C_\beta^k$ for Constant $C_\beta$.*

*Proof of Proposition 1.* We prove Proposition 1 is true for all $k \geq 2$ by induction. First, $\mathbf{H}^{(k)}$ for $k = 2$ satisfies the condition automatically; therefore, Proposition 1 is true when $k = 2$. Then, we demonstrate that if Proposition 1 is true for $k - 1$, it remains valid for $k$. For odd values of $i$, the token-wise outputs of the $k$-th attention layer corresponding to the first and second heads satisfy:

$$\mathbf{Q}_{\pm1}^{(k)}\mathbf{h}_i^{(k)} = \begin{bmatrix} \mathbf{0}_{d+1} \\ \mathbf{M}_{\pm1}^{Q,(k)}\boldsymbol{\beta}_i^{(k)} \\ -B \end{bmatrix}; \quad \mathbf{K}_{\pm1}^{(k)}\mathbf{h}_i^{(k)} = \begin{bmatrix} \mathbf{0}_{d+1} \\ [\mathbf{X}^\top]_{:,\lfloor\frac{i+1}{2}\rfloor} \\ 1 \end{bmatrix}; \quad \mathbf{V}_{\pm1}^{(k)}\mathbf{h}_i^{(k)} = \begin{bmatrix} \mathbf{0}_{d+1} \\ \mathbf{M}_{\pm1}^{V,(k)}[\mathbf{X}^\top]_{:,\lfloor\frac{i+1}{2}\rfloor} \\ 0 \end{bmatrix}.$$

The token-wise outputs of the third and fourth heads for odd $i$ satisfy

$$\mathbf{Q}_{\pm2}^{(k)}\mathbf{h}_i^{(k)} = \begin{bmatrix} \mathbf{0}_d \\ m_{\pm2}^{Q,(k)} \\ \mathbf{0}_{d+1} \end{bmatrix}; \quad \mathbf{K}_{\pm2}^{(k)}\mathbf{h}_i^{(k)} = \mathbf{0}_{2d+2}; \quad \mathbf{V}_{\pm2}^{(k)}\mathbf{h}_i^{(k)} = \begin{bmatrix} \mathbf{0}_{d+1} \\ \mathbf{M}_{\pm2}^{V,(k)}[\mathbf{X}^\top]_{:,\lfloor\frac{i+1}{2}\rfloor} \\ 0 \end{bmatrix}.$$

Besides, for any $i$ that is an even number, the token-wise outputs of the $k$-th attention layer corresponding to the first and second heads satisfy:

$$\mathbf{Q}_{\pm 1}^{(k)}\mathbf{h}_i^{(k)} = \begin{bmatrix} \mathbf{0}_{d+1} \\ \mathbf{M}_{\pm 1}^{Q,(k)}\boldsymbol{\beta}_i^{(k)} \\ 0 \end{bmatrix}; \quad \mathbf{K}_{\pm 1}^{(k)}\mathbf{h}_i^{(k)} = \begin{bmatrix} \mathbf{0}_{d+1} \\ [\mathbf{X}^\top]_{:,\lfloor \frac{i+1}{2}\rfloor} \\ 0 \end{bmatrix}; \quad \mathbf{V}_{\pm 1}^{(k)}\mathbf{h}_i^{(k)} = \begin{bmatrix} \mathbf{0}_{d+1} \\ \mathbf{M}_{\pm 1}^{V,(k)}[\mathbf{X}^\top]_{:,\lfloor \frac{i+1}{2}\rfloor} \\ 0 \end{bmatrix}.$$

The token-wise outputs of the third and fourth heads for even $i$ satisfy

$$\mathbf{Q}_{\pm 2}^{(k)}\mathbf{h}_i^{(k)} = \mathbf{0}_{2d+2}; \quad \mathbf{K}_{\pm 2}^{(k)}\mathbf{h}_i^{(k)} = \begin{bmatrix} \mathbf{0}_d \\ y_{\frac{i}{2}} \\ \mathbf{0}_{d+1} \end{bmatrix}; \quad \mathbf{V}_{\pm 2}^{(k)}\mathbf{h}_i^{(k)} = \begin{bmatrix} \mathbf{0}_{d+1} \\ \mathbf{M}_{\pm 2}^{V,(k)}[\mathbf{X}^\top]_{:,\lfloor \frac{i+1}{2}\rfloor} \\ 0 \end{bmatrix}.$$

Therefore, for $\mathbf{h}_i$ with an odd index and head with index $u \in \{1, -1\}$, we have

$$\sigma\Big(\big\langle \mathbf{Q}_u^{(k)}\mathbf{h}_i^k, \mathbf{K}_u^{(k)}\mathbf{h}_j^k \big\rangle\Big) \cdot \mathbf{V}_u^{(k)}\mathbf{h}_j^{(k)}$$

$$= \sigma\Big((\boldsymbol{\beta}_i^{(k)})^\top (\mathbf{M}_u^Q)^\top [\mathbf{X}^\top]_{:,\lfloor \frac{i+1}{2}\rfloor} - \mathbb{1}_{\{j\%2=1\}}(j)B\Big) \cdot \begin{bmatrix} \mathbf{0}_{d+1} \\ \mathbf{M}_u^{V,(k)}[\mathbf{X}^\top]_{:,\lfloor \frac{i+1}{2}\rfloor} \\ 0 \end{bmatrix}. \quad \text{(C.5)}$$

Also, for $\mathbf{h}_i$ with an odd index and head with index $u \in \{2, -2\}$, we have

$$\sigma\Big(\big\langle \mathbf{Q}_u^{(k)}\mathbf{h}_i^k, \mathbf{K}_u^{(k)}\mathbf{h}_j^k \big\rangle\Big) \cdot \mathbf{V}_u^{(k)}\mathbf{h}_j^{(k)} = \mathbb{1}_{\{j\%2=0\}} \cdot \sigma\Big(m_u^{Q,(k)} \cdot y_{\frac{j}{2}}\Big) \cdot \begin{bmatrix} \mathbf{0}_{d+1} \\ \mathbf{M}_u^{V,(k)}[\mathbf{X}^\top]_{:,\lfloor \frac{i+1}{2}\rfloor} \\ 0 \end{bmatrix}.$$

$$\text{(C.6)}$$

We specify the parameters of the self-attention layer as

$$\mathbf{M}_1^{Q,(k)} = -\mathbf{I}_{d\times d}, \quad \mathbf{M}_1^{V,(k)} = \gamma^{(k)}\mathbf{M}^V,$$
$$\mathbf{M}_{-1}^{Q,(k)} = \mathbf{I}_{d\times d}, \quad \mathbf{M}_{-1}^{V,(k)} = -\gamma^{(k)}\mathbf{M}^V,$$
$$m_2^{q,(k)} = 1, \quad \mathbf{M}_2^{V,(k)} = \gamma^{(k)}\mathbf{M}^V,$$
$$m_{-2}^{q,(k)} = -1, \quad \mathbf{M}_{-2}^{V,(k)} = -\gamma^{(k)}\mathbf{M}^V.$$

Then, for head index $u \in \{1, -1\}$ we have

$$(\boldsymbol{\beta}_i^{(k)})^\top (\mathbf{M}_u^Q)^\top [\mathbf{X}^\top]_{:,\lfloor \frac{i+1}{2}\rfloor} - B \le \big|(\boldsymbol{\beta}_i^{(k)})^\top [\mathbf{X}^\top]_{:,\lfloor \frac{i+1}{2}\rfloor}\big| - B$$
$$\le \big\|\boldsymbol{\beta}_i^{(k)}\big\|_1 \big\|[\mathbf{X}^\top]_{:,\lfloor \frac{i+1}{2}\rfloor}\big\|_\infty - B \quad \text{(C.7)}$$
$$\le \big\|\boldsymbol{\beta}_i^{(k)} - \boldsymbol{\beta}^*\big\|_1 \big\|[\mathbf{X}^\top]_{:,\lfloor \frac{i+1}{2}\rfloor}\big\|_\infty + b_{\boldsymbol{\beta}}b_{\mathbf{x}} - B,$$

where Equation (C.7) is given by Hölder's Inequality. Therefore, combining this with the assumption $\|\mathbf{x}\|_\infty \le b_{\mathbf{x}}$ and $\|\boldsymbol{\beta}^*\|_1 \le b_{\boldsymbol{\beta}}$, we obtain

$$(\boldsymbol{\beta}_i^{(k)})^\top (\mathbf{M}_u^Q)^\top [\mathbf{X}^\top]_{:,\lfloor \frac{i+1}{2}\rfloor} - B \le b_{\mathbf{x}}\big\|\boldsymbol{\beta}_i^{(k)} - \boldsymbol{\beta}^*\big\|_1 + b_{\boldsymbol{\beta}}b_{\mathbf{x}} - B. \quad \text{(C.8)}$$

Recall that for odd $i$, we assume that Proposition 1 is true for $k-1$, then $\boldsymbol{\beta}_i^{(k)}$ satisfies

$$\boldsymbol{\beta}_i^{(k)} = \mathcal{S}_{\theta^{(k-1)}}\Big(\boldsymbol{\beta}_i^{(k-1)} - \gamma^{(k-1)}(\mathbf{D}_n)^\top([\mathbf{X}]_{1:\lfloor \frac{i+1}{2}\rfloor,:}\boldsymbol{\beta}_i^{(k)} - \mathbf{y}_{1:\lfloor \frac{i+1}{2}\rfloor})\Big)$$

and $|\boldsymbol{\beta}_{2n+1}^{(k-1)}| \le C_{\boldsymbol{\beta}}^k$. Then, we obtain

$$|\boldsymbol{\beta}_i^{(k)}|$$
$$\le |\boldsymbol{\beta}_i^{(k-1)}| + |\gamma^{(k-1)}|\|\mathbf{M}^V\|\Big\|\frac{1}{i}[\mathbf{X}]_{1:\lfloor \frac{i+1}{2}\rfloor,:}^\top[\mathbf{X}]_{1:\lfloor \frac{i+1}{2}\rfloor,:}\Big\| |\boldsymbol{\beta}_i^{(k-1)}| + b_{\boldsymbol{\beta}}|\gamma^{(k-1)}|\|\mathbf{M}^V\|\Big\|\frac{1}{i}[\mathbf{X}]_{1:\lfloor \frac{i+1}{2}\rfloor,:}\Big\|$$
$$\le C_{\boldsymbol{\beta}}^k + b_{\mathbf{x}}^2 d|\gamma^{(k-1)}|\|\mathbf{M}^V\|\frac{\lfloor \frac{i+1}{2}\rfloor}{i}C_{\boldsymbol{\beta}}^k + b_{\boldsymbol{\beta}}b_{\mathbf{x}}|\gamma^{(k-1)}|\|\mathbf{M}^V\|\frac{\lfloor \frac{i+1}{2}\rfloor}{i}$$
$$\le C_{\boldsymbol{\beta}}^k + b_{\mathbf{x}}^2 d|\gamma^{(k-1)}|\|\mathbf{M}^V\|C_{\boldsymbol{\beta}}^k + b_{\boldsymbol{\beta}}b_{\mathbf{x}}|\gamma^{(k-1)}|\|\mathbf{M}^V\|. \quad \text{(C.9)}$$

Equation (C.9) arises from the fact that $\lfloor \frac{i+1}{2} \rfloor / i \leq 1$. Denoting $\gamma_{\max} = \max_k \|\gamma^{(k)}\|$ and letting $C_{\boldsymbol{\beta}} = 1 + b_{\mathbf{x}}^2 d \gamma_{\max} \|\mathbf{M}^V\| + b_{\boldsymbol{\beta}} b_{\mathbf{x}} \gamma_{\max} \|\mathbf{M}^V\|$, we have

$$|\boldsymbol{\beta}_i^{(k)}| \leq C_{\boldsymbol{\beta}}^k + b_{\mathbf{x}}^2 d |\gamma^{(k-1)}| \|\mathbf{M}^V\| C_{\boldsymbol{\beta}}^k + b_{\boldsymbol{\beta}} b_{\mathbf{x}} |\gamma^{(k-1)}| \|\mathbf{M}^V\| \leq C_{\boldsymbol{\beta}}^{k+1}.$$

Therefore, we have

$$b_{\mathbf{x}} \|\boldsymbol{\beta}_i^{(k)} - \boldsymbol{\beta}^*\|_1 \leq b_{\mathbf{x}} \sqrt{d} (b_{\boldsymbol{\beta}} + C_{\boldsymbol{\beta}}^k).$$

Then, by setting $B \geq b_{\boldsymbol{\beta}} b_{\mathbf{x}} + b_{\mathbf{x}} \sqrt{d} (b_{\boldsymbol{\beta}} + C_{\boldsymbol{\beta}}^K)$, we obtain that $B > \left| (\boldsymbol{\beta}_i^{(k)})^\top [\mathbf{X}^\top]_{:,\lfloor \frac{i+1}{2} \rfloor} \right|$ for all $k \leq K$ and any odd $i$. Then, combining with Equation (C.8) we have

$$\sigma\left( (\boldsymbol{\beta}_i^{(k)})^\top (\mathbf{M}_u^Q)^\top [\mathbf{X}^\top]_{:,\lfloor \frac{i+1}{2} \rfloor} - B \right) = 0.$$

Further, from Equation (C.5), for $i$ is odd number and $j \in [1, 3, \cdots, i]$ we have

$$\sigma\left( \left\langle \mathbf{Q}_u^{(k)} \mathbf{h}_i^k, \mathbf{K}_u^{(k)} \mathbf{h}_j^k \right\rangle \right) \cdot \mathbf{V}_u^{(k)} \mathbf{h}_j^{(k)} = \mathbf{0}_{2d+2}.$$

Besides, for $i$ being an odd number and $j \in [2, 4, \cdots, i-1]$, we have

$$\sigma\left( \left\langle \mathbf{Q}_u^{(k)} \mathbf{h}_i^k, \mathbf{K}_u^{(k)} \mathbf{h}_j^k \right\rangle \right) \cdot \mathbf{V}_u^{(k)} \mathbf{h}_j^{(k)} = \sigma\left( (\boldsymbol{\beta}_i^{(k)})^\top (\mathbf{M}_u^Q)^\top [\mathbf{X}^\top]_{:,\lfloor \frac{i+1}{2} \rfloor} \right) \cdot \begin{bmatrix} \mathbf{0}_{d+1} \\ \mathbf{M}_u^{V,(k)} [\mathbf{X}^\top]_{:,\lfloor \frac{i+1}{2} \rfloor} \\ 2 \end{bmatrix}.$$

Thus, when $h_i$ is of odd index, summing over all $j \leq i$ and $u \in \{1, -1\}$, we obtain:

$$\sum_{u \in \{-1, +1\}} \sum_{j=1}^{i} \sigma\left( \left\langle \mathbf{Q}_u^{(k)} \mathbf{h}_i^k, \mathbf{K}_u^{(k)} \mathbf{h}_j^k \right\rangle \right) \cdot \mathbf{V}_u^{(k)} \mathbf{h}_j^{(k)}$$

$$= \sum_{u \in \{-1, +1\}} \sum_{j=2}^{i-1} \sigma\left( \left\langle \mathbf{Q}_u^{(k)} \mathbf{h}_i^k, \mathbf{K}_u^{(k)} \mathbf{h}_j^k \right\rangle \right) \cdot \mathbf{V}_u^{(k)} \mathbf{h}_j^{(k)}$$

$$= \sum_{j=2}^{i-1} -(\boldsymbol{\beta}_i^{(k)})^\top [\mathbf{X}^\top]_{:,\lfloor \frac{j+1}{2} \rfloor} \cdot \begin{bmatrix} \mathbf{0}_{d+1} \\ \gamma^{(k)} \mathbf{M}^V [\mathbf{X}^\top]_{:,\lfloor \frac{j+1}{2} \rfloor} \\ 0 \end{bmatrix}$$

$$= \begin{bmatrix} \mathbf{0}_{d+1} \\ -\gamma^{(k)} \mathbf{M}^V [\mathbf{X}^\top]_{:,1:\frac{i-1}{2}} [\mathbf{X}]_{1:\frac{i-1}{2},:} \boldsymbol{\beta}_i^{(k)} \\ 0 \end{bmatrix}. \tag{C.10}$$

Next, we consider heads with indexes $m \in \{2, -2\}$. From Equation (C.6) we obtain

$$\sum_{u \in \{-2, +2\}} \sum_{j=1}^{i} \sigma\left( \left\langle \mathbf{Q}_u^{(k)} \mathbf{h}_i^k, \mathbf{K}_u^{(k)} \mathbf{h}_j^k \right\rangle \right) \cdot \mathbf{V}_u^{(k)} \mathbf{h}_j^{(k)}$$

$$= \sum_{u \in \{-2, +2\}} \sum_{j=2}^{i-1} \sigma\left( \left\langle \mathbf{Q}_u^{(k)} \mathbf{h}_i^k, \mathbf{K}_u^{(k)} \mathbf{h}_j^k \right\rangle \right) \cdot \mathbf{V}_u^{(k)} \mathbf{h}_j^{(k)}$$

$$= \sum_{j=2}^{i-1} y_{\frac{j}{2}} \cdot \begin{bmatrix} \mathbf{0}_{d+1} \\ \gamma^{(k)} \mathbf{M}^V [\mathbf{X}^\top]_{:,\frac{j}{2}} \\ \mathbf{0}_2 \end{bmatrix}$$

$$= \begin{bmatrix} \mathbf{0}_{d+1} \\ \gamma^{(k)} \mathbf{M}^V [\mathbf{X}^\top]_{:,1:\frac{i-1}{2}} \mathbf{y}_{1:\frac{i-1}{2}} \\ \mathbf{0}_2 \end{bmatrix}. \tag{C.11}$$

Combining Equation (C.10) and Equation (C.11), we obtain the following equation for $\mathbf{h}_i$ with an odd index:

$$\mathbf{h}_i^{(k+1)} = \text{MLP}\left(\mathbf{h}_i^{(k)} + \frac{1}{i}\sum_{u\in\{\pm1,\pm2\}}\sum_{j=1}^i \sigma\left(\left\langle\mathbf{Q}_u^{(k)}\mathbf{h}_i^k, \mathbf{K}_u^{(k)}\mathbf{h}_j^k\right\rangle\right)\cdot\mathbf{V}_u^{(k)}\mathbf{h}_j^{(k)}\right)$$

$$= \begin{bmatrix} [\mathbf{X}^\top]_{:,\lfloor\frac{i+1}{2}\rfloor} \\ 0 \\ \boldsymbol{\beta}_i^{k+1} \\ 1 \end{bmatrix},$$

where

$$\boldsymbol{\beta}_i^{k+1} = \mathcal{S}_{\theta^{(k)}}\left(\boldsymbol{\beta}_i^{(k)} - \frac{\gamma^{(k)}}{i}\mathbf{M}^V[\mathbf{X}^\top]_{:,1:\frac{i-1}{2}}[\mathbf{X}]_{1:\frac{i-1}{2},:}\boldsymbol{\beta}_i^{(k)} + \gamma^{(k)}\mathbf{M}^V[\mathbf{X}^\top]_{:,1:\frac{i-1}{2}}\mathbf{y}_{1:\frac{i-1}{2}}\right)$$

$$= \mathcal{S}_{\theta^{(k)}}\left(\boldsymbol{\beta}_i^{(k)} - \frac{\gamma^{(k)}}{i}\mathbf{M}^V[\mathbf{X}^\top]_{:,1:\frac{i-1}{2}}\left([\mathbf{X}]_{1:\frac{i-1}{2},:}\boldsymbol{\beta}_i^{(k)} - \mathbf{y}_{1:\frac{i-1}{2}}\right)\right). \tag{C.12}$$

Similarly, we obtain the following equation for $\mathbf{h}_i$ with an even index:

$$\mathbf{h}_i^{(k+1)} = \text{MLP}\left(\mathbf{h}_i^{(k+1)} + \frac{1}{i}\sum_{u\in\{\pm1,\pm2\}}\sum_{j=1}^i \sigma\left(\left\langle\mathbf{Q}_u^{(k)}\mathbf{h}_i^k, \mathbf{K}_u^{(k)}\mathbf{h}_j^k\right\rangle\right)\cdot\mathbf{V}_u^{(k)}\mathbf{h}_j^{(k)}\right)$$

$$= \begin{bmatrix} [\mathbf{X}^\top]_{:,\lfloor\frac{i+1}{2}\rfloor} \\ 0 \\ \boldsymbol{\beta}_i^{k+1} \\ 0 \end{bmatrix}. \tag{C.13}$$

Note that an explicit formulation of $\boldsymbol{\beta}_i^{(k+1)}$ is not required when $i$ is even. Next, combining Equation (C.12) and Equation (C.13) gives

$$\mathbf{H}^{(k+1)} = \begin{bmatrix} [\mathbf{X}^\top]_{:,1} & [\mathbf{X}^\top]_{:,1} & \cdots & [\mathbf{X}^\top]_{:,N} & [\mathbf{X}^\top]_{:,N} & \mathbf{x}_{N+1} \\ 0 & y_1 & \cdots & 0 & y_N & 0 \\ \boldsymbol{\beta}_1^{(k+1)} & \boldsymbol{\beta}_2^{(k+1)} & \cdots & \boldsymbol{\beta}_{2N-1}^{(k+1)} & \boldsymbol{\beta}_{2N}^{(k+1)} & \boldsymbol{\beta}_{2N+1}^{(k+1)} \\ 1 & 0 & \cdots & 1 & 0 & 1 \end{bmatrix}.$$

The proof for Proposition 1 is thus complete. $\qquad\square$

## D DEFERRED PROOFS IN SECTION 5.1

### D.1 PROOF OF THEOREM 5.1

In Section 4.1, we assume that each row of the measurement matrix $\mathbf{X}$ is independently sampled from an isometric sub-Gaussian distribution with zero mean and covariance $\text{diag}(\sigma_1^2,\cdots,\sigma_d^2)$. Without loss of generality, let $\sigma_1 \geq \sigma_2 \geq \cdots \geq \sigma_d$. We start the proof of Theorem 5.1 by introducing the auxiliary lemmas: Lemma 1, Lemma 2, and Lemma 3.

**Lemma 1.** *Assume $\boldsymbol{\beta}^{(k)}$ and $\boldsymbol{\beta}^{(k+1)}$ satisfy:*

$$\boldsymbol{\beta}^{(k+1)} = \mathcal{S}_{\theta^{(k)}}\left(\boldsymbol{\beta}^{(k)} - \gamma\mathbf{D}^\top\left(\mathbf{X}\boldsymbol{\beta}^{(k)} - \mathbf{y}\right)\right). \tag{D.1}$$

*Define $\sigma_{\min} = \min_{i\in\mathbb{S}}\left|\mathbf{D}_{:,i}^\top\mathbf{X}_{:,i}\right|$, $\sigma_{\max} = \max_{i\neq j}\left|\mathbf{D}_{:,i}^\top\mathbf{X}_{:,j}\right|$ and let $\theta^{(k)} = \gamma\sigma_{\max}c_1 e^{-c_2(k-2)}$, where $c_1 = \|\boldsymbol{\beta}^*\|_1$ and $c_2 = \log\left(\frac{1}{\gamma(2S-1)\sigma_{\max}+|1-\gamma\sigma_{\min}|}\right)$. Then, if $\|\boldsymbol{\beta}^{(k)} - \boldsymbol{\beta}^*\|_1 \leq c_1 e^{-c_2(k-2)}$ and $\mathbf{D}$, $\gamma$, $\theta^{(k)}$ satisfy the following conditions:*

$$\gamma(2S-1)\sigma_{\max} + |1-\gamma\sigma_{\min}| \leq 1,$$
$$0 \leq \gamma\mathbf{D}_{:,i}^\top\mathbf{X}_{:,i} \leq 1 \quad \forall i \in [d], \tag{D.2}$$

*then $\boldsymbol{\beta}^{(k+1)}$ satisfies*

$$\|\boldsymbol{\beta}^{(k+1)} - \boldsymbol{\beta}^*\|_1 \leq c_1 e^{-c_2(k-1)}.$$

*Proof of Lemma 1.* Note that we are considering the noiseless model: $\mathbf{y} = \mathbf{X}\boldsymbol{\beta}^*$. We begin by rewriting Equation (D.1):

$$\boldsymbol{\beta}^{(k+1)} = \mathcal{S}_{\theta^{(k)}}\left(\boldsymbol{\beta}^{(k)} - \gamma^{(k)}\mathbf{D}^\top(\mathbf{X}\boldsymbol{\beta}^{(k)} - \mathbf{y})\right)$$

$$= \mathcal{S}_{\theta^{(k)}}\left(\boldsymbol{\beta}^{(k)} - \gamma^{(k)}\mathbf{D}^\top\mathbf{X}(\boldsymbol{\beta}^{(k)} - \boldsymbol{\beta}^*)\right).$$

Consider entries of $\boldsymbol{\beta}^{(K+1)}$ in the support of $\boldsymbol{\beta}^*$:

$$\boldsymbol{\beta}_{\mathbb{S}}^{(k+1)} = \mathcal{S}_{\theta^{(k)}}\left(\boldsymbol{\beta}_{\mathbb{S}}^{(k)} - \gamma^{(k)}(\mathbf{D}_{:,\mathbb{S}})^\top\mathbf{X}_{:,\mathbb{S}}(\boldsymbol{\beta}_{\mathbb{S}}^{(k)} - \boldsymbol{\beta}_{\mathbb{S}}^*)\right)$$

$$\in \boldsymbol{\beta}_{\mathbb{S}}^{(k)} - \gamma^{(k)}(\mathbf{D}_{:,\mathbb{S}})^\top\mathbf{X}_{:,\mathbb{S}}(\boldsymbol{\beta}_{\mathbb{S}}^{(k)} - \boldsymbol{\beta}_{\mathbb{S}}^*) - \theta^{(k)}\partial\ell_1(\boldsymbol{\beta}_{\mathbb{S}}^{(k+1)}),$$

where $\partial\ell_1(\boldsymbol{\beta})$ is the sub-gradient of $|\boldsymbol{\beta}|$:

$$[\partial\ell_1(\boldsymbol{\beta})]_i = \begin{cases} \{\text{sign}([\boldsymbol{\beta}]_i)\} & \text{if } [\boldsymbol{\beta}]_i \neq 0, \\ [-1, 1] & \text{if } [\boldsymbol{\beta}]_i = 0. \end{cases}$$

Then, for any $i \in \mathbb{S}$, we have

$$\boldsymbol{\beta}_i^{(k+1)} \in \boldsymbol{\beta}_i^{(k)} - \gamma^{(k)}(\mathbf{D}_{:,i})^\top\mathbf{X}_{:,\mathbb{S}}(\boldsymbol{\beta}_{\mathbb{S}}^{(k)} - \boldsymbol{\beta}_{\mathbb{S}}^*) - \theta^{(k)}\partial\ell_1(\boldsymbol{\beta}_{\mathbb{S}}^{(k+1)}).$$

Note that $\boldsymbol{\beta}_i^{(k)} - \gamma^{(k)}(\mathbf{D}_{:,i})^\top\mathbf{X}_{:,\mathbb{S}}(\boldsymbol{\beta}_{\mathbb{S}}^{(k)} - \boldsymbol{\beta}_{\mathbb{S}}^*)$ can be rewritten as

$$\boldsymbol{\beta}_i^{(k)} - \gamma^{(k)}(\mathbf{D}_{:,i})^\top\mathbf{X}_{:,\mathbb{S}}(\boldsymbol{\beta}_{\mathbb{S}}^{(k)} - \boldsymbol{\beta}_{\mathbb{S}}^*)$$

$$= \boldsymbol{\beta}_i^{(k)} - \gamma^{(k)}\sum_{j \in \mathbb{S}, j \neq i}(\mathbf{D}_{:,i})^\top\mathbf{X}_{:,j}(\boldsymbol{\beta}_j^{(k)} - \boldsymbol{\beta}_j^*) - \gamma^{(k)}(\mathbf{D}_{:,i})^\top\mathbf{X}_{:,i}(\boldsymbol{\beta}_i^{(k)} - \boldsymbol{\beta}_i^*)$$

$$= \boldsymbol{\beta}_i^* - \gamma^{(k)}\sum_{j \in \mathbb{S}, j \neq i}(\mathbf{D}_{:,i})^\top\mathbf{X}_{:,j}(\boldsymbol{\beta}_j^{(k)} - \boldsymbol{\beta}_j^*) + \left(1 - \gamma^{(k)}(\mathbf{D}_{:,i})^\top\mathbf{X}_{:,i}\right)(\boldsymbol{\beta}_i^{(k)} - \boldsymbol{\beta}_i^*).$$

Therefore,

$$\boldsymbol{\beta}_i^{(k+1)} - \boldsymbol{\beta}_i^* \in -\gamma^{(k)}\sum_{j \in \mathbb{S}, j \neq i}(\mathbf{D}_{:,i})^\top\mathbf{X}_{:,j}(\boldsymbol{\beta}_j^{(k)} - \boldsymbol{\beta}_j^*) + \left(1 - \gamma^{(k)}(\mathbf{D}_{:,i})^\top\mathbf{X}_{:,i}\right)(\boldsymbol{\beta}_i^{(k)} - \boldsymbol{\beta}_i^*) - \theta^{(k)}\partial\ell_1(\boldsymbol{\beta}_i^{(k+1)}).$$

Based on the definition of $\partial\ell_1$, we derive the following inequality:

$$|\boldsymbol{\beta}_i^{(k+1)} - \boldsymbol{\beta}_i^*| \leq \gamma^{(k)}\sum_{j \in \mathbb{S}, j \neq i}\left|(\mathbf{D}_{:,i})^\top\mathbf{X}_{:,j}\right||\boldsymbol{\beta}_j^{(k)} - \boldsymbol{\beta}_j^*| + \left|1 - \gamma^{(k)}(\mathbf{D}_{:,i})^\top\mathbf{X}_{:,i}\right||\boldsymbol{\beta}_i^{(k)} - \boldsymbol{\beta}_i^*| + \theta^{(k)}.$$

Recall that $0 < \gamma^{(k)}(\mathbf{D}_{:,i})^\top\mathbf{X}_{:,i} < 1$, we obtain that

$$|\boldsymbol{\beta}_i^{(k+1)} - \boldsymbol{\beta}_i^*| \leq \gamma^{(k)}\sigma_{\max}\sum_{j \in \mathbb{S}, j \neq i}|\boldsymbol{\beta}_j^{(k)} - \boldsymbol{\beta}_j^*| + \left|1 - \gamma^{(k)}\sigma_{\min}\right||\boldsymbol{\beta}_i^{(k)} - \boldsymbol{\beta}_i^*| + \theta^{(k)}.$$

From Lemma 2, we have support$(\boldsymbol{\beta}^{(k+1)}) \in \mathbb{S}$, thus, $\|\boldsymbol{\beta}^{(k+1)} - \boldsymbol{\beta}^*\|_1 = \|\boldsymbol{\beta}_{\mathbb{S}}^{(k+1)} - \boldsymbol{\beta}_{\mathbb{S}}^*\|_1$. Then,

$$\|\boldsymbol{\beta}^{(k+1)} - \boldsymbol{\beta}^*\|_1 \leq \sum_{i \in \mathbb{S}}\left(\gamma^{(k)}\sigma_{\max}\sum_{j \in \mathbb{S}, j \neq i}|\boldsymbol{\beta}_j^{(k)} - \boldsymbol{\beta}_j^*| + \left|1 - \gamma^{(k)}\sigma_{\min}\right||\boldsymbol{\beta}_i^{(k)} - \boldsymbol{\beta}_i^*| + \theta^{(k)}\right)$$

$$\leq \gamma^{(k)}(S-1)\sigma_{\max}\sum_{i \in \mathbb{S}}|\boldsymbol{\beta}_i^{(k)} - \boldsymbol{\beta}_i^*| + \left|1 - \gamma^{(k)}\sigma_{\min}\right|\left\|\boldsymbol{\beta}_i^{(k)} - \boldsymbol{\beta}_i^*\right\|_1 + S\theta^{(k)}$$

$$= \left(\gamma^{(k)}(S-1)\sigma_{\max} + \left|1 - \gamma^{(k)}\sigma_{\min}\right|\right)\left\|\boldsymbol{\beta}_i^{(k)} - \boldsymbol{\beta}_i^*\right\|_1 + S\theta^{(k)}$$

$$\leq \left(\gamma^{(k)}(2S-1)\sigma_{\max} + \left|1 - \gamma^{(k)}\sigma_{\min}\right|\right)c_1 e^{-c_2(k-1)}.$$

Combining with $c_2 = \log\left(\frac{1}{\gamma^{(k)}\sigma_{\max}(2S-1) + |1 - \gamma^{(k)}\sigma_{\min}|}\right)$, we obtain

$$\|\boldsymbol{\beta}^{(k+1)} - \boldsymbol{\beta}^*\|_1 \leq c_1 e^{-c_2 k}.$$

Therefore the proof is complete. $\qquad\square$

**Lemma 2.** *Suppose all conditions mentioned in Theorem 5.1 hold. For $k \in \mathbb{N}$, if*

$$\text{support}(\boldsymbol{\beta}^{(k)}) \in \mathbb{S} \quad and \quad \|\boldsymbol{\beta}_{\mathbb{S}}^{(k)} - \boldsymbol{\beta}_{\mathbb{S}}^*\| \leq c_1 e^{-c_2(k-1)},$$

*then it holds that*

$$\text{support}(\boldsymbol{\beta}^{(k+1)}) \in \mathbb{S}.$$

*Proof of Lemma 2.* For a fixed $k$, if $\text{support}(\boldsymbol{\beta}^{(k)}) \in \mathbb{S}$ and $\|\boldsymbol{\beta}_{\mathbb{S}}^{(k)} - \boldsymbol{\beta}_{\mathbb{S}}^*\| \leq c_1 e^{-c_2(k-1)}$, then we have

$$\boldsymbol{\beta}_{\mathbb{S}^C}^{(k+1)} = \mathcal{S}_{\theta^{(k)}}\left(\boldsymbol{\beta}_{\mathbb{S}^C}^{(k)} - \gamma^{(k)}(\mathbf{D}_{:,\mathbb{S}^C})^\top \mathbf{X}_{:,\mathbb{S}}(\boldsymbol{\beta}_{\mathbb{S}}^k - \boldsymbol{\beta}_{\mathbb{S}}^*)\right)$$

$$= \mathcal{S}_{\theta^{(k)}}\left(-\gamma^{(k)}(\mathbf{D}_{:,\mathbb{S}^C})^\top \mathbf{X}_{:,\mathbb{S}}(\boldsymbol{\beta}_{\mathbb{S}}^k - \boldsymbol{\beta}_{\mathbb{S}}^*)\right),$$

where $\mathbb{S}^C = [d']\backslash\mathbb{S}$. Then, for all $i \in \mathbb{S}^C$, we obtain

$$\boldsymbol{\beta}_i^{(k+1)} = \mathcal{S}_{\theta^{(k)}}\left(-\gamma^{(k)} \sum_{j\in\mathbb{S}}(\mathbf{D}_{:,i})^\top \mathbf{X}_{:,j}(\boldsymbol{\beta}_j^k - \boldsymbol{\beta}_j^*)\right).$$

Note that

$$\left|-\gamma^{(k)} \sum_{j\in\mathbb{S}}(\mathbf{D}_{:,i})^\top \mathbf{X}_{:,j}(\boldsymbol{\beta}_j^k - \boldsymbol{\beta}_j^*)\right| \leq \gamma^{(k)}\sigma_{\max}\|\boldsymbol{\beta}^{(k)} - \boldsymbol{\beta}^*\|_1$$

$$\leq \gamma^{(k)}\sigma_{\max}c_1 e^{-c_2(k-1)},$$

Given $\theta^{(k)}$ as defined in Theorem 1, we obtain:

$$\left|-\gamma^{(k)} \sum_{j\in\mathbb{S}}(\mathbf{D}_{:,i})^\top \mathbf{X}_{:,j}(\boldsymbol{\beta}_j^{(k)} - \boldsymbol{\beta}_j^*)\right| \leq \theta^{(k)}.$$

Consequently, it follows that $\boldsymbol{\beta}_i^{(k)} = 0$ for all $i \in \mathbb{S}^C$. $\qquad\qquad\square$

**Lemma 3.** *Suppose Assumption 1 holds and let $\mathbf{D} = \frac{1}{2N+1}\mathbf{X}(\mathbf{M}^V)^\top$. Then, there exists an $\mathbf{M}^V$ such that the following inequalities hold with probability at least $1 - 3\delta$:*

$$\min_{i\in\mathbb{S}}\left|(\mathbf{D}_{:,i})^\top \mathbf{X}_{:,i}\right| \geq \frac{N}{N+2}\left(1 - \sqrt{\frac{\log S - \log \delta}{Nc}}\right),$$

$$\max_{i\in\mathbb{S}}\left|(\mathbf{D}_{:,i})^\top \mathbf{X}_{:,i}\right| \leq \frac{N}{N+2}\left(1 + \sqrt{\frac{\log S - \log \delta}{Nc}}\right),$$

$$\max_{i\neq j}\left|(\mathbf{D}_{:,i})^\top \mathbf{X}_{:,j}\right| \leq \sqrt{\frac{\log d - \log \delta}{Nc}}.$$

*Proof of Lemma 3.* Recall that $\sigma_1 \geq \sigma_2 \geq \cdots \geq \sigma_d$. Let $\mathbf{M}^V = \frac{2}{\sigma_d^2}\mathbf{I}_{d\times d}$. To prove the lemma, we first introduce a lower bound for $\min_{i\in\mathbb{S}}\left|\mathbf{D}_{:,i}^\top\mathbf{X}_{:,i}\right|$. Note that

$$(\mathbf{D}_{:,i})^\top \mathbf{X}_{:,i} = \frac{1}{2N+1}\mathbf{M}^V(\mathbf{X}_{:,i})^\top \mathbf{X}_{:,i} = \frac{2}{2N+1}\cdot\frac{1}{\sigma_d^2}\sum_{j=1}^N \mathbf{X}_{j,i}^2 \geq \frac{2}{2N+1}\sum_{j=1}^N \frac{1}{\sigma_i^2}\mathbf{X}_{j,i}^2.$$

Note that $\mathbf{X}_{j,i}^2$ is a sub-exponential random variable. Then, from the tail bound for sub-exponential random variables, there exists a constant $c$ such that

$$\mathbb{P}\left\{\frac{1}{N}\sum_{j=1}^N(\frac{1}{\sigma_i}\mathbf{X}_{j,i}^2) \geq 1 - s\right\} \geq 1 - \exp\left(-Nc\min\{s^2, s\}\right).$$

Consider $0 < s \leq 1$, we have

$$\mathbb{P}\Big\{\frac{1}{N}\sum_{j=1}^{N}(\frac{1}{\sigma_i}\mathbf{X}_{j,i}^2) \geq 1 - s\Big\} \geq 1 - \exp\big(-Ncs^2\big),$$

it follows that

$$\mathbb{P}\Big\{\frac{1}{N}\min_{i\in\mathbb{S}}\frac{1}{\sigma_d^2}\sum_{j=1}^{N}\mathbf{X}_{j,i}^2 \geq 1 - s\Big\}$$

$$\geq \mathbb{P}\Big\{\frac{1}{N}\min_{i\in\mathbb{S}}\sum_{j=1}^{N}\frac{1}{\sigma_i^2}\mathbf{X}_{j,i}^2 \geq 1 - s\Big\}$$

$$= 1 - \mathbb{P}\Big\{\bigcup_{i\in\mathbb{S}}\Big\{\frac{1}{N}\sum_{j\in[1]}^{N}\frac{1}{\sigma_i^2}\mathbf{X}_{j,i}^2 < 1 - s\Big\}\Big\}$$

$$\geq 1 - \sum_{i\in\mathbb{S}}\mathbb{P}\Big\{\frac{1}{N}\sum_{j=1}^{N}\frac{1}{\sigma_i^2}\mathbf{X}_{j,i}^2 < 1 - s\Big\}$$

$$\geq 1 - S\exp\big(-Ncs^2\big).$$

Let $s = \sqrt{\frac{\log S - \log\delta}{Nc}}$. Then, with probability at least $1 - \delta$, we have

$$\frac{1}{N}\min_{i\in\mathbb{S}}\frac{1}{\sigma_d^2}\sum_{j=1}^{N}\mathbf{X}_{j,i}^2 \geq 1 - \sqrt{\frac{\log S - \log\delta}{Nc}}.$$

Therefore, with probability at least $1 - \delta$, it holds that

$$\min_{i\in\mathbb{S}}\big|(\mathbf{D}_{:,i})^\top\mathbf{X}_{:,i}\big| = \frac{2}{2N+1}\min_{i\in\mathbb{S}}\frac{1}{\sigma_d^2}\sum_{j=1}^{N}\mathbf{X}_{j,i}^2 \geq \frac{N}{N+2}\Big(1 - \sqrt{\frac{\log S - \log\delta}{Nc}}\Big).$$

Similarly, we have the following inequality holding with probability at least $1 - \delta$:

$$\max_{i\in\mathbb{S}}\big|(\mathbf{D}_{:,i})^\top\mathbf{X}_{:,i}\big| \leq \frac{N}{N+2}\Big(1 + \sqrt{\frac{\log S - \log\delta}{Nc}}\Big).$$

Next, we will provide a high probability upper bound for $\max_{i\neq j}\big|(\mathbf{D}_{:,i})^\top\mathbf{X}_{:,j}\big|$. Observe that when $i \neq j$, we have

$$(\mathbf{D}_{:,i})^\top\mathbf{X}_{:,j} = \frac{1}{2N+1}\mathbf{M}^Q(\mathbf{X}_{:,i})^\top\mathbf{X}_{:,j} = \frac{2}{2N+1}\sum_{k=1}^{N}\frac{1}{\sigma_d^2}\mathbf{X}_{k,i}\mathbf{X}_{k,j},$$

where $\mathbf{X}_{k,i}\mathbf{X}_{k,j}$ is a centered sub-exponential random variable. Then, from the tail bound for sub-exponential random variables, there exist constants $c$ and $0 < s \leq 1$ such that

$$\mathbb{P}\Big\{\Big|\frac{1}{N}\sum_{k=1}^{N}(\frac{1}{\sigma_d^2}\mathbf{X}_{k,i}\mathbf{X}_{k,j})\Big| \leq s\Big\} \geq 1 - \exp\big(-Ncs^2\big).$$

Then, we have

$$\mathbb{P}\left\{\max_{i\neq j}\left|\frac{2}{2+2N}\sum_{k=1}^{N}(\frac{1}{\sigma_d}\mathbf{X}_{k,i}\mathbf{X}_{k,j})\right|\leq s\right\}$$

$$\geq \mathbb{P}\left\{\max_{i\neq j}\left|\frac{1}{N}\sum_{k=1}^{N}(\frac{1}{\sigma_d}\mathbf{X}_{k,i}\mathbf{X}_{k,j})\right|\leq s\right\}$$

$$= 1-\mathbb{P}\left\{\bigcup_{i,j:i\neq j}\left\{\left|\frac{1}{N}\sum_{k=1}^{N}(\frac{1}{\sigma_d}\mathbf{X}_{k,i}\mathbf{X}_{k,j})\right|> s\right\}\right\}$$

$$\geq 1-\sum_{i,j:i\neq j}\mathbb{P}\left\{\left|\frac{1}{N}\sum_{k=1}^{N}(\frac{1}{\sigma_d}\mathbf{X}_{k,i}\mathbf{X}_{k,j})\right|> s\right\}$$

$$\geq 1-\sum_{i,j}\mathbb{P}\left\{\left|\frac{1}{N}\sum_{k=1}^{N}(\frac{1}{\sigma_d}\mathbf{X}_{k,i}\mathbf{X}_{k,j})\right|> s\right\}$$

$$\geq 1-d^2\exp\left(-Ncs^2\right).$$

Let $s=\sqrt{\frac{\log d-\log\delta}{Nc}}$, thus with probability at least $1-\delta$, we obtain

$$\max_{i\neq j}\left|(\mathbf{D}_{:,i})^\top\mathbf{X}_{:,i}\right|\leq\sqrt{\frac{\log d-\log\delta}{Nc}}.$$

Then, the proof is complete. $\qquad\square$

To prove Theorem 5.1, we first state an equivalent theorem here:

**Theorem D.1** (Equivalent form of Theorem 5.1). *Suppose Assumption 1 holds. Let $\delta\in(0,1)$. For a $K$-layer Transformer model with the structure described in Section 4.3, there exists a set of parameters in the Transformer such that for any $n\in[N_0,N]$, with probability at least $1-3\delta$, we have*

$$\|\boldsymbol{\beta}_{2n+1}^{(K+1)}-\boldsymbol{\beta}^*\|\leq c_1 e^{-\alpha_n K},$$

*where $\boldsymbol{\beta}_{2n+1}^{(K+1)}=[\mathbf{H}^{(K+1)}]_{d+1:2d+1,2n+1}$ and*

$$\alpha_n=-\log\left(1-\frac{2}{3}\gamma+\gamma(2S-1)\sqrt{\frac{\log d-\log\delta}{nc}}+\gamma\sqrt{\frac{\log S-\log\delta}{nc}}\right),\qquad(\text{D.3})$$

*$c_1=\|\boldsymbol{\beta}^*\|_1$ and $N_0=8(4S-2)^2\frac{\log d+\log S-\log\delta}{c}$. Here, $c$ is a positive constant and $\gamma$ can be any positive constant less than $1.6$.*

To prove Theorem D.1, first, for $\mathbf{D}_n=\frac{1}{2n+1}[\mathbf{X}]_{1:n,:}(\mathbf{M}^V)^\top$ defined in Equation (4.4), we define an event:

$$A=\left\{\min_{i\in\mathbb{S}}\left|([\mathbf{D}_n]_{:,i})^\top\mathbf{X}_{1:n,i}\right|\geq\frac{n}{n+2}\left(1-\sqrt{\frac{\log S-\log\delta}{nc}}\right),\right.$$

$$\max_{i\in\mathbb{S}}\left|([\mathbf{D}_n]_{:,i})^\top\mathbf{X}_{1:n,i}\right|\leq\frac{n}{n+2}\left(1+\sqrt{\frac{\log S-\log\delta}{nc}}\right),$$

$$\left.\max_{i\neq j}\left|([\mathbf{D}_n]_{:,i})^\top\mathbf{X}_{1:n,i}\right|\leq\sqrt{\frac{\log d-\log\delta}{nc}}\right\}.\qquad(\text{D.4})$$

Next, we introduce a proposition as following.

**Proposition 2.** *Suppose Assumption 1 and event $A$ defined in Equation (D.4) hold. For a $K$-layer Transformer model with the structure described in Section 4.3, and let the input sequence satisfy Equation (C.4). Then, there exists a set of parameters such that if $n\in[N_0,N]$, we have*

$$\|\boldsymbol{\beta}_{2n+1}^{(K+1)}-\boldsymbol{\beta}^*\|\leq c_1 e^{-\alpha_n K},$$

where $\boldsymbol{\beta}_{2n+1}^{(K+1)} = [\mathbf{H}^{(K+1)}]_{d+1:2d+1,2n+1}$, $c_1 = \|\boldsymbol{\beta}^*\|_1$, and $N_0 = 8(4S-2)^2 \frac{\log d + \log S - \log \delta}{c}$. $\alpha_n$ is given in Equation (D.3). Here, $c$ is a positive constant and $\gamma$ can be any positive constant less than $1.6$.

*Proof of Proposition 2.* We prove it by induction. First, for $k = 1$, we initialize $\boldsymbol{\beta}_{2n+1}^{(1)} = \mathbf{0}_d$, thus $\|\boldsymbol{\beta}_{2n+1}^{(1)} - \boldsymbol{\beta}^*\|_1 = \|\boldsymbol{\beta}^*\|_1$. Note that $c_1 = e^\alpha \|\boldsymbol{\beta}^*\|_1$, so Proposition 2 holds for $k = 1$.

Next, we assume Proposition 2 holds for $k$. From the proof of Theorem 5.1, the input sequence of the $(k+1)$-th Transformer layer is of the following structure:

$$\mathbf{H}^{(k+1)} = \begin{bmatrix} [\mathbf{X}^\top]_{:,1} & [\mathbf{X}^\top]_{:,1} & \cdots & [\mathbf{X}^\top]_{:,N} & [\mathbf{X}^\top]_{:,N} & \mathbf{x}_{N+1} \\ 0 & y_1 & \cdots & 0 & y_N & 0 \\ \boldsymbol{\beta}_1^{(k+1)} & \boldsymbol{\beta}_2^{(k+1)} & \cdots & \boldsymbol{\beta}_{2N-1}^{(k+1)} & \boldsymbol{\beta}_{2N}^{(k+1)} & \boldsymbol{\beta}_{2N+1}^{(k+1)} \\ 1 & 0 & \cdots & 1 & 0 & 1 \end{bmatrix},$$

where

$$\boldsymbol{\beta}_{2n+1}^{(k+1)} = \mathcal{S}_{\theta^{(k)}} \left( \boldsymbol{\beta}_{2n+1}^{(k)} - \gamma^{(k)} (\mathbf{D}_n)^\top ([\mathbf{X}]_{1:n,:} \boldsymbol{\beta}_{2n+1}^{(k)} - \mathbf{y}_{1:n}) \right),$$

From Lemma 1, let $\gamma^{(k)} = \gamma$, then for $c_1 = \|\boldsymbol{\beta}^*\|_1$, $c_2 = \log\left(\frac{1}{\gamma(2S-1)\sigma_{\max} + |1 - \gamma\sigma_{\min}|}\right)$, we have $\|\boldsymbol{\beta}_{2n+1}^{(k)} - \boldsymbol{\beta}^*\| \le c_1 e^{-c_2(k-1)}$ if the following conditions hold:

$$\gamma(2S-1)\sigma_{\max} + |1 - \gamma\sigma_{\min}| \le 1, \tag{D.5}$$

$$0 \le \gamma([\mathbf{D}_n]_{:,i})^\top \mathbf{X}_{1:n,i} \le 1, \quad \forall i \in [d]. \tag{D.6}$$

First, let

$$n \ge N_0 \quad \text{where} \quad N_0 = 8(4S-2)^2 \frac{\log d + \log S - \log \delta}{c}. \tag{D.7}$$

By rearranging Inequality D.7, we have

$$\sqrt{n} \ge 2\sqrt{2}(4S-2)\sqrt{\frac{\log d + \log S - \log \delta}{c}}$$

$$\ge \sqrt{2}(4S-2)\sqrt{\frac{\log d + \log S - 2\log \delta}{c}}$$

$$\ge (4S-2)\left(\sqrt{\frac{\log d - \log \delta}{c}} + \sqrt{\frac{\log S - \log \delta}{c}}\right)$$

$$\ge (4S-2)\sqrt{\frac{\log d - \log \delta}{c}} + \sqrt{\frac{\log S - \log \delta}{c}}. \tag{D.8}$$

Inequality D.8 is equivalent to

$$\frac{1}{\sqrt{n}}\left((2S-1)\sqrt{\frac{\log d - \log \delta}{c}} + \frac{1}{2}\sqrt{\frac{\log S - \log \delta}{c}}\right) \le \frac{1}{2}$$

$$\implies (2S-1)\sqrt{\frac{\log d - \log \delta}{nc}} \le \frac{1}{2}\left(1 - \sqrt{\frac{\log S - \log \delta}{nc}}\right)$$

$$\implies (2S-1)\sqrt{\frac{\log d - \log \delta}{nc}} \le \frac{n}{n+2}\left(1 - \sqrt{\frac{\log S - \log \delta}{nc}}\right). \tag{D.9}$$

Combining Inequality D.9 with Equation (D.4) we have

$$\sigma_{\max}(2S-1) \le (2S-1)\sqrt{\frac{\log d - \log \delta}{nc}}$$

$$\le \frac{n}{n+2}\left(1 - \sqrt{\frac{\log S - \log \delta}{nc}}\right)$$

$$\le \sigma_{\min}. \tag{D.10}$$

Next, recall that $\mathbf{M}^V = \frac{2}{\sigma_d^2}\mathbf{I}_{d\times d}$ and $([\mathbf{D}_n]_{:,i})^\top \mathbf{X}_{1:n,i} = \frac{1}{2n+1}(\mathbf{X}_{1:n,i})^\top \mathbf{M}^V \mathbf{X}_{1:n,i}$. Therefore, for any $i \in [d]$, we have $\gamma([\mathbf{D}_n]_{:,i})^\top \mathbf{X}_{1:n,i} \geq 0$ if $\gamma \geq 0$. Note that if $\gamma$ satisfies the following condition:

$$\gamma \leq \min\left\{\frac{1}{\sigma_{\min}}, \frac{1}{\max\{([\mathbf{D}_n]_{:,i})^\top \mathbf{X}_{1:n,i}\}}\right\}. \tag{D.11}$$

Then, Inequality D.6 will hold and $1 - \gamma\sigma_{\min} \geq 0$. Besides, we have

$$\gamma(2S-1)\sigma_{\max} + |1 - \gamma\sigma_{\min}| \leq \gamma\sigma_{\min} + |1 - \gamma\sigma_{\min}| \tag{D.12}$$
$$= \gamma\sigma_{\min} + 1 - \gamma\sigma_{\min} \tag{D.13}$$
$$= 1,$$

where Inequality D.12 is derived from Inequality D.10, and Equation (D.13) is obtained using Equation (D.11). Consequently, Inequalities D.5 and D.6 hold if event $A$ occurs. Furthermore, under the condition that event $A$ occurs, by setting

$$\gamma \leq \frac{1}{\frac{n}{n+2}\left(1 + \sqrt{\frac{\log S - \log \delta}{nc}}\right)}, \tag{D.14}$$

we have $\gamma$ satisfies Equation (D.11). Note that we assume $n \geq N_0$. Thus, we have

$$\frac{1}{\frac{n}{n+2}\left(1 + \sqrt{\frac{\log S - \log \delta}{nc}}\right)} \geq \frac{1}{1 + \sqrt{\frac{\log S - \log \delta}{N_0 c}}} \geq \frac{2}{1 + \sqrt{\frac{1}{8(4S-2)}}} \geq 1.6.$$

Therefore, any choice of $\gamma$ such that $\gamma \leq 1.6$ will ensure that Inequality (D.14) is satisfied. Next, noting that event $A$ holds and $\gamma$ satisfies Equation (D.11), we obtain

$$\|\boldsymbol{\beta}_{2n+1}^{(k)} - \boldsymbol{\beta}^*\|$$
$$\leq \|\boldsymbol{\beta}^*\|_1 \Big(\gamma(2S-1)\sigma_{\max} + |1 - \gamma\sigma_{\min}|\Big)^{k-1}$$
$$\leq \|\boldsymbol{\beta}^*\|_1 \left(\gamma(2S-1)\sqrt{\frac{\log d - \log \delta}{nc}} + 1 - \gamma\frac{n}{n+2}\left(1 - \sqrt{\frac{\log S - \log \delta}{nc}}\right)\right)^{k-1}$$
$$\leq \|\boldsymbol{\beta}^*\|_1 \left(\gamma(2S-1)\sqrt{\frac{\log d - \log \delta}{nc}} + 1 - \gamma\frac{n}{n+2} + \gamma\frac{n}{n+2}\sqrt{\frac{\log S - \log \delta}{nc}}\right)^{k-1}$$
$$\leq \|\boldsymbol{\beta}^*\|_1 \left(\gamma(2S-1)\sqrt{\frac{\log d - \log \delta}{nc}} + 1 - \frac{2}{3}\gamma + \gamma\sqrt{\frac{\log S - \log \delta}{nc}}\right)^{k-1}$$
$$= c_1 e^{-\alpha_n(k-1)}.$$

Therefore, the proof is complete. □

Note that Lemma 3 demonstrates that event $A$ occurs with a probability of at least $1 - 2\delta$. Therefore, the proof of Theorem D.1 is completed by combining Lemma 3 with Proposition 2.

## D.2 PROOF OF REMARK 4

Note that in Appendix D.2, we assume there is a constraint on the support of $\boldsymbol{\beta}^*$: its support set lies within a subset $\mathbb{S} \subset [1:d]$, with $S < |\mathbb{S}| \leq d$, where $S$ denotes the sparsity of $\boldsymbol{\beta}^*$. We establish the corollary by demonstrating the following corollary:

**Corollary D.1** (Equivalent statement of Remark 4.). *Suppose Assumption 1 holds and $S \leq |\mathbb{S}| \leq d$. For a $K$-layer Transformer model with the structure described in Section 4.3, and let the input sequence satisfy Equation (C.4). Then, there exists a set of parameters such that if $n \geq N_0^S$, with probability at least $1 - \delta$, we have*

$$\|\boldsymbol{\beta}_{2n+1}^{(K)} - \boldsymbol{\beta}^*\| \leq c_1 e^{-\alpha_n^S K},$$

where $\boldsymbol{\beta}_{2n+1}^{(K)} = [\mathbf{H}^{(K+1)}]_{d+1:2d+1,2N+1}$ and

$$\alpha_n^S = -\log\left(1 - \frac{2}{3}\gamma + \gamma(2S-1)\sqrt{\frac{\log|\mathbb{S}| - \log\delta}{nc}} + \gamma\sqrt{\frac{\log S - \log\delta}{nc}}\right), \tag{D.15}$$

$c_1 = \|\boldsymbol{\beta}^*\|_1$, and $N_0^p = 8(4S-2)^2 \frac{\log|\mathbb{S}| + \log S - \log\delta}{c}$. Here, $c$ is a positive constant, and $\gamma$ can be any positive constant less than $1.6$.

*Proof of Corollary D.1.* W.l.o.g., assume $\mathbb{S} = \{1, 2, \ldots, |\mathbb{S}|\}$. First, we modify the MLP layers based on Equation (C.2). We maintain the structure of $\mathbf{W}_2$ as described in Equation (C.2), and reconstruct $\mathbf{W}_1$ and $\mathbf{b}^{(k)}$. We define the following submatrices:

$$\mathbf{W}_{1,\text{sub}(1)} = \begin{bmatrix} \mathbf{0}_{(d+1)\times(d+1)} & \mathbf{0}_{(d+1)\times|\mathbb{S}|} & \mathbf{0}_{(d+1)\times(d-|\mathbb{S}|)} & \mathbf{0}_{d+1} \\ \mathbf{0}_{|\mathbb{S}|\times(d+1)} & \mathbf{I}_{|\mathbb{S}|\times|\mathbb{S}|} & \mathbf{0}_{|\mathbb{S}|\times(d-|\mathbb{S}|)} & \mathbf{0}_{|\mathbb{S}|} \\ \mathbf{0}_{(d-|\mathbb{S}|)\times(d+1)} & \mathbf{0}_{(d-|\mathbb{S}|)\times|\mathbb{S}|} & \mathbf{I}_{(d-|\mathbb{S}|)\times(d-|\mathbb{S}|)} & \mathbf{0}_{(d-|\mathbb{S}|)} \\ \mathbf{0}_{1\times(d+1)} & \mathbf{0}_{1\times|\mathbb{S}|} & \mathbf{0}_{1\times(d-|\mathbb{S}|)} & 0 \end{bmatrix},$$

$$\mathbf{W}_{1,\text{sub}(2)} = \begin{bmatrix} \mathbf{0}_{(d+1)\times(d+1)} & \mathbf{0}_{(d+1)\times|\mathbb{S}|} & \mathbf{0}_{(d+1)\times(d-|\mathbb{S}|)} & \mathbf{0}_{d+1} \\ \mathbf{0}_{|\mathbb{S}|\times(d+1)} & -\mathbf{I}_{|\mathbb{S}|\times|\mathbb{S}|} & \mathbf{0}_{|\mathbb{S}|\times(d-|\mathbb{S}|)} & \mathbf{0}_{|\mathbb{S}|} \\ \mathbf{0}_{(d-|\mathbb{S}|)\times(d+1)} & \mathbf{0}_{(d-|\mathbb{S}|)\times|\mathbb{S}|} & \mathbf{I}_{(d-|\mathbb{S}|)\times(d-|\mathbb{S}|)} & \mathbf{0}_{(d-|\mathbb{S}|)} \\ \mathbf{0}_{1\times(d+1)} & \mathbf{0}_{1\times|\mathbb{S}|} & \mathbf{0}_{1\times(d-|\mathbb{S}|)} & 0 \end{bmatrix},$$

$$\mathbf{W}_{1,\text{sub}(3)} = \begin{bmatrix} \mathbf{0}_{(d+1)\times(d+1)} & \mathbf{0}_{(d+1)\times|\mathbb{S}|} & \mathbf{0}_{(d+1)\times(d-|\mathbb{S}|)} & \mathbf{0}_{d+1} \\ \mathbf{0}_{|\mathbb{S}|\times(d+1)} & \mathbf{I}_{|\mathbb{S}|\times|\mathbb{S}|} & \mathbf{0}_{|\mathbb{S}|\times(d-|\mathbb{S}|)} & \mathbf{0}_{|\mathbb{S}|} \\ \mathbf{0}_{(d-|\mathbb{S}|)\times(d+1)} & \mathbf{0}_{(d-|\mathbb{S}|)\times|\mathbb{S}|} & -\mathbf{I}_{(d-|\mathbb{S}|)\times(d-|\mathbb{S}|)} & \mathbf{0}_{(d-|\mathbb{S}|)} \\ \mathbf{0}_{1\times(d+1)} & \mathbf{0}_{1\times|\mathbb{S}|} & \mathbf{0}_{1\times(d-|\mathbb{S}|)} & 0 \end{bmatrix},$$

$$\mathbf{W}_{1,\text{sub}(4)} = \begin{bmatrix} \mathbf{0}_{(d+1)\times(d+1)} & \mathbf{0}_{(d+1)\times|\mathbb{S}|} & \mathbf{0}_{(d+1)\times(d-|\mathbb{S}|)} & \mathbf{0}_{d+1} \\ \mathbf{0}_{|\mathbb{S}|\times(d+1)} & -\mathbf{I}_{|\mathbb{S}|\times|\mathbb{S}|} & \mathbf{0}_{|\mathbb{S}|\times(d-|\mathbb{S}|)} & \mathbf{0}_{|\mathbb{S}|} \\ \mathbf{0}_{(d-|\mathbb{S}|)\times(d+1)} & \mathbf{0}_{(d-|\mathbb{S}|)\times|\mathbb{S}|} & \mathbf{I}_{(d-|\mathbb{S}|)\times(d-|\mathbb{S}|)} & \mathbf{0}_{(d-|\mathbb{S}|)} \\ \mathbf{0}_{1\times(d+1)} & \mathbf{0}_{1\times|\mathbb{S}|} & \mathbf{0}_{1\times(d-|\mathbb{S}|)} & 0 \end{bmatrix}.$$

To simplify notations, let

$$\mathbf{W}_1 = \begin{bmatrix} \mathbf{W}_{1,\text{sub}(1)} \\ \mathbf{W}_{1,\text{sub}(2)} \\ \mathbf{W}_{1,\text{sub}(3)} \\ \mathbf{W}_{1,\text{sub}(4)} \end{bmatrix}, \mathbf{b}^{(k)} = \begin{bmatrix} \mathbf{0}_{5d+5} \\ -\theta^{(k)} \cdot \mathbf{1}_{|\mathbb{S}|} \\ \mathbf{0}_{2d+2-|\mathbb{S}|} \\ \theta^{(k)} \cdot \mathbf{1}_{|\mathbb{S}|} \\ \mathbf{0}_{1+d-|\mathbb{S}|} \end{bmatrix}.$$

The output of the modified MLP layer is

$$\text{MLP}(\mathbf{h}_i) = \begin{bmatrix} [\mathbf{h}_i]_{1:d+1} \\ \mathcal{S}_{\theta^{(k)}}([\mathbf{h}_i]_{d+2:d+|\mathbb{S}|+1}) \\ \mathbf{0}_{d-|\mathbb{S}|} \\ [\mathbf{h}_i]_{2d+2} \end{bmatrix}.$$

Next, we also modify the parameters in the self-attention layers. For convenience, we introduce a projection matrix $\mathbf{I}^S = \text{diag}(\mathbf{I}_{|\mathbb{S}|\times|\mathbb{S}|}, \mathbf{0}_{(d-|\mathbb{S}|)\times(d-|\mathbb{S}|)})$. The reconstructed self-attention layers are described below:

$$\mathbf{Q}_{\pm1}^{(k)} = \begin{bmatrix} \mathbf{0}_{(d+1)\times(d+1)} & \mathbf{0}_{(d+1)\times d} & \mathbf{0}_{d+1} \\ \mathbf{0}_{d\times(d+1)} & \mathbf{I}^S\mathbf{M}_{\pm1}^{Q,(k)} & \mathbf{0}_d \\ \mathbf{0}_{1\times(d+1)} & \mathbf{0}_{1\times d} & -B \end{bmatrix}, \quad \mathbf{Q}_{\pm2}^{(k)} = \begin{bmatrix} \mathbf{0}_{d\times(2d+1)} & \mathbf{0}_d \\ \mathbf{0}_{1\times(2d+1)} & m_{\pm2}^{Q,(k)} \\ \mathbf{0}_{(d+1)\times(2d+1)} & \mathbf{0}_{d+1} \end{bmatrix}$$

$$\mathbf{K}_{\pm1}^{(k)} = \begin{bmatrix} \mathbf{0}_{(d+1)\times d} & \mathbf{0}_{(d+1)\times(d+1)} & \mathbf{0}_{d+1} \\ \mathbf{I}^S & \mathbf{0}_{d\times(d+1)} & \mathbf{0}_d \\ \mathbf{0}_{1\times d} & \mathbf{0}_{1\times(d+1)} & 1 \end{bmatrix}, \quad \mathbf{K}_{\pm2}^{(k)} = \begin{bmatrix} \mathbf{0}_{d\times d} & \mathbf{0}_d & \mathbf{0}_{d\times d+1} \\ \mathbf{0}_{1\times d} & 1 & \mathbf{0}_{1\times(d+1)} \\ \mathbf{0}_{(d+1)\times d} & \mathbf{0}_{d+1} & \mathbf{0}_{(d+1)\times(d+1)} \end{bmatrix}$$

$$\mathbf{V}_{\pm1}^{(k)} = \begin{bmatrix} \mathbf{0}_{(d+1)\times d} & \mathbf{0}_{(d+1)\times(d+2)} \\ \mathbf{I}^S\mathbf{M}_{\pm1}^{V,(k)} & \mathbf{0}_{d\times(d+2)} \\ \mathbf{0}_{2\times d} & \mathbf{0}_{2\times(d+2)} \end{bmatrix}, \quad \mathbf{V}_{\pm2}^{(k)} = \begin{bmatrix} \mathbf{0}_{(d+1)\times d} & \mathbf{0}_{(d+1)\times d+2} \\ \mathbf{I}^S\mathbf{M}_{\pm2}^{V,(k)} & \mathbf{0}_{d\times(d+2)} \\ \mathbf{0}_{2\times d} & \mathbf{0}_{2\times(d+2)} \end{bmatrix}.$$

Therefore, when the input sequence is

$$\mathbf{H}^{(k)} = \begin{bmatrix} [\mathbf{X}^\top]_{:,1} & [\mathbf{X}^\top]_{:,1} & \cdots & [\mathbf{X}^\top]_{:,N} & [\mathbf{X}^\top]_{:,N} & \mathbf{x}_{N+1} \\ 0 & y_1 & \cdots & 0 & y_N & 0 \\ \boldsymbol{\beta}_1^{(k)} & \boldsymbol{\beta}_2^{(k)} & \cdots & \boldsymbol{\beta}_{2N-1}^{(k)} & \boldsymbol{\beta}_{2N}^{(k)} & \boldsymbol{\beta}_{2N+1}^{(k)} \\ 1 & 0 & \cdots & 1 & 0 & 1 \end{bmatrix},$$

and the corresponding output of the self-attention layer satisfies

$$\mathbf{H}^{(k+1)} = \mathrm{diag}(\mathbf{I}^S, 1, \mathbf{I}^S, 1)\mathbf{H}^{(k+1)}.$$

Then, it is equivalent to consider the input sequence as

$$\begin{bmatrix} [\mathbf{X}^\top]_{1:|\mathbb{S}|,1} & [\mathbf{X}^\top]_{1:|\mathbb{S}|,2} & \cdots & [\mathbf{X}^\top]_{1:|\mathbb{S}|,N} & [\mathbf{X}^\top]_{1:|\mathbb{S}|,N} & [\mathbf{x}_{N+1}]_{1:|\mathbb{S}|} \\ 0 & y_1 & \cdots & 0 & y_N & 0 \\ [\boldsymbol{\beta}_1^{(k)}]_{1:|\mathbb{S}|} & [\boldsymbol{\beta}_2^{(k)}]_{1:|\mathbb{S}|} & \cdots & [\boldsymbol{\beta}_{2N-1}^{(k)}]_{1:|\mathbb{S}|} & [\boldsymbol{\beta}_{2N}^{(k)}]_{1:|\mathbb{S}|} & [\boldsymbol{\beta}_{2N+1}^{(k)}]_{1:|\mathbb{S}|} \\ 1 & 0 & \cdots & 1 & 0 & 1 \end{bmatrix}.$$

Consequently, by applying Theorem 5.1 to this sub-problem, we derive the corresponding corollary. □

# E  DEFERRED PROOFS IN SECTION 5.2

## E.1  PROOF OF THEOREM 5.2

Note that in Theorem 5.2, we have two results: first, the label prediction loss for the linear read-out function, and second, the label prediction loss for the quadratic read-out function.

We start by introducing the following lemma, which provides an upper bound on $\|\boldsymbol{\beta}_j^{(k)}\|$ for every $j$ and $k \geq 2$.

**Lemma 4.** *If $\boldsymbol{\beta}^{(k)}$ satisfies Equation (C.12) and $\|\boldsymbol{\beta}^{(k)}\| \leq C_{\mathbf{x}}$, then we have $\|\boldsymbol{\beta}^{(k+1)}\| \leq C_{\mathbf{x}}$, where $C_{\mathbf{x}} = \frac{db_{\mathbf{x}}b_{\boldsymbol{\beta}}(N+1)}{\sigma_{\min}(\mathbf{X}^\top\mathbf{X})}$.*

*Proof of Lemma 4.* From Equation (C.12), we derive that

$$\boldsymbol{\beta}_i^{k+1} = \mathcal{S}_{\theta^{(k)}}\left(\boldsymbol{\beta}_i^{(k)} - \frac{\gamma^{(k)}}{i}\mathbf{M}^V[\mathbf{X}^\top]_{:,1:\frac{i-1}{2}}\left([\mathbf{X}]_{1:\frac{i-1}{2},:}\boldsymbol{\beta}_i^{(k)} - Y_{1:\frac{i-1}{2}}\right)\right)$$

$$= \mathcal{S}_{\theta^{(k)}}\left(\left(\mathbf{I} - \frac{\gamma^{(k)}}{i}\mathbf{M}^V[\mathbf{X}^\top]_{:,1:\frac{i-1}{2}}[\mathbf{X}]_{1:\frac{i-1}{2},:}\right)\boldsymbol{\beta}_i^{(k)} - \frac{\gamma^{(k)}}{i}\mathbf{M}^V[\mathbf{X}^\top]_{:,1:\frac{i-1}{2}}Y_{1:\frac{i-1}{2}}\right).$$

Since $\|\mathcal{S}_{\theta^{(k)}}(x)\| \leq \|x\|$, it follows that

$$\|\boldsymbol{\beta}_i^{k+1}\| \leq \left\|\left(\mathbf{I} - \frac{\gamma^{(k)}}{i}\mathbf{M}^V[\mathbf{X}^\top]_{:,1:\frac{i-1}{2}}[\mathbf{X}]_{1:\frac{i-1}{2},:}\right)\boldsymbol{\beta}_i^{(k)} - \frac{\gamma^{(k)}}{i}\mathbf{M}^V[\mathbf{X}^\top]_{:,1:\frac{i-1}{2}}Y_{1:\frac{i-1}{2}}\right\|$$

$$\overset{\leq}{\scriptstyle(i)} \left(1 - \frac{\gamma^{(k)}}{i \cdot \sigma_d^2}\sigma_{\min}\left([\mathbf{X}^\top]_{:,1:\frac{i-1}{2}}[\mathbf{X}]_{1:\frac{i-1}{2},:}\right)\right)\|\boldsymbol{\beta}_i^{(k)}\| + \frac{\gamma^{(k)}}{i \cdot \sigma_d^2}\|[\mathbf{X}^\top]_{:,1:\frac{i-1}{2}}Y_{1:\frac{i-1}{2}}\|$$

$$\overset{\leq}{\scriptstyle(ii)} \left(1 - \frac{\gamma^{(k)}}{(N+1) \cdot \sigma_d^2}\sigma_{\min}\left(\mathbf{X}^\top\mathbf{X}\right)\right)\|\boldsymbol{\beta}_i^{(k)}\| + \frac{\gamma^{(k)}}{\sigma_d^2}db_{\mathbf{x}}b_{\boldsymbol{\beta}}.$$

The satisfaction of Inequality (i) is achieved by setting $\mathbf{M}^V = \frac{2}{\sigma_d^2}\mathbf{I}_{d \times d}$, while Inequality (ii) is verified through the application of the Cauchy interlacing theorem. Consequently, by integrating the expression $C_{\mathbf{x}} = \frac{db_{\mathbf{x}}b_{\boldsymbol{\beta}}(N+1)}{\sigma_{\min}(\mathbf{X}^\top\mathbf{X})}$ with the aforementioned inequality, we obtain

$$\|\boldsymbol{\beta}_i^{k+1}\| \leq \left(1 - \frac{\gamma^{(k)}}{(N+1) \cdot \sigma_d^2}\sigma_{\min}\left(\mathbf{X}^\top\mathbf{X}\right)\right)\frac{db_{\mathbf{x}}b_{\boldsymbol{\beta}}(N+1)}{\sigma_{\min}\left(\mathbf{X}^\top\mathbf{X}\right)} + \frac{\gamma^{(k)}}{\sigma_d^2}db_{\mathbf{x}}b_{\boldsymbol{\beta}} \leq C_{\mathbf{x}},$$

Thus, the proof is complete. □

Next, we prove the following theorem to demonstrate the prediction loss for the linear read-out function.

**Theorem E.1.** *Suppose Assumption 1 holds. Consider a Transformer with $K + 1$ layers, where the first $K$ layers have the structure described in Section 4.3, and the input sequence is as defined in Equation* (C.4). *Then, if $n \geq N_0$, there exists a set of parameters in the Transformer such that the following inequality holds with probability at least $1 - \delta' - n\delta$:*

$$\|y_n - \widehat{y}_n\| \leq b_{\mathbf{x}}\left(1 - \frac{2}{3}\gamma\right)^K + \frac{c_4 K}{\sqrt{n}}\left(1 - \frac{2}{3}\gamma\right)^{K-1},$$

*where*

$$N_0 = 8(4S - 2)^2 \frac{\log d + \log S - \log \delta}{c},$$

$$c_4 = \frac{2\sqrt{2}b_{\mathbf{x}}\eta}{\sqrt{p}}\sqrt{\frac{\log d + \log S - \log \delta}{c}} + \frac{2B'}{\sqrt{p}} + \frac{2C_{\mathbf{x}}b_{\mathbf{x}}}{\sqrt{p}},$$

*where $c$ is a positive constant, $\eta$ is defined in Equation* (E.7), *$\delta'$ $B'$, $p$ are constants defined in Appendix E.1.*

We are now prepared to demonstrate Theorem E.1.

*Proof.* For this layer, we assign one attention head to follow the structure described below. Consider the case where $i = 2n - 1$ and $n \gg N_0$, and denote $[\mathbf{X}^\top]_{:,n}$ as $\mathbf{x}_n$. For the first attention head, its structure is as follows:

$$\mathbf{Q}_1^{\text{last}}\mathbf{h}_i^{(K+1)} = \begin{bmatrix} \mathbf{0}_d \\ 1 \\ \mathbf{x}_n \\ 0 \end{bmatrix}, \quad \mathbf{K}_1^{\text{last}}\mathbf{h}_i^{(K+1)} = \begin{bmatrix} \mathbf{0}_d \\ -B' \\ \boldsymbol{\beta}_i^{(K+1)} \\ 0 \end{bmatrix}, \quad \mathbf{V}_1^{\text{last}}\mathbf{h}_i^{(K+1)} = \begin{bmatrix} \mathbf{0}_d \\ \mathbb{1}_{\{i\%2=1\}} \\ \mathbf{0}_{d+1} \end{bmatrix}.$$

Therefore, we obtain

$$(\mathbf{Q}_1^{\text{last}}\mathbf{h}_i^{(K+1)})^\top (\mathbf{K}_1^{\text{last}}\mathbf{h}_j^{(K+1)}) = \mathbf{x}_n^\top \boldsymbol{\beta}_j^{(K+1)} - B'.$$

Let $p = \mathbb{P}\{\mathbf{x}_n^\top \boldsymbol{\beta}_j^{(K+1)} \geq B'\}$, which is the probability that the inner product $\mathbf{x}_n^\top \boldsymbol{\beta}_j^{(K+1)}$ exceeds the threshold $B'$. This probability is the same for any $n$ due to the symmetry of the distribution. We introduce a random variable $o_j \in \{0, 1\}$ to represent the following: we set $o_j = 0$ if $\mathbf{x}_n^\top \boldsymbol{\beta}_j^{(K+1)} < B'$ and $o_j = 1$ if $\mathbf{x}_n^\top \boldsymbol{\beta}_j^{(K+1)} \geq B'$. We define the following events:

$$\mathcal{E}_1 = \left\{ \sum_{j \leq N_0} o_j = 0 \right\}, \qquad \mathcal{E}_2(\epsilon) = \left\{ pn - \frac{p}{2} - \epsilon \leq \sum_{j \leq n} o_j \leq pn - \frac{p}{2} + \epsilon \right\}.$$

Assume $\mathcal{E}_1 \cap \mathcal{E}_2(\epsilon)$ holds. Then, we obtain

$$\frac{1}{i} \sum_{j=1}^{i} \left\langle \mathbf{Q}_1^{\text{last}}\mathbf{h}_i^{(K+1)}, \mathbf{K}_1^{\text{last}}\mathbf{h}_j^{(K+1)} \right\rangle \cdot \mathbf{V}_1^{\text{last}}\mathbf{h}_j^{(K+1)}$$

$$= \frac{1}{i} \sum_{j=1}^{i} \sigma\left(\mathbf{x}_n^\top \boldsymbol{\beta}_j^{(K+1)} - B'\right) \cdot \begin{bmatrix} \mathbf{0}_d \\ \mathbb{1}_{\{i\%2=1\}} \\ \mathbf{0}_{d+1} \end{bmatrix}$$

$$= \frac{1}{2n - 1} \sum_{j:o_j=1} \left(\mathbf{x}_n^\top \boldsymbol{\beta}_j^{(K+1)} - B'\right) \cdot \begin{bmatrix} \mathbf{0}_d \\ \mathbb{1}_{\{i\%2=1\}} \\ \mathbf{0}_{d+1.} \end{bmatrix}.$$

Let the last MLP layer satisfies

$$\text{MLP} \begin{bmatrix} \mathbf{0}_d \\ \widehat{y} \\ \mathbf{0}_{d+1.} \end{bmatrix} = \begin{bmatrix} \mathbf{0}_d \\ \frac{2}{p}\widehat{y} + B' \\ \mathbf{0}_{d+1,} \end{bmatrix},$$

it follows that

$$\text{MLP}\left(\frac{1}{i}\sum_{j=1}^{i}\left\langle \mathbf{Q}_1^{\text{last}}\mathbf{h}_i^{(K+1)}, \mathbf{K}_1^{\text{last}}\mathbf{h}_j^{(K+1)}\right\rangle \cdot \mathbf{V}_1^{\text{last}}\mathbf{h}_j^{(K+1)}\right)$$

$$=\begin{bmatrix} \mathbf{0}_d \\ \frac{2}{2np-p}\sum_{j:o_j=1}\mathbf{x}_n^\top\boldsymbol{\beta}_j^{(K+1)} - \frac{2(\sum o_j)}{2np-p}B' + B' \\ \mathbf{0}_{d+1}. \end{bmatrix}.$$

Denote $\widehat{y}_n = \frac{2}{2np-p}\sum_{j:o_j=1}\mathbf{x}_n^\top\boldsymbol{\beta}_j^{(K+1)} - \frac{2(\sum o_j)}{2np-p}B' + B'$, therefore

$$|y_n - \widehat{y}_n| = \left|\mathbf{x}_n^\top\boldsymbol{\beta}^* - \frac{2}{2np-p}\sum_{j:o_j=1}\mathbf{x}_n^\top\boldsymbol{\beta}_j^{(K+1)} + \frac{2(\sum o_j)}{2np-p}B' - B'\right|$$

$$\leq \|\mathbf{x}_n\|\left\|\boldsymbol{\beta}^* - \frac{2}{2np-p}\sum_{j:o_j=1}\mathbf{x}_n^\top\boldsymbol{\beta}_j^{(K+1)}\right\| + \left|\frac{2(\sum o_j)}{2np-p}B' - B'\right|$$

$$\leq b_\mathbf{x}\left\|\boldsymbol{\beta}^* - \frac{2}{2np-p}\sum_{j:o_j=1}\boldsymbol{\beta}_j^{(K+1)}\right\| + \frac{2\epsilon}{2np-p}|B'|. \tag{E.1}$$

We randomly select a set $\{o'_j : j \geq 2N_0 - 1\}$ such that $|\{o'_j\}| \leq \epsilon$, satisfying the following conditions:

- If $|\{o_j : o_j = 1\}| \geq pn - \dfrac{r}{2}$, then

$$\left|\{o_j : o_j = 1\} \setminus \{o'_j\}\right| = pn - \frac{r}{2};$$

- If $|\{o_j : o_j = 1\}| < pn - \dfrac{r}{2}$, then

$$\left|\{o_j : o_j = 1\} \cup \{o'_j\}\right| = pn - \frac{r}{2}.$$

We define the set $\mathcal{O}$ as follows:

$$\mathcal{O} = \begin{cases} \{o_j : o_j = 1\} \setminus \{o'_j\}, & \text{if } |\{o_j : o_j = 1\}| \geq pn - \dfrac{r}{2}, \\[2mm] \{o_j : o_j = 1\} \cup \{o'_j\}, & \text{if } |\{o_j : o_j = 1\}| < pn - \dfrac{r}{2}. \end{cases}$$

Therefore, we have

$$\left\|\boldsymbol{\beta}^* - \frac{2}{2np-p}\sum_{j:o_j=1}\mathbf{x}_n^\top\boldsymbol{\beta}_j^{(K+1)}\right\|$$

$$\leq \left\|\boldsymbol{\beta}^* - \frac{1}{np-\frac{p}{2}}\sum_{j:o_j\in\mathcal{O}}\boldsymbol{\beta}_j^{(K+1)}\right\| + \left\|\frac{1}{np-\frac{p}{2}}\sum_{j:o_j\in\{o'_j\}}\mathbf{x}_n^\top\boldsymbol{\beta}_j^{(K+1)}\right\|$$

$$\leq \left\|\boldsymbol{\beta}^* - \frac{1}{np-\frac{p}{2}}\sum_{j:o_j\in\mathcal{O}}\boldsymbol{\beta}_j^{(K+1)}\right\| + \frac{C_\mathbf{x}b_\mathbf{x}\epsilon}{np-\frac{p}{2}}, \tag{E.2}$$

where Equation (E.2) follows from Lemma 4. Observe that

$$\left\|\boldsymbol{\beta}^* - \frac{1}{np-\frac{p}{2}}\sum_{j:o_j\in\mathcal{O}}\boldsymbol{\beta}_j^{(K+1)}\right\| \leq \frac{1}{np-\frac{p}{2}}\sum_{j:o_j\in\mathcal{O}}\left\|\boldsymbol{\beta}^* - \boldsymbol{\beta}_j^{(K+1)}\right\|.$$

Then, from Theorem 5.1, the following inequality holds with probability at least $1 - 3n\delta$:

$$\sum_{i:o_{2i+1}\in\mathcal{O}} \|\boldsymbol{\beta}^* - \boldsymbol{\beta}_{2i+1}^{(K+1)}\| \le \sum_{i:o_{2i+1}\in\mathcal{O}} \|\boldsymbol{\beta}^* - \boldsymbol{\beta}_{2i+1}^{(K+1)}\|_1 \le \sum_{i:o_{2i+1}\in\mathcal{O}} \|\boldsymbol{\beta}^*\|_1 e^{-\alpha_i K}, \quad \text{(E.3)}$$

where

$$\alpha_i = -\log\left(1 - \frac{2}{3}\gamma + \gamma(2S-1)\sqrt{\frac{\log d - \log\delta}{ic}} + \gamma\sqrt{\frac{\log S - \log\delta}{ic}}\right).$$

From Cauchy–Schwarz Inequality we have:

$$\gamma(2S-1)\sqrt{\frac{\log d - \log\delta}{ic}} + \gamma\sqrt{\frac{\log S - \log\delta}{ic}} \le 2\gamma(2S-1)\sqrt{\frac{\log d + \log S - \log\delta}{ic}}. \quad \text{(E.4)}$$

Combining Inequalities E.3 and E.4 gives

$$\sum_{i:o_{2i+1}\in\mathcal{O}} \|\boldsymbol{\beta}^* - \boldsymbol{\beta}_{2i+1}^{(K+1)}\| \le \|\boldsymbol{\beta}^*\|_1 \sum_{i:o_{2i+1}\in\mathcal{O}} \left(1 - \frac{2}{3}\gamma + 2\gamma(2S-1)\sqrt{\frac{\log d + \log S - \log\delta}{ic}}\right)^K.$$

Note that, by induction, it is straightforward to prove that the following inequality holds if $1 - \frac{2}{3}\gamma \le 1$:

$$\left(1 - \frac{2}{3}\gamma + 2\gamma(2S-1)\sqrt{\frac{\log d + \log S - \log\delta}{ic}}\right)^K$$
$$\le \left(1 - \frac{2}{3}\gamma\right)^K + \eta_i K\left(1 - \frac{2}{3}\gamma\right)^{K-1}\sqrt{\frac{\log d + \log S - \log\delta}{ic}}, \quad \text{(E.5)}$$

when $\eta_i$ satisfies

$$\eta_i \ge \frac{\left(1 - \frac{2}{3}\gamma\right)}{\left(1 - \frac{2}{3}\gamma\right) - (K-1)\sqrt{\frac{\log d + \log S - \log\delta}{ic}}}, \quad \text{(E.6)}$$

which can be satisfied by simply setting

$$\eta_i = \eta = \frac{\left(1 - \frac{2}{3}\gamma\right)}{\left(1 - \frac{2}{3}\gamma\right) - (K-1)\sqrt{\frac{\log d + \log S - \log\delta}{N_0 c}}}. \quad \text{(E.7)}$$

Therefore, Inequality (E.6) holds for any $i \ge N_0$. Based on Inequality (E.5), we have

$$\sum_{i:o_{2i+1}\in\mathcal{O}} \left(1 - \frac{2}{3}\gamma + 2\gamma(2S-1)\sqrt{\frac{\log d + \log S - \log\delta}{ic}}\right)^K$$
$$\le \sum_{i=N_0+1}^{N_0+np-\frac{p}{2}} \left(1 - \frac{2}{3}\gamma + 2\gamma(2S-1)\sqrt{\frac{\log d + \log S - \log\delta}{ic}}\right)^K$$
$$\le \sum_{i=N_0+1}^{N_0+np-\frac{p}{2}} \left(1 - \frac{2}{3}\gamma\right)^K + \eta K\left(1 - \frac{2}{3}\gamma\right)^{K-1} \sum_{i=N_0+1}^{N_0+np-\frac{p}{2}} \sqrt{\frac{\log d + \log S - \log\delta}{ic}}$$
$$\le \left(np - \frac{p}{2}\right)\left(1 - \frac{2}{3}\gamma\right)^K + \eta K\left(1 - \frac{2}{3}\gamma\right)^{K-1}\sqrt{\frac{\log d + \log S - \log\delta}{c}} \sum_{i=N_0+1}^{N_0+np-\frac{p}{2}} \frac{1}{\sqrt{i}}$$
$$\le \left(np - \frac{p}{2}\right)\left(1 - \frac{2}{3}\gamma\right)^K + \eta K\left(1 - \frac{2}{3}\gamma\right)^{K-1}\sqrt{\frac{\log d + \log S - \log\delta}{c}} 2\left(\sqrt{N_0 + np - \frac{p}{2}} - \sqrt{N_0}\right).$$
$$\text{(E.8)}$$

By combining Inequalities (E.3) and (E.7), we obtain

$$\sum_{i:o_{2i+1}\in\mathcal{O}}\|\boldsymbol{\beta}^*-\boldsymbol{\beta}_{2i+1}^{(K+1)}\|$$

$$\leq (np-\frac{p}{2})\left(1-\frac{2}{3}\gamma\right)^K+\eta K\left(1-\frac{2}{3}\gamma\right)^{K-1}\sqrt{\frac{\log d+\log S-\log\delta}{c}}$$

$$\leq (np-\frac{p}{2})\left(1-\frac{2}{3}\gamma\right)^K+2\eta K\sqrt{np-\frac{p}{2}}\left(1-\frac{2}{3}\gamma\right)^{K-1}\sqrt{\frac{\log d+\log S-\log\delta}{c}}.$$

Therefore,

$$\frac{1}{np-\frac{p}{2}}\sum_{i=N_0+1}^{n}\|\boldsymbol{\beta}^*-\boldsymbol{\beta}_{2i+1}^{(K+1)}\|\leq\left(1-\frac{2}{3}\gamma\right)^K+\frac{2\sqrt{2}\eta K}{\sqrt{np}}\left(1-\frac{2}{3}\gamma\right)^{K-1}\sqrt{\frac{\log d+\log S-\log\delta}{c}}. \tag{E.9}$$

Combining Equation (E.1), Equation (E.2) and Equation (E.9) we obtain

$$\|y_n-\widehat{y}_n\|$$

$$\leq b_{\mathbf{x}}\left(1-\frac{2}{3}\gamma\right)^K+\frac{2\sqrt{2}b_{\mathbf{x}}\eta K}{\sqrt{np}}\left(1-\frac{2}{3}\gamma\right)^{K-1}\sqrt{\frac{\log d+\log S-\log\delta}{c}}+\frac{2\epsilon}{np}|B'|+\frac{2C_{\mathbf{x}}b_{\mathbf{x}}\epsilon}{np}.$$

Set $\epsilon=K\sqrt{np}(1-2\gamma/3)^K$, we obtain

$$\|y_n-\widehat{y}_n\|\leq b_{\mathbf{x}}\left(1-\frac{2}{3}\gamma\right)^K+\frac{c_4 K}{\sqrt{n}}\left(1-\frac{2}{3}\gamma\right)^{K-1},$$

where

$$c_4=\frac{2\sqrt{2}b_{\mathbf{x}}\eta}{\sqrt{p}}\sqrt{\frac{\log d+\log S-\log\delta}{c}}+\frac{2|B'|}{\sqrt{p}}+\frac{2C_{\mathbf{x}}b_{\mathbf{x}}}{\sqrt{p}}.$$

Then, the theorem follows by denoting $\delta'$ as $\delta'=\mathbb{P}\{\neg\mathcal{E}_1\}+\mathbb{P}\{\neg\mathcal{E}_2(K\sqrt{np}(1-2\gamma/2)^K)$. $\square$

Finally, we establish the result concerning the label prediction loss presented in Theorem 5.2 through the following corollary:

**Corollary E.1.** *Suppose Assumption 1 holds. For a Transformer with $K$ layers, where all layer structures are described in Section 4.3, and the input sequence is set as in Equation (C.4), there exists a set of parameters in the Transformer and an explicitly quadratic readout function $F$ such that, if $n\geq N_0$, the following inequality holds with probability at least $1-\delta$:*

$$\|y_{n+1}-\widehat{y}_{n+1}\|\leq c_6 e^{-\alpha_n K},$$

*where $c_6=\sqrt{d}b_{\mathbf{x}}c_1$, and $c_1$ and $\alpha_n$ are defined in Theorem 5.1.*

*Proof of Corollary E.1.* We set the $\mathbf{V}$ matrix in the read out function $F_{\mathbf{V}}$ as

$$\mathbf{V}=\begin{bmatrix}\mathbf{0}_{(d+1)\times d}\\\mathbf{I}_{d\times d}\\\mathbf{0}_{1\times d}\end{bmatrix}.$$

From Proposition 1, when $i$ is an odd index, $\mathbf{h}_i^{(K+1)}$ is of the following structure:

$$\mathbf{h}_i^{(K+1)}=\begin{bmatrix}[\mathbf{X}^\top]_{:,\frac{i+1}{2}}\\0\\\boldsymbol{\beta}_i^{(K+1)}\\1\end{bmatrix}.$$

Therefore, the output after the read-out function is

$$\widehat{y}_{\frac{i+1}{2}} = F_{\mathbf{V}}(\mathbf{h}_i^{(K+1)}) = [\mathbf{X}]_{\frac{i+1}{2},:}\boldsymbol{\beta}_i^{(K+1)}.$$

Note that $\left\|[\mathbf{X}]_{\frac{i+1}{2},:}\right\| \leq \sqrt{d}b_{\mathbf{x}}$, we have

$$\left\|y_{\frac{i+1}{2}} - \widehat{y}_{\frac{i+1}{2}}\right\| \leq \left\|[\mathbf{X}]_{\frac{i+1}{2},:}\right\|\left\|\boldsymbol{\beta}^* - \boldsymbol{\beta}_i^{(K+1)}\right\| \leq \sqrt{d}b_{\mathbf{x}}\left\|\boldsymbol{\beta}^* - \boldsymbol{\beta}_i^{(K+1)}\right\|.$$

Combining with Theorem 5.1, the proof is thus complete. □

Therefore, by combining Corollary E.1 and Theorem E.1, we arrive at the conclusion of Theorem 5.2.

## F ADDITIONAL EXPERIMENT RESULTS

In Section 6, we demonstrate that classical LISTA-type algorithms, such as LISTA, perform poorly when applied to varying measurement matrices after being trained on a fixed measurement matrix $\mathbf{X}$. In this section, we further show that unlike LISTA-VM, these classical LISTA-type algorithms fail to handle in-context sparse recovery problems, even when trained on varying measurements.

The training setup is identical to how we train LISTA-VM. Specifically, we set the number of iterations to $K = 12$. During each epoch, we randomly sample 100 measurement matrices, each generating 500 instances from 500 randomly generated sparse vectors, resulting in a total of 50,000 instances per epoch. For each epoch, we minimize the sparse vector prediction loss $\sum_{j=1}^{N} \|\widehat{\boldsymbol{\beta}}_j - \boldsymbol{\beta}_j\|^2$ using gradient descent. We train the model for a total of 340 epochs and conduct in-context testing at the end of every epoch.

In the results shown in Figure 2, we observe that the prediction error on varying $\mathbf{X}$ for meta-trained LISTA and LISTA-CP remains around 3. As illustrated in the figure, there is no observable trend indicating improvement in the error throughout the training process.

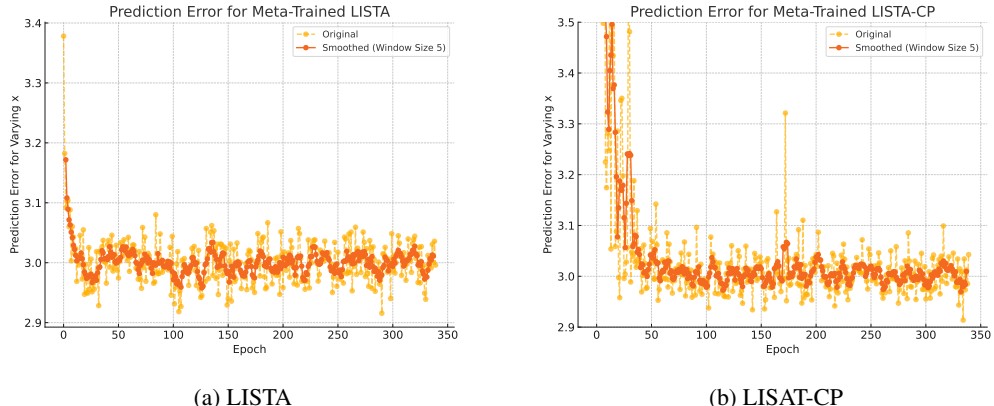

(a) LISTA                                (b) LISAT-CP

Figure 2: Experimental results for meta-trained classic LISTA-type algorithms.

For comparison, in Figure 3, we also provide the results of the prediction error on varying $\mathbf{X}$ for meta-trained LISTA-VM. The final prediction error is around $0.68$, which is significantly more promising compared to classic LISTA-type algorithms.

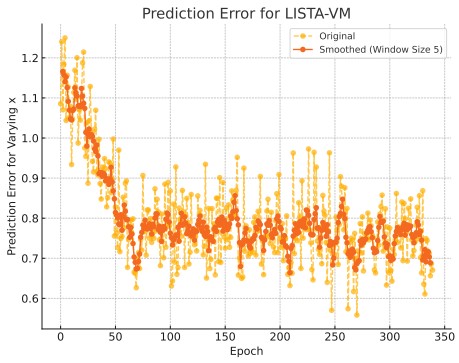

Figure 3: Experimental results for meta-trained LISTA-VM

