# OpenReview forum: "On the Learn-to-Optimize Capabilities of Transformers in In-Context Sparse Recovery"
_ICLR.cc/2025/Conference — ICLR 2025 Poster_

### Official Review · Reviewer_FsZo · 2024-10-31

**Soundness:** 3
**Presentation:** 4
**Contribution:** 4
**Rating:** 6
**Confidence:** 4

**Summary:**

The authors of this paper study sparse recovery using transformers in an in-context scenario. They provided theoretical results and conducted experiments regarding the ability of transformers to execute an L2O algorithm.

**Strengths:**

The paper is well-written, with clearly presented theoretical findings and compelling experimental results. The insights are both intriguing and hold valuable potential for contributing to the scientific community.

**Weaknesses:**

I have two concerns which should be better explained in the paper:
- The embedding form in Equation 4.2 appears somewhat ad hoc; do you have any references supporting this choice? I understand that this choice is closely tied to the design of the four attention heads in Appendix C1, so I wonder: are alternative forms feasible, and if so, how might they impact the theoretical results?

- In the experimental section, the authors test with GPT-2 using 8 heads, which differs from the 4-head configuration in the theoretical analysis. What motivates this discrepancy? Additionally, I encourage the authors to discuss how varying the number of heads might affect the theoretical findings.

**Questions:**

See in the comments in Weakness.

---

> ### Author Response · Authors · 2024-11-22
> **Response to Reviewer FsZo**
>
> We thank the reviewer for the careful reading and thoughtful comments. Please see our responses below.
>
> **Weakness 1:** The embedding form in Equation 4.2 appears somewhat ad hoc; do you have any references supporting this choice?
>
> **Response to Weakness 1:** We appreciate the reviewer’s insights. The embedding form we used is inspired by similar approaches in [1], where placeholders in the embedding are utilized for implicit parameter updates and indicators. We revised our manuscript from line 305 to line 306 to clarify this similarity.
>
> We acknowledge that alternative embeddings and attention head configurations could also demonstrate the Transformer’s capability to implement L2O algorithms. For instance, embedding positional information (e.g., index \(i\)) directly into the input sequence, as demonstrated in Section B.2 of [1] for decoder-based Transformers, could yield similar convergence results (linear convergence rate) while simplifying certain aspects of the proof. However, in this work, we intentionally omitted positional information from the embedding because it is not essential for the in-context sparse recovery task. Our goal was to keep the embedding structure as simple and task-specific as possible to focus on the critical elements necessary for in-context sparse recovery.
>
> **Weakness 2:** In the experimental section, the authors test with GPT-2 using 8 heads, which differs from the 4-head configuration in the theoretical analysis. What motivates this discrepancy? Additionally, I encourage the authors to discuss how varying the number of heads might affect the theoretical findings.
>
> **Response to Weakness 2:** We thank the reviewer for the careful reading. The experimental use of GPT-2 with 8 heads was chosen to evaluate the Transformer’s performance in a standard configuration widely used in practical applications. To ensure alignment with the theoretical analysis, we included the *Small TF* model in our experiments, which strictly adheres to the 4-head configuration and ReLU activation function as described in the theoretical setup. We revised our manuscript from line 474 to line 476 to clarify the purpose of introducing GPT-2 model.
>
> Increasing the number of heads improves the model’s capacity to capture diverse features, potentially enhancing generalization in sparse recovery tasks. This could lead to a more effective L2O algorithm compared to the LISTA-VM implementation introduced for the 4-head Transformer structure. We thank the reviewer for this insightful suggestion and will explore in future work whether additional heads can result in a more efficient L2O algorithm and further optimize the performance in in-context sparse recovery tasks.
>
> **Reference**
>
> [1] Bai, Yu, et al. "Transformers as statisticians: Provable in-context learning with in-context algorithm selection." Advances in neural information processing systems 36 (2024).

---

> > ### Comment · Reviewer_FsZo · 2024-11-27
> >
> > Thank you for the responses. My comments are well addressed.

---

### Official Review · Reviewer_7MPy · 2024-11-02

**Soundness:** 3
**Presentation:** 3
**Contribution:** 3
**Rating:** 8
**Confidence:** 4

**Summary:**

This paper threw an interesting hypothesis: the trained transformers essentially play the role of the L2O algorithm when solving the inverse problem. Enriched theory analysis and derivation are proposed, which are almost correct. This hypothesis is validated underlying a sparse recovery problem with respect to ISTA-based alternating algorithms. The fundamental idea can be summarized as proving the convergence and accuracy follow a similar criterion in ISTA-variant algorithms. Simulations on the trained transformer and ISTA-based algorithms for solving sparse recovery tasks are presented, and the blind inverse problem is particularly picked out to show the stronger performance of the transformer, and thus leads to the L2O capability.

**Strengths:**

The manuscript is articulated with clarity and is accessible for comprehension.
The proposed hypothesis presents a valuable avenue for in-depth analysis.
Preliminary concepts are provided with comprehensive equations and definitions.
Elaborate proofs are accessible in the appendix.
The quality of English is commendable and articulated naturally.

**Weaknesses:**

I made a rejection at this stage as the following shortbacks, I am willing to change my score after the rebuttal phase with respect to the author's responses.
1. The entire work is divided. The primary concept, prove that transformer has the L2O approximation capability in forward process, is isolated to the main task, sparse recovery problem. In this version ,this paper tends to mash-up the L2O approximation on transformer and sparse recovery problem. Essentially, each of them deserves to publish an independent paper.
2. Followed by 1, this causes the following issues: i) no analysis and illustrations on the necessity of L2O for sparse recovery tasks; ii) the bridge between the fundamental proofs and evaluation tasks is fragile; iii) the increments of this paper compared to . Von Oswald et al. (2023a) and  Bai et al. (2024) are invisible, though the authors claim the difference is the L2O consideration.
3. The writing of this paper is also a mixture. The logical flow of the introduction is saltatory. The fundamental motivation is not clear enough to convince me that the L2O algorithmic principle is the explanation of the success of the transformer because of there are L2O-based algorithms that achieve similar performance.
4. The whole section 3 is overwhelming. I don't think it deserves so much context to present details network layer symbols and definition. Besides, Definition 3.1-3.3 is kind of a definition abuse. I highly suggest a concise and shorter version of this section.
5. The key parts, Sections 4.3, 5.1, and 5.2, show a trivial way of proving the author's hypothesis. Performing LISTA-type algorithms do not provide a direct derivation of the L2O of the transformer forward process, at least to me.
6. Last but the most fatal issue lies in simulation. The authors tend to show the superior performance of a trained transformer compared to those alternating minimization-based ISTA-type algorithms, concluding the L2O capability of transformer in the blind inverse problem of sparse recovery where matrix X is not given. This is unfair and not reasonable. The trained transformer obviously surpasses those model-based algorithms which are designed for matrix X is given. The transformer solves the blind inverse problem due to the data-driven prior from the training process, instead of L2O capability that focuses on improving the optimization strategy during the iterations. A fair comparison should be set among algorithms with learning ability and capable of solving blind inverse problems, such as
[1] Zero-shot image restoration using denoising diffusion null-space model
[2] Metalearning-based alternating minimization algorithm for nonconvex optimization
[3] Invertible Diffusion Models for Compressed Sensing
[4] OmniSSR: Zero-shot Omnidirectional Image Super-Resolution using Stable Diffusion Model
[5] Blind Super-Resolution Via Meta-Learning and Markov Chain Monte Carlo Simulation

**Questions:**

See weakness

---

> ### Author Response · Authors · 2024-11-22
> **Response to Reviewer 7MPy, part 1**
>
> We thank the reviewer for the careful reading and thoughtful comments. Please see our responses below.
>
> **Weakness 1:** The entire work is divided. The primary concept is isolated to the main task, the sparse recovery problem.
>
> **Response to Weakness 1:** We thank the reviewer for the comments. We would like to clarify the relationship between the L2O capability of Transformers and the sparse recovery problem as follows.
>
> The empirical successes of Transformers during in-context learning motivate a line of work aiming to explain the mechanism behind the ICL capabilities of Transformers. Previous works [3,5] show that Transformers can perform gradient-descent algorithms, which enables the ICL capability. However, the slow (e.g., sublinear) convergence of first-order gradient descent-based algorithms may not be sufficient to explain the superb performance of Transformers in ICL in general. This observation motivates researchers to explore alternative explanations behind ICL, and our work is along this line and trying to verify the hypothesis that Transformers' Learn-to-Optimize capability is the enabler of superb ICL performance.
>
> While L2O is broadly defined, to materialize the hypothesis, we choose in-context sparse recovery as an example task that Transformers perform. Our selection is due to the following reasons.  Sparse recovery is a classical signal processing problem that is of significant practical interest across various domains, such as compressive sensing in medical imaging and spectrum sensing.
> Recent works show that Transformers can implement gradient descent-based algorithms with sublinear convergence rates for in-context sparse recovery [3,4]. However, empirical findings indicate that Transformers can solve in-context sparse recovery more efficiently than gradient descent-based approaches [3]. Meanwhile, there exists a plethora of L2O algorithms that solve the classical sparse recovery problem efficiently with linear convergence guarantees [1,2]. Therefore, examining the L2O capabilities of Transformers in solving the in-context sparse recovery task becomes a natural and promising selection to validate our hypothesis.
>
> We have revised the Introduction (Line 69 - Line 75) to clarify this connection more explicitly.

---

> ### Author Response · Authors · 2024-11-22
> **Response to Reviewer 7MPy, part 2**
>
> **Weakness 2:** i) no analysis and illustrations on the necessity of L2O for sparse recovery tasks; ii) the bridge between the fundamental proofs and evaluation tasks is fragile; iii) the increments of this paper compared to. Von Oswald et al. (2023a) and Bai et al. (2024) are invisible.
>
> **Response to Weakness 2:** We thank the reviewer for the comments and address each point as follows:
>
> * **Necessity of L2O for Sparse Recovery Tasks:** We agree that L2O algorithms are not necessary for sparse recovery. As noted in lines 274–288, there exist traditional gradient-based algorithms, such as ISTA, as well as L2O-based algorithms, such as LISTA-CP, in the literature. Our objective, however, is not to show the necessity of L2O for sparse recovery problems but to use sparse recovery as an example task to show that Transformers **can** perform L2O algorithms to solve the sparse recovery task during ICL.
>
> * **Connection Between Proofs and Evaluation Tasks:** Our proof shows that   Transformers have the capability to implement an L2O algorithm for in-context sparse recovery tasks (Theorem 4.1) with a linear convergence rate (Theorem 5.1). In our experiments, we focus on the performances of Transformers in solving in-context sparse recovery, and compare with both classical gradient-based and L2O algorithms under the same setting. Our experimental results indicate that the performance of a Transformer with the structure specified in Theorem 4.1 (labeled as 'Small TF' in Figure 1) achieves comparable performances with the L2O algorithms. Therefore, we believe the evaluation tasks align well with our theoretical results.
>
> * **Improvements Over Related Work:** We respectively disagree with the reviewer on the difference between our work and existing works. First, while previous works demonstrate Transformers’ capability to implement gradient descent-based algorithms, our work indicates that Transformers can actually perform L2O algorithms on sparse recovery for the first time, **advancing our understanding of the ICL mechanism of Transformers**. Second, as stated in lines 78–81, the key improvement of the theoretical results compared to the related works [3,4] lies in the **convergence rate**. Previous works only demonstrate sublinear convergence rates for Transformers to implement gradient descent-based algorithms for sparse recovery, while our work highlights **linear convergence guarantees** for Transformers implementing L2O algorithms. Third, our proof techniques fundamentally differ from those used in related works. Specifically, previous studies [3,5] focus on analyzing the Transformer's ability to approximate gradient descent updates through its attention mechanism. In contrast, we introduce a novel approach where the learned matrices  $\{\mathbf{M}^{(k)}\}$ serve as approximations of the inverse Hessian, incorporating curvature information into the Transformer's update rules. This effectively enables the updating rule to function as an L2O algorithm. This technique is a key factor that allows for linear convergence, distinguishing our theoretical framework from the gradient-based analyses in related works.
>
> **Weakness 3:** The writing of this paper is a mixture.
>
> **Response to Weakness 3:** We thank the reviewer for the comment. We have revised our paper (Lines 69-75) to streamline the flow of the work and clarify the connection between our major hypothesis (i.e., the L2O capability of Transformers) and the specific task (i.e., sparse recovery) we choose to examine. We would like to point out that providing a possible explanation of the mechanism behind the ICL capabilities of Transformers has gained increasing attention in the research community. Our research falls this line of work and advances the state of the art in the sense that it provides a more plausible explanation for the superb ICL performance of Transformers.
>
> **Weakness 4:** Section 3 is overwhelming.
>
> **Response to Weakness 4:** We appreciate the reviewer's suggestion. In our revised version, we have condensed Section 3 to focus on the essential architectural elements required for understanding the theoretical results and empirical evaluations.
>
> **Weakness 5:** The key parts show a trivial way of proving the author's hypothesis
>
> **Response to Weakness 5:** We would like to clarify that our main objective is to demonstrate the *capability* of Transformers to implement an L2O algorithm for effectively solving in-context sparse recovery tasks. Our way to prove the hypothesis resembles approaches in previous works, such as [3,5,6]. For instance, in [6], the authors verify the hypothesis that the ICL capability of Transformers may be enabled by their ability to perform gradient descent. To achieve this, they first demonstrate that Transformers can implement various types of operators, which they then use to show the capacity of Transformers to perform gradient descent for linear regression by constructing such operators explicitly.

---

> ### Author Response · Authors · 2024-11-22
> **Response to Reviewer 7MPy, part 3**
>
> **Weakness 6:** The authors tend to show the superior performance of a trained transformer compared to those alternating minimization-based ISTA-type algorithms, concluding the L2O capability of transformer in the blind inverse problem of sparse recovery where matrix X is not given, is not reasonable. Should compare with algorithms with the learning ability and capability of solving blind inverse problems.
>
> **Response to Weakness 6:**  It seems that the reviewer has some misunderstanding of the problem setup, which we would like to clarify as follows:
>
> * First of all, the problem Transformers and all baseline algorithms aim to solve is **not** a blind inverse problem. Instead, all of them are solving **the same sparse recovery problem**, where the sensing matrix is **given**.
>
> * Second, for the L2O baseline algorithms considered in our experiments, they also need to be pre-trained, similar to the way we pre-train the Transformer with the structure described in Theorem 4.1 (labeled as 'Small TF' in Figure 1) and GPT-2. Thus, similar to the Transformers, they are able to extract "data-driven prior from the training process". That's essentially the reason why the L2O algorithm can surpass gradient-based iterative algorithms and achieve faster convergence.
>
>     In this work, we consider the task: in-context sparse recovery problem, which is **not** a blind inverse problem. Instead, the problem assumes that the measurement matrices are **provided** during both pre-training and inference, enabling the Transformer to leverage structural information from the matrices and sparse vectors to perform efficient recovery. This setup differs from blind inverse problems, where the measurement matrix is unknown and must be inferred. Our task formulation ensures a fair comparison with other methods designed for the given matrix scenario.
>
> * We appreciate the reviewer providing references to related works, including diffusion-based methods. However, as this work focuses on in-context sparse recovery (not blind inverse problems), comparing our results with diffusion-based models is neither applicable nor necessary. Additionally, in-context learning for diffusion models remains an under-explored area in the literature, and even formulating the problem for these models in the context of sparse recovery lacks sufficient literature at this stage.
>
> We hope this explanation clarifies our problem setup.
>
> **References**
>
> [1] Chen, Xiaohan, et al. "Theoretical linear convergence of unfolded ISTA and its practical weights and thresholds." Advances in Neural Information Processing Systems 31 (2018).
>
> [2] Liu, Jialin, and Xiaohan Chen. "ALISTA: Analytic weights are as good as learned weights in LISTA." International Conference on Learning Representations (ICLR). 2019.
>
> [3] Bai, Yu, et al. "Transformers as statisticians: Provable in-context learning with in-context algorithm selection." Advances in neural information processing systems 36 (2024).
>
> [4] Chen, Xingwu, Lei Zhao, and Difan Zou. "How transformers utilize multi-head attention in in-context learning? a case study on sparse linear regression." arXiv preprint arXiv:2408.04532 (2024).
>
> [5] Von Oswald, Johannes, et al. "Transformers learn in-context by gradient descent." International Conference on Machine Learning. PMLR, 2023.
>
> [6]  Akyürek, Ekin, et al. "What learning algorithm is in-context learning? investigations with linear models." arXiv preprint arXiv:2211.15661 (2022).

---

> > ### Comment · Reviewer_7MPy · 2024-11-22
> > **L2O definition mis-understanding  and related discussions**
> >
> > I appreciate the authors for their comprehensive responses. While I am amenable to the revision commitments made in the context of their reply, several concerns remain unclear to me:
> >
> > i) The conceptualization and comprehension of Learning to Optimize (L2O) require clarification. L2O can be understood in light of Finn’s work, which emphasizes adaptation to new tasks through optimization strategies developed during pre-training on sets of queries and supports. Moreover, the references provided illustrate the application of L2O strategies derived from solution processes to specific optimization problems. A thorough examination of these categories would reveal their fundamental distinctions.
> >
> > ii) In relation to the previous point, the principal confusion and contention within the proposed work arise from the claim that all baseline models are pre-trained and the derivation endeavors to demonstrate the mechanism of transformers in enabling L2O. The notion of L2O concerning varying sensing matrices posits the ability to generalize to a new sparse recovery problem with a slightly altered sensing matrix. This proposition appears incongruous since end-to-end models are capable of achieving this, and it can be adequately explained by deep learning rather than L2O. Hence, it appears that the authors seek to substantiate the transformer’s forward process using an L2O mechanism that optimizes the sparse recovery problem beyond the gradient-based algorithm, providing a robust and adaptive optimization strategy throughout the iterative resolution of this task, as noted when ISTA depends on a given matrix. Specifically, the diffusion employed in this mechanism serves as a static optimizer to incorporate data priors and implement an L2O strategy in addressing these inverse problems.
> >
> > iii) At this juncture, the cited references and additional solutions that similarly learn to optimize inverse problems in an alternating manner, akin to ISTA, should be incorporated in the simulation case for fairness and comprehensiveness. Despite the challenges of integrating such simulations during the rebuttal stage, it is crucial to elucidate a clear definition and focus for L2O, alongside illustrating these connections.
> >
> > iv) As the response indicated, all baselines are pre-trained. When compared with the transformer, it appears unjustified and irrational to demonstrate L2O in resolving an inverse problem based on sparse recovery. The L2O mechanism must be substantiated through either adaptive learning capabilities in distinct sparse recovery scenarios—with markedly different sensing matrices—or via L2O abilities pertinent to optimization strategies for the same sparse recovery task. It seems the authors may not fully grasp the distinctions between these basic definitions.
> >
> > v) Consequently, the current exposition suggests that the authors initially intended to enhance the concept of L2O as a more advanced form of gradient-based algorithms, in line with prevailing research trends. However, an inadequate understanding and definition of L2O may lead to misconceptions regarding its foundational mechanism for solving a series of tasks or a specific sparse recovery task. Upon thorough examination of the prior works mentioned, this research approach appears aligned with the latter concept, which accounts for the favorable performance of transformers via a gradient-based optimization strategy. In the context of L2O, the disparity is greater than what is proffered in this paper. Simply replicating previous research trajectories is insufficient to convincingly demonstrate L2O as the success factor for transformers rather than gradient-based algorithms, nor is the focus on specific L2O scenarios delineated clearly.
> >
> > Could you please provide an elucidation of the critical issues at hand and deliver a comprehensive explanation of what constitutes the essence of L2O in your research?

---

> > > ### Author Response · Authors · 2024-11-25
> > > **Response to Reviewer 7MPy's comments, part 1**
> > >
> > > Dear Reviewer 7MPy,
> > >
> > > We thank the reviewer for the prompt feedback for our rebuttal. We are especially thankful for the thoughtful and constructive comments. We appreciate the opportunity to further clarify the conceptualization of L2O in our work.
> > >
> > > **1) The Learning to Optimize (L2O) framework and its conceptualization in sparse recovery.**
> > > The L2O framework, as summarized in the review paper [1], is an optimization paradigm that develops an optimization method (i.e., a solver) by training across a set of similar problems (tasks) sampled from a task distribution. While the training process is often offline and time-consuming, the objective of L2O is to improve the optimization efficiency and accuracy when the method is deployed online and any new task sampled from the same distribution is encountered.
> > >
> > > As a general optimization framework, L2O has demonstrated its advantage over classical static optimization frameworks in various optimization problems and applications. Sparse recovery is arguably one of the most representative ones, with several L2O algorithms well studied and even theoretically analyzed, such as LISTA, ALISTA, LISTA-CP, etc. Figure 2 in [1] shows that those L2O algorithms converge much faster than the two popular iterative solvers ISTA, FISTA. We note that for those L2O solvers, the training tasks are **randomly generated by fixing the sensing matrix $X$ but varying the underlying sparse vector $\beta$**. The trained L2O solvers are able to solve different sparse recovery tasks (testing instances) randomly generated in the same way.
> > >
> > > **2) The hypothetical L2O capability of Transformers during ICL.**
> > > In this work, our focus is the underlying in-context learning (ICL) mechanism of Transformers in solving complicated optimization tasks, such as sparse recovery. While existing works show that Transformers can perform classical gradient descent (GD) based iterations layer-wisely, we think it does not adequately explain the fast convergence of the ICL solution for Transformers **with only a few layers**. This is because GD is known to be slow (sub-linear convergence) in solving complicated optimization tasks; Thus, a Transformer with a few layers should not be able to produce accurate results if GD is indeed what it performs during ICL.
> > >
> > > Motivated by this observation, we conjecture that Transformers may actually perform certain L2O algorithms during ICL, i.e., it develops a solver during its pre-training with various training tasks sampled from a task distribution, and then adopts the trained solver on tasks sampled from the same distribution during ICL.
> > >
> > > **3) Examine the hypothesis through the lens of sparse recovery.**
> > > To verify our hypothesis, we have to substantiate it on certain specific ICL tasks. Sparse recovery becomes a natural selection, due to its inherent optimization complexity, and the existing well-studied L2O algorithms to solve it. The intuition is, if we are able to demonstrate that Transformers can perform iterative operations layer-wisely as in those L2O algorithms, L2O can be a plausible explanation for the fast convergence during ICL.
> > >
> > > Our theoretical results in Theorem 4.1 and Theorem 5.1 show that, indeed, there exists a Transformer that is able to **perform similar iterative operations as in LISTA** (one prominent L2O algorithm in solving sparse recovery), with **provable linear convergence** during ICL. Those results significantly improve the SOTA sublinear convergence achieved under GD-based operations and match with the observed fast convergence during ICL. Therefore, it serves as strong evidence to validate our hypothesis.
> > >
> > > **4) Generalization across varying sensing matrices.**
> > > We point out that the Transformer constructed in Theorem 4.1 not only mimics the iterative operations of LISTA, it actually results in a more robust and adaptive version of LISTA, termed as LISTA-VM. Here VM stands for "Varying sensing Matrices". As noted in 1), all existing L2O algorithms, such as LISTA, are designed for **a fixed sensing matrix $X$**, where different training and testing tasks are generated by sampling $\beta$ according to a distribution. While we can easily design a Transformer to replicate the operations of LISTA for this setting, the Transformer described in Theorem 4.1 enhances LISTA in the sense that **it can handle varying sensing matrice** as well.
> > >
> > > Nevertheless, the claimed L2O capability of Transformers is mainly based on 3). The generalization capability across varying $X$ is an interesting by-product of our analysis, and is not the main evidence we use to support our hypothesis.

---

> ### Author Response · Authors · 2024-11-25
> **Response to Reviewer 7MPy's comments, part 2**
>
> **5) Incorporate other L2O solvers in simulation.** We thank the reviewer for the suggestion. We currently have studied three L2O baseline algorithms, namely LISTA, ALISTA, LISTA-CP, in our experiments, together with two classical iterative algorithms, namely ISTA and FISTA. We have also examined LISTA-VM, the new enhanced version of LISTA. We are actively exploring other L2O algorithms in the context of sparse recovery, and will report the results when they become available.
>
> Thank you once again for your valuable feedback. We hope that these explanations address your concerns adequately. We are more than happy to address any additional questions or concerns you may have.
>
> Warm regards,
>
> The Authors

---

> ### Comment · Reviewer_7MPy · 2024-11-25
> **Rebuttal feedback and decision**
>
> Thank you for the clarifications. The authors’ rebuttal feedback has addressed most of my concerns, and I am satisfied with the clarifications and demonstration in the manuscript regarding the L2O mechanism presented in the revised paper. Though, the varying beta does not align with my expectations concerning learning to optimize within sparse recovery tasks, as it typically exhibits less variance than the measurement matrix. I believe it is acceptable to independently complete the proof of the proposed hypothesis: the transformer forward process demonstrates an approximation of learning-involved iterative algorithms beyond the vanilla and variant gradient descent algorithms. Nevertheless, I strongly recommend that the authors further explore learning to optimize over iterations with a fixed inverse problem, referencing "learning to learn gradient descent by gradient descent" and its variants for multi-variable optimization tasks, as suggested in the recommended reference. These methods, which are employed to learn a dynamic optimization strategy based on the trajectory of iterations, are more appropriate for the proposed learning to optimize mechanism than training-based schemes.
>
> Lastly, I highly recommend executing a demonstration revision prior to the submission of the final version. This strategic action follows the insightful 2nd round discussions regarding the comprehensive definition of L2O and the essential clarifications in the introductory section. Accomplishing this before you finalize your version will provide a dual advantage: captivating readers who are deeply entrenched in the specialized research of ICL explanation while simultaneously broadening the scope to attract wider potential citations from the expansive fields of optimization and the machine learning community. At this stage, I would like to raise my final score to accept and look forward to following your final version in the near future.

---

> > ### Author Response · Authors · 2024-11-25
> > **Thank you for your response**
> >
> > Dear Reviewer 7MPy,
> >
> > We are glad to know that our responses have addressed most of your concerns, and we truly appreciate that you recognize the contributions of this work and increase your rating! Your thoughtful comments have helped us greatly improve the quality of this work, and we will definitely keep polishing the paper and incorporating your valuable suggestions into the final version.
> >
> > Thank you for pointing us to the interesting reference. The Learning to learn approach seems closely related to the L2O framework we consider, and it may give us a new and promising perspective to interpret the ICL mechanism of Transformers. We will take a closer look at the reference and explore this new direction as our next step.
> >
> > Thank you again for the insightful comments and suggestions!
> >
> > Warm regards,
> >
> > The Authors

---

### Official Review · Reviewer_bS8s · 2024-11-03

**Soundness:** 3
**Presentation:** 2
**Contribution:** 2
**Rating:** 6
**Confidence:** 2

**Summary:**

The authors leverage the ability of Transformers to perform in-context learning (ICL) for solving the sparse recovery problem. It is shown that the Transformers can implement first-order optimization algorithms (such as ISTA) and their learned variants (e.g., LISTA) for sparse approximation. Moreover, it is shown theoretically that a K-layer transformer has a linear convergence rate in K (with high probability). The ability of transformers to solve the sparse recovery problem corresponding to a different sensing matrix than what it was trained on is also demonstrated.

**Strengths:**

1. The theoretical insights on the versatility of the transformer architecture are interesting and useful for a broader class of problems. For example, the fact that LISTA-type algorithms can be implemented using a transformer opens up the possibility of using transformers for building more powerful learned reconstruction operators for more general inverse problems.

2. The convergence rate results (for sparse estimation and prediction) place the proposed approach on a concrete theoretical footing.

3. Generalizability to a different sensing matrix during the inference time is a very powerful property of a learned approach, which (in my knowledge) is not generally possessed by standard learned reconstruction operators trained on a specific measurement matrix.

**Weaknesses:**

1. The notion of ICL in the context of sparse recovery is not very well explained. I would suggest rewriting Sec. 4.1 with a better explanation of the setup (i.e., exactly what the model is trained on and what exactly is given as input to the model during inference).

2. The convergence result (Theorem 5.1) merely ascertains the existence of a set of parameters such that the recovery is accurate with high probability, and does not state anything about the convergence of a pre-trained transformer on a new problem instance.

**Questions:**

- In my opinion, the phrase L2O is not appropriate in the context of this paper (or any approach for constructing a learned network to solve an inverse problem for that matter). These methods essentially try to approximate the conditional mean of the parameter of interest given the measured data and do not find an approximation to a variational reconstruction problem such as LASSO (not without any constraints on the learnable units in the architecture, at least).

- How does a K-layer pre-trained transformer perform if more layers are added during the inference time (by extending the learned layers using some reasonable strategy)? If this leads to divergence, it would not be appropriate to claim that this approach learns to approximate the LASSO solution. This just gives us a more powerful and expressive architecture that does a better job of approximating the conditional mean estimator of the underlying sparse vector from its linear measurement.

- Could you explain whether Figure 1 considers the same sensing matrix during training and inference? Maybe I missed it, but I was not able to figure it out from Section 6.

- How is the support constraint incorporated into your framework? In particular, why does the performance of the proposed method improve with these constraints, while the other baseline methods perform roughly the same?

- While first-order optimization methods for LASSO can only achieve a sublinear rate (and not anything faster, provably), the result in Theorem 5.1 appears paradoxical. I would suggest highlighting the key differences in the setting and assumptions made in the optimization literature and this work to make this bit clear. I understand that the rate in Theorem 5.1 holds with high probability, but is it possible to construct a sparse signal for which a pre-trained transformer fails to achieve a linear rate?

- Assumption 1 seems somewhat non-standard to what is generally assumed about the sensing matrix in the compressed sensing literature. For instance, a Gaussian sensing matric would not satisfy this assumption.

---

> ### Author Response · Authors · 2024-11-22
> **Response to Reviewer bS8s, part 1**
>
> **Weakness**
>
> **Weakness 1:** The notion of ICL in the context of sparse recovery is not well explained.
>
> **Response to Weakness 1:** We appreciate the reviewer’s suggestion. We have revised Section 4.1 from Line 243 to Line 250 to clarify the ICL setup for sparse recovery. Specifically, we detail that the model is trained on various sparse recovery instances, with the measurement matrices, sparse vectors and noise terms randomly sampled from given distributions. A new sparse recovery instance, defined by a unique measurement matrix and query vector, is provided as input during inference. This setup allows the model to demonstrate its ability to generalize across different measurement matrices and recovery tasks.
>
> **Weakness 2:** The convergence result does not state anything about the convergence of a pre-trained transformer on a new problem instance.
>
> **Response to Weakness 2:** We apologize for the confusion. Theorem 5.1 is actually about the convergence performance of a pre-trained Transformer ** on any new randomly generated problem instance during inference**, where $K$ is the number of layers of the Transformer. We have clarified this in the revised draft to avoid confusion.
>
> We note that how to theoretically characterize the convergence dynamics of the Transformer during pre-training and obtain the desired parameters of the Transformer with rigorous guarantee is very challenging, mainly due to the multi-layer structure of the constructed Transformer. Our empirical results in Sec. 6, however, indicate that the pre-training process indeed results in a pre-trained Transformer ('Small TF' in Figure 1) with superb performance on new sparse recovery instances during ICL, corroborating Theorem 5.1.  We leave the theoretical analysis of the pre-training dynamics as our future work.
>
> **Questions:**
>
> **Question 1:** The phrase L2O is not appropriate in the context of this paper.
>
> **Response to Question 1** We appreciate the reviewer’s perspective.
> In the sparse recovery literature, Learning to Optimize (L2O) is traditionally defined as a framework for designing algorithms that iteratively refine solutions to optimization problems such as LASSO. These algorithms, including LISTA [1] and its variants [2,3], are specifically designed for sparse recovery tasks and achieve provable convergence properties.
>
> For a multi-layer Transformer, the model does not simply approximate the conditional mean estimator given measurement data; instead, it iteratively updates its estimation of the sparse vector through a sequence of learned transformations. Each layer of the Transformer performs operations analogous to iterative steps in the L2O algorithms, progressively refining its estimation.
>
> In this work, we demonstrate that Transformers can mimic existing L2O algorithms like LISTA by performing similar iterative optimization steps from one layer to the next layer. Specifically, the Transformer's architecture is capable of encoding and executing optimization steps similar to the LISTA algorithm within its forward pass, allowing it to achieve sparse recovery efficiently. The learned algorithm goes beyond direct conditional mean estimation by embedding sparse vector estimation and iterative refinement into the forward pass of the Transformer.
>
> **Question 2:** How does a $K$-layer pre-trained transformer perform if more layers are added during the inference time?
>
> **Response to Question 2:** As explained in our response to Question 1, in the context of sparse recovery, L2O refers to a collection of solvers with learnable parameters to solve LASSO, where those parameters are obtained during pre-training. The Transformer we consider essentially mimics the behavior of such L2O solvers, with parameters being determined during pre-training as well. Due to the equivalence between the functionality of the pre-trained Transformer and the L2O solvers, we claim that the Transformer has the L2O capability for sparse recovery during ICL.
>
> In Theorem 4.1, we show that a pre-trained Transformer can perform one iteration of the LISTA-type algorithm in each layer. We are not quite sure about the reviewer's point of adding more layers during inference and using it to verify our claim. We would appreciate it if the reviewer could elaborate on it, your elaboration would greatly assist us in addressing your concerns effectively.

---

> > ### Comment · Reviewer_bS8s · 2024-11-24
> >
> > Thanks for the clarification.
> >
> > The phrase L2O, in my understanding, is used in scenarios where a deep network is trained to solve an optimization problem (see, e.g., the review paper "Learning to Optimize: A Primer and A Benchmark" by Chen et al.). Any networks, transformers, convolutional networks, or MLPs, trained on an end-to-end supervised MSE loss, do not necessarily learn to solve an underlying optimization problem (e.g., a variational problem for sparse recovery), unless, of course, additional constraints are imposed on the architecture (see, e.g., "Accelerated forward-backward optimization using deep learning" by Banert et al.). However, this is an issue of semantics, which I am okay with ignoring.
> >
> > What I meant by "adding more layers during inference" is the following: If an L2O architecture, designed as an unrolled network with N layers, is trained to optimize an underlying objective (provably), its performance should not degrade if one adds more than N layers (assume, for simplicity, that the layers share weight and one simply stacks the same pre-trained layer beyond N). See, for instance, the paper entitled "Data-driven mirror descent with input-convex neural networks" by Tan et al.). This is, in general, not true for a network trained on supervised MSE.
> >
> > Overall, the authors' responses helped me understand the paper better and I believe that the paper is a valuable contribution when it comes to building reconstruction networks for inverse problems that are generalizable to new forward operators during inference. I would like to maintain my original evaluation score.

---

> ### Author Response · Authors · 2024-11-22
> **Response to Reviewer bS8s, part 2**
>
> **Question 3:** Could you explain if Figure 1 considers the same sensing matrix during training and inference?
>
> **Response to Question 3:** In Figure 1, for the three baseline algorithms (LISTA, LISTA-CP, and ALISTA), we include experiments that evaluate their performance during inference in two cases: (1) when the measurement matrix remains identical to that used during pretraining, reported as "Fixed X," and (2) when the measurement matrix varies through random sampling, reported as "Varying X." We thank the reviewer for pointing this out and have updated the manuscript accordingly from Lines 459 to 462 to make this clearer.
>
> **Question 4:** How is the support constraint incorporated? Why does the performance of the proposed method improve while others are roughly the same?
>
> **Response to Question 4:** The support constraint is **implicitly captured** during pre-training of the Transformer, where the training instances are generated by sampling  \(\beta\) under the same support constraint. While it is unclear how a general Transformer captures such constraint, Remark 4 suggests that with possible parameters of the Transformer to ensure correct support identification, the estimation error during ICL can be reduced. This effect is elucidated by the significantly improved performance of LISTA-VM-SS compared with LISTA-WM in Figure 1(b), where we set all columns in $\mathbf{X}$ whose indices are not in the prior support set to be zero to capture the support constraint. Baseline methods, however, lack mechanisms to estimate the support and adapt their search spaces based on support constraints, thus can not achieve performance improvement.
>
> **Question 5:** The result in Theorem 5.1 appears paradoxical to first-order optimization methods theory. Is it possible to construct a sparse signal for which a pre-trained transformer fails to achieve a linear rate?
>
> **Response to Question 5:** We appreciate the reviewer’s observation. Theorem 5.1 does not contradict the well-known result that first-order optimization methods converge at a sublinear rate for sparse recovery. This is because our proposed algorithm incorporates curvature information into the update rule. As detailed in the *Main challenge and key ideas of the proof* section following Theorem 5.1, the learned $\mathbf{M}^{(k)}$ matrices serve as an approximation of the inverse Hessian, leveraging the statistical properties of the problem. This adjustment effectively accelerates convergence and enables the linear rate observed in our analysis. We thank the reviewer for this insightful suggestion, we have revised our manuscript by adding a Remark (Line 396 to Line 403) to clarify the critical differences between our method and first-order optimization methods.
>
> Regarding the possibility of failure to achieve linear convergence, Theorem 5.1 provides a high probability guarantee under the assumptions specified in the paper. For sensing matrices satisfying Assumption 1 and generated from a stochastic process (e.g., truncated Gaussian), there is indeed a nonzero probability of failure, as the bound is probabilistic. Furthermore, as indicated by Theorem 5.1, achieving a linear convergence rate with a very high probability (which is $1-\delta$) requires the number of instances $n$ to scale as $-\log\delta$.
>
> **Question 6:** Assumption 1 seems non-standard compared with compressed sensing literature.
>
> **Response to Question 6:** We appreciate the reviewer’s observation. Indeed, Assumption 1 deviates from the conventional compressed sensing literature that relies directly on properties like the Restricted Isometry Property (RIP). In this work, we focus on bounded sensing matrices because they offer theoretical convenience when analyzing Transformers with ReLU activation and help prevent ill-conditioned scenarios that could lead to instability or exploding values. This assumption has also been adopted in related works, such as [4].
>
> We acknowledge that extending the analysis to unbounded sensing matrices, such as Gaussian matrices that do not satisfy this assumption, is an important direction. We thank the reviewer for highlighting this point, and we plan to incorporate this perspective in our ongoing and future work.
>
> **References:**
>
> [1] Gregor, Karol, and Yann LeCun. "Learning fast approximations of sparse coding." Proceedings of the 27th international conference on international conference on machine learning. 2010.
>
> [2] Chen, Xiaohan, et al. "Theoretical linear convergence of unfolded ISTA and its practical weights and thresholds." Advances in Neural Information Processing Systems 31 (2018).
>
> [3] Liu, Jialin, and Xiaohan Chen. "ALISTA: Analytic weights are as good as learned weights in LISTA." International Conference on Learning Representations (ICLR). 2019.
>
> [4] Bai, Yu, et al. "Transformers as statisticians: Provable in-context learning with in-context algorithm selection." Advances in neural information processing systems 36 (2024).

---

> ### Author Response · Authors · 2024-11-25
> **Thank you for your reply!**
>
> Dear Reviewer bS8s,
>
> Thank you very much for your valuable feedback on our rebuttal! We are glad that our responses have helped improve your understanding, and we appreciate that you recognize the contributions of this work and have maintained a positive score!
>
> We agree with you that the L2O framework refers to learning a solver to solve an optimization problem. In this paper, the optimization problem we consider is LASSO. We will make this point clearer in the revision.
>
> Thank you for clarifying your suggestion of "adding more layers during inference" and pointing us to the reference. We are running experiments to test this idea and will report the results when they are ready.
>
> We thank you again for your constructive feedback to help us improve the quality of this work!
>
> Warm regards,
>
> The Authors

---

### Official Review · Reviewer_iC5a · 2024-11-04

**Soundness:** 4
**Presentation:** 4
**Contribution:** 4
**Rating:** 8
**Confidence:** 4

**Summary:**

This paper presents a theoretical study of transformers to address the hypothesis regarding in-context learning for sparse recovery problems, as discussed in lines 62-65. Transformers are an integral part of modern deep learning architectures, and their contributions cannot be overstated. In particular, this work focuses on making theoretical claims that transformers can perform LISTA-type algorithms. The authors support this with a rigorous theoretical claim in Theorem 5.1, which demonstrates with high probability that transformers can recover the optimal sparse vector. Lastly, the authors empirically validate their claim in Figure 1.

**Strengths:**

**Strengths:**

1. **Strong Theoretical Contributions:**
   The paper presents a compelling theoretical claim addressing a crucial aspect of machine learning with significant potential for diverse applications. Moreover, the proof techniques introduced are versatile and can be extended to other domains beyond sparse vector recovery, such as compressed sensing and various inverse problems.

2. **Clarity and Rigor of Proofs:**
   The proofs are well-written and easy to follow, both in the main text and in the appendix.

3. **Robust Empirical Validation:**
   The empirical experiments are thorough and effectively corroborate the theoretical findings

**Weaknesses:**

The hypothesis outlined in lines 62-65 appears somewhat disconnected from the application to sparse recovery. The connection between in-context learning (ICL) and sparse recovery is not clearly established, making it challenging to understand why this particular relationship is being explored.

**Questions:**

1). Could the authors clarify their rationale for choosing to study this problem in the context of sparse vector recovery and explain how it relates to the hypothesis presented in this work?

---

> ### Author Response · Authors · 2024-11-22
> **Response to Reviewer iC5a**
>
> We thank the reviewer for the favorable rating and thoughtful comments. Please see our responses below.
>
> **Weakness 1:** The hypothesis outlined in lines 62-65 disconnected from the application to sparse recovery.
>
> **Response to Weakness 1:** We appreciate the reviewer’s helpful comment. The motivation for applying our hypothesis to sparse recovery stems from the problem’s inherent complexity and its potential to demonstrate the L2O capabilities of Transformers’ during ICL. Unlike simpler tasks such as linear regression, which achieves **linear** convergence with gradient descent, sparse recovery introduces unique challenges due to its sparsity constraints. These constraints typically lead to **sublinear** convergence with vanilla gradient descent but allow for **provably linear** convergence with L2O algorithms. Given the well-established theoretical and empirical performance of L2O algorithms in solving sparse recovery problems, we use sparse recovery as a showcase to demonstrate the L2O capacity of Transformers and validate our hypothesis. We have clarified this in the revision (Line 69 to Line 76) to make this motivation clearer and better connected to the rest of the paper.
>
>
> **Question 1:** Could the authors clarify their rationale for choosing in the context of sparse vector recovery and explain how it relates to the hypothesis presented in this work?
>
> **Response to Question 1:** Please refer to our response to Weakness 1.

---

> ### Comment · Reviewer_iC5a · 2024-11-25
> **Response**
>
> Thank you for responding and clarifying. Based on the careful responses of the authors I will keep my score the same.

---

> > ### Author Response · Authors · 2024-11-26
> > **Thank you for your response**
> >
> > Dear reviewer iC5a,
> >
> > We sincerely appreciate your helpful comments, positive feedback and keeping your positive score. Thank you once again for your valuable input and for contributing to enhancing the quality of our work!
> >
> > Best regards,
> >
> > Authors

---

### Meta-Review · Area_Chair_Me55 · 2024-12-20

**Metareview:**

The authors show that sparse recovery can be solved using in-context learning (ICL) with transformer models. Specifically, they show that transformers with ICL can implement LISTA-style sparse recovery algorithms, and that the convergence rate of the recovery algorithm is linear in the depth of the transformer model. There also appear to be numerical benefits compared to older algorithms such as LISTA and ALISTA.

The paper is very nicely written and the contributions are clear. The results are not terribly surprising and follow a fairly logical next step pointed by the existing literature: there have been a host of recent papers showing that transformers can do gradient descent with ICL; LISTA is just gradient descent with learnable parameters interleaved with soft thresholding; and soft thresholding can be implemented using the ReLU layers in the MLPs of transformers. The results on convergence rates seem to also be mostly a consequence from previous papers. Nonetheless, it takes effort to make everything work, and the authors do a good job putting all the building blocks together in a clear manner.

**Additional Comments On Reviewer Discussion:**

There was an extensive and productive discussion between the reviewers and the authors, primarily focusing on  I recommend that the authors reflect this discussion while preparing the final version of the manuscript.

---

### Decision · Program_Chairs · 2025-01-22

Accept (Poster)